# LORA-MOO: Learning Ordinal Relations and Angles for Expensive Many-Objective Optimization

## Abstract

Many-objective optimization (MOO) simultaneously optimizes many conflicting objectives to identify the Pareto front - a set of diverse solutions that represent different optimal balances between conflicting objectives. For expensive MOO problems, due to their costly function evaluations, computationally cheap surrogates have been widely used in MOO to save evaluation budget. However, as the number of objectives increases, the cost of learning and surrogation, as well as the difficulty of maintaining solution diversity, increases rapidly. In this paper, we propose LORA-MOO, a surrogate-assisted MOO algorithm that learns surrogates from spherical coordinates. This includes an ordinal-regression-based surrogate for convergence and $M - 1$ regression-based surrogates for diversity. $M$ is the number of objectives. Such a surrogate modeling method makes it possible to use a single ordinal surrogate to do the surrogate-assisted search, and the remaining surrogates are used to select solution for expensive evaluations, which enhances the optimization efficiency. The ordinal regression surrogate is developed to predict ordinal relation values as radial coordinates, estimating how desirable the candidate solutions are in terms of convergence. The solution diversity is maintained via angles between solutions, which is a parameter-free. Experimental results show that LORA-MOO significantly outperforms other surrogate-assisted MOO methods on most MOO benchmark problems and real-world applications.

## 1 Introduction

Many-objective optimization problems (MOOPs) are widely exist in many real-world applications, such as production scheduling [26], traffic signal control [33], and water resource engineering [21]. These MOOPs have conflicting objectives to optimize, and thus all objectives cannot reach their optimum simultaneously. As a result the optimum of MOOPs is the *Pareto front (PF)*: A set of non-dominated solutions that represent different optimal balance between conflicting objectives. Multi-/many-objective optimization (MOO) [1] aims to find non-dominated solutions that are close to the PF and also well distributed along the PF, indicating that MOO should consider both convergence and diversity.

Various evolutionary optimization algorithms have been proposed to solve MOOPs [10]. These optimization algorithms usually require plenty of solution samplings and evaluations to find converged and diverse non-dominated solutions. However, in many real-world MOOPs, the evaluation of solution performance could be expensive [41]. In these expensive MOOPs, the evaluation budget only allows a limited number of solutions to be evaluated on the expensive objective functions. To address expensive MOOPs, evolutionary optimization algorithms are combined with computationally cheap

---

[1]Multi-objective optimization has 2 or 3 objectives, many-objective optimization has 4 or more objectives.

surrogates to enhance sampling efficiency and save evaluations, which are known as surrogate-assisted evolutionary algorithms (SAEAs).

Yet, it is a perennial challenge to use surrogates in a more effective and efficient way for SAEAs, especially when optimization problems have many objectives. For example, conventional SAEAs usually use regression-based surrogates to approximate each objective function separately [5, 34]. For MOOPs, many objectives indicate maintaining many surrogates for surrogate-assisted search and selection, which results in a low efficiency of SAEAs. In addition, it is difficult to maintain solution diversity in high-dimensional objective space. Some SAEAs [24, 43, 5] need to investigate proper parametric strategies to generate reference vectors or divide objective space into subspaces. Recently, a family of classification-based SAEAs [31, 17] attempted to use a single surrogate to learn pairwise dominance relations. However, the training with pairwise relations implies an exponential increase in the size of training dataset. Therefore, a natural question is that whether we can reduce the cost of maintaining many surrogates without increasing the cost of training a single surrogate. Furthermore, whether we can use an non-parametric diversity maintenance strategy to handle the objective space of MOOPs, instead of designing complex reference vectors or points?

In this paper, we propose a different way to implement surrogate-assisted evolutionary optimization for expensive MOOPs, named LORA-MOO, where a single surrogate is developed to learn ordinal relations for convergence purpose, and several angular surrogates are generated from spherical coordinates to maintain diversity. Our major contributions are summarized as follows:

- We develop a novel ordinal-regression-based model to approximate the ordinal landscape of expensive MOOPs. Our ordinal surrogate is able to handle many objectives simultaneously and assist MOO algorithms to complete the model-based search. Artificial ordinal relations are generated via a clustering method to improve the learning quality of ordinal relations for many objectives. Unlike the pairwise relations learned through classification, the ordinal relations would not increase the size of training dataset, hence high efficiency.

- We introduce the idea of spherical coordinates approximation into surrogate-assisted evolutionary optimization and proposed LORA-MOO to solve expensive MOOPs. Different from existing SAEAs which learn approximation models from Cartesian coordinates, we fit several regression-based surrogates to approximate angular coordinates, while our ordinal surrogate can be treated as a radial coordinate. An non-parametric approach is developed to select diverse solutions for expensive evaluations via our angular coordinate surrogates.

- Extensive experiments on benchmark and real-world optimization problems are conducted under a range of scales and numbers of objectives. Empirical results show that our LORA-MOO is effective. It is able to obtain a well-distributed solution set that outperforms the state-of-the-arts.

## 2 Related Work

### 2.1 Multi-/Many-Objective Surrogate-Assisted Evolutionary Algorithms

**Regression-based SAEAs.** Regression-based SAEAs employ regression-based surrogates such as Kriging [36, 39] to approximate either the objective values of solutions or the objective functions of expensive problems [22]. To maintain solution diversity, ParEGO [24] employs a Kriging model to iteratively approximate an aggregate objective function which aggregates all objectives into one via a set of pre-defined scale vectors. In MOEA/D-EGO [43], plenty of scale vectors are generated uniformly to decompose the target MOOP into many single-objective subproblems. K-RVEA [5] also designs a set of scale vectors as reference vectors to maintain solution diversity. Similarity or density estimation is an alternative option for maintaining diversity. For instance, KTA2 [34] estimates the distribution status of non-dominated solutions by defining a similarity or density indicator.

**Classification-based SAEAs.** In model-based optimization, the optimization is guided by the relation between solutions rather than accurate objective values. Therefore, there is a tendency for recently proposed SAEAs to use classification-based surrogates to learn the relation between solutions directly. CSEA [31] trains a neural network to justify whether candidate solutions can be dominated by given reference points or not. $\theta$-DEA-DP [42] uses two neural networks to predict the Pareto dominance relation and $\theta$-dominance relation between two solutions, respectively. REMO [17] employs a neural network to fit a ternary classifier, which is able to learn the dominance relation between

pairs of solutions. Compared with regression-based SAEAs, although classification-based SAEAs take advantage of learning solution relations directly, their drawbacks are also clear: The prediction of solution relations lacks the information of how solutions are distributed in the objective space, making it difficult for classification-based SAEAs to maintain solution diversity. In [31, 17], a radial projection selection approach is adapted to select diverse reference points. However, its effect on diversity maintenance is limited. In addition, although classification-based SAEAs maintain only one surrogate, the cost of learning pairwise relations from large datasets is inevitably increased.

**SAEAs based on Other Surrogates.** HSMEA [15] uses an ensemble of multiple surrogates in the optimization. In addition, a new category of surrogates, namely ordinal regression surrogate [40] or level-based classification surrogate [28], is proposed recently to combine regression-based surrogates with classification-based surrogates. However, the shortcoming remains the same as these surrogates lack the information of solution distribution, especially when the number of objectives is large.

## 2.2 Multi-Objective Bayesian Optimization

**MOBO.** Bayesian Optimization (BO) [35, 18] is also a typical model-based optimization method for expensive optimization, while multi-objective BO (MOBO) methods are designed for expensive MOOPs [7, 8, 27, 1]. Some MOBO generalizes the acquisition functions such as upper confidence bound (UCB) [46], expected improvement (EI) [14], Thompson sampling [3], to solve expensive MOOPs. In addition, entropy search methods have also been employed in MOBO [2, 37]. To maintain solution diversity, the EI of a multi-objective performance indicator, Hypervolume (HV) [45], was used as the acquisition function in recent MOBO [6, 27]. Based on the Hypervolume improvement (HVI), PSL [27] proposes a learning method to approximate the whole Pareto set for MOBO, and PDBO [1] automatically selects the best acquisition function for objective functions in each iteration. However, the time complexity of computing HV increases exponentially with the number of objectives, which may limit the application of MOBO methods on optimization problems with many objectives.

**Connection to SAEAs.** Both SAEAs and MOBO are model-based optimization methods. A SAEA is also a MOBO if it uses probability models as surrogates, and a MOBO is also a SAEA if it searches candidate solutions with evolutionary search algorithms. Therefore, some model-based optimization methods belong to both SAEAs and MOBO [24, 14, 43].

# 3  LORA-MOO: Optimization via Learning Ordinal Relations and Angles

This section first introduces the LORA-MOO framework, followed by detailed algorithm descriptions.

## 3.1  LORA-MOO Framework

The pseudocode of LORA-MOO is depicted in Alg. 1, it consists of four phases:

1. Initialization: An initial dataset of size $11D$ - 1 (As suggested in the literature [24]) are sampled from the decision space using the Latin hypercube sampling (LHS) [30] (line 1), where $D$ is the dimensionality of decision variables. The sampled solutions are evaluated on objective functions $f$ and then saved in an archive $S_A$ (line 2).

2. Surrogate modeling: For all solutions $x \in S_A$, quantify their ordinal values (line 4) and calculate their angular coordinates (line 9). The set of ordinal values $S_o$ is used to train the ordinal surrogate $h_o$ (line 5). The angular coordinates are used to fit $M - 1$ angular surrogates $h_{ai}$ separately (line 10).

3. Sampling (Search and Selection): Run an optimizer on surrogate $h_o$ to generate a population of candidate solutions $P$ (line 6). Select optimal candidate solutions $x_1^*$, $x_2^*$ from $P$ based on surrogates $h_o$, $h_{ai}$, respectively (lines 7 and 11).

4. Update: Evaluate new optimal candidate solutions $x_1^*$, $x_2^*$ on expensive objective functions $f$, update archive $S_A$ and the number of used function evaluations $FE$ (lines 8 and 12). The algorithm will go to phase 2 until the evaluation budget $FE_{max}$ has run out.

**Algorithm 1** LORA-MOO framework

---

**Input:** $M$ objective functions of the optimization problem $f(\boldsymbol{x}) = (f_1(\boldsymbol{x}), \ldots, f_M(\boldsymbol{x}))$;
      Evaluation budget: The number of allowed function evaluations $FE_{max}$.

**Procedure:**
1: Sample a set of solutions $\{\boldsymbol{x}_1, \ldots, \boldsymbol{x}_{11D-1}\}$ and evaluate them on $f$.
2: Save all evaluated solutions $(\boldsymbol{x}, f(\boldsymbol{x}))$ in an archive $S_A$. Set the number of used function evaluations $FE = |S_A|$.
3: **while** $FE < FE_{max}$ **do**
4:     Ordinal training set $S_o \leftarrow$ Quantify ordinal values for all $\boldsymbol{x}_i \in S_A$ (Alg. 2).
5:     Ordinal surrogate $h_o \leftarrow$ Train Kriging$(S_A, S_o)$.
6:     Population of candidate solutions $P \leftarrow$ Run an optimizer on $h_o$ (Alg. 3).
7:     $\boldsymbol{x}_1^* \leftarrow$ Use the ordinal surrogate to select a solution from $P$ by convergence criterion.
8:     Evaluate $\boldsymbol{x}_1^*$ and update $S_A = S_A \cup \{(\boldsymbol{x}_1^*, f(\boldsymbol{x}_1^*))\}$, $FE = FE + 1$.
9:     Angular training set $S_a \leftarrow$ Calculate angular coordinates for all $\boldsymbol{x}_i \in S_A$.
10:    $M$-1 angular surrogates $h_{ai} \leftarrow$ Train Kriging $(S_A, S_a), i = 1, \ldots, M - 1$.
11:    $\boldsymbol{x}_2^* \leftarrow$ Use angular surrogates to select a solution from $P$ by diversity criterion (Alg. 4).
12:    Evaluate $\boldsymbol{x}_2^*$ and update $S_A = S_A \cup \{(\boldsymbol{x}_2^*, f(\boldsymbol{x}_2^*))\}$, $FE = FE + 1$.
13: **end while**

**Output:** Non-dominated solutions in archive $S_A$.

---

## 3.2 Surrogate Modeling

The ordinal surrogate $h_o$ is mainly trained on dominance-based ordinal relations, additional clustering-based artificial ordinal relations will be introduced for training if the number of objectives $M$ is large. In addition, for an $M$-objective problem, $M$-1 angular surrogates $h_{ai}$ are trained on angular coordinates. These surrogates are used in the selection procedure for solution diversity but are idle in the search procedure.

### 3.2.1 Learning dominance-based ordinal relations.

In LORA-MOO, the concept of ordinal regression [40] is adapted to learn dominance-based ordinal relations. Clearly, the dominance-based ordinal relation between a set of reference points $S_{RP}$ and a given solution $\boldsymbol{x}$ is quantified as a relation value. Such a relation value is a numerical value that used for training the ordinal-regression surrogate $h_o$. The quantification of relation values consists of two steps: The selection of reference points $S_{RP}$ and the computation of relation values.

**Selection of Reference Points.** We propose the definition of $\lambda$-dominance relationship to simplify the selection of reference points.

**Definition 1.** *($\lambda$-Dominance Relationship)*
*A solution $\boldsymbol{x}^1$ is said to $\lambda$-dominate another solution $\boldsymbol{x}^2$ (denoted by $\boldsymbol{x}^1 \prec_\lambda \boldsymbol{x}^2$) if and only if:*

$$g_\lambda(\boldsymbol{x}^1) \prec g_\lambda(\boldsymbol{x}^2), \tag{1}$$

*where $\lambda \geq 0$ is the dominance coefficient and $g_\lambda$ is a smooth objective function defined as:*

$$f_{in}(\boldsymbol{x}) = \frac{f_i(\boldsymbol{x}) - z_i^*}{|z_i^{nad} - z_i^*|}, \tag{2}$$

$$g_{\lambda,i}(\boldsymbol{x}) = f_{in}(\boldsymbol{x}) + \lambda max(f_{jn}(\boldsymbol{x})), j \in \{1, \ldots, M\}, \tag{3}$$

*where $f_{in}$ denotes a normalized objective function, $\boldsymbol{z}^* = \{z_1^*, \ldots, z_M^*\}$, $\boldsymbol{z}^{nad} = \{z_1^{nad}, \ldots, z_M^{nad}\}$ are ideal point and nadir point for the current non-dominated solutions, respectively.*

More detailed definitions about the background of MOO are available in Appendix A. All non-$\lambda$-dominated solutions in $S_A$ are selected as reference points $S_{RP}$. There are two reasons to introduce the definition of $\lambda$-dominance:

- The $\lambda$-dominance can smoothen the original PF by excluding dominance resistant solutions (DRSs) [16, 38]. DRSs are solutions that are best or close to best on one or several objectives but extremely poor on at least one of the remaining objectives. Such a solution is apparently not desirable but may be regarded as one of the best solutions since there may not exist any other solutions dominating it in the solution set.

- Second, $\lambda$-dominance can eliminate some similar non-dominated solutions from the Pareto set, which can be used to adjust the size of Pareto set. When the number of objectives $M$ is large, it is possible that a majority of past evaluated samples are non-dominated to each other. To balance the number of reference points and remaining samples, we introduce the dominance coefficient $\lambda$ to sightly reduce the ratio of reference points in $S_A$. This alleviates the situation of extreme imbalance of samples in different ordinal levels (see the division of ordinal levels below).

**Computation of Relation Values.** To quantify ordinal relation values, we first calculate extension coefficients $ec(\boldsymbol{x})$ for each $\boldsymbol{x} \in S_A$. $ec(\boldsymbol{x})$ is defined as the minimal coefficient $ec \geq 1$ to make a solution $\boldsymbol{x}$ non-$\lambda$-dominated to all solutions $\boldsymbol{x}'$ in the extended reference:

$$ec(\boldsymbol{x}) = \arg \min_{ec \geq 1} \nexists \boldsymbol{x}' \in S_{RP} : (\boldsymbol{x}' * ec) \prec_\lambda \boldsymbol{x}. \tag{4}$$

Although extension coefficient $ec(\boldsymbol{x})$ quantifies the distance between a solution $\boldsymbol{x}$ and reference $S_{RP}$, it has not been used to train the ordinal regression-based surrogate directly. To generate a stable ordinal regression-based surrogate, solutions in $S_A$ are divided into $N_o = max(n_o, |S_A|/|S_{RP}|)$ ordinal levels, where $n_o$ is a pre-defined parameter denoting the minimal number of ordinal levels. The solutions in $S_{RP}$ are classified into the non-dominated ordinal level, thus the relation value $v_1 = 1.0$ is assigned to them. Remaining solutions in $S_A$ are sorted by their extension coefficients $ec(\boldsymbol{x})$ and then divided into $N_o$-1 ordinal levels uniformly. The relation value $v_i = 1 - \frac{i-1}{N_o-1}$ will be assigned to the solutions $\boldsymbol{x}$ in the $i^{th}$ ordinal level. Lastly, relation values serve as radial coordinates and a Kriging model is employed to approximate them.

### 3.2.2 Artificial clustering-based ordinal relations.

When the number of objectives $M$ is large, most evaluated solutions in archive $S_A$ could be non-dominated solutions, indicating that these solutions will be divided into the same non-dominated ordinal level and thus treated as reference points $S_{RP}$. This is harmful to the ordinal surrogate modeling due to the extreme imbalance between the numbers of training samples in different ordinal levels. To reduce the ratio of $S_{RP}$, we use a clustering method to generate $n\_clusters$ clusters for $S_{RP}$, where $n\_clusters$ is the half of the size of $S_{RP}$. All solutions $\boldsymbol{x} \in S_{RP}$ are mapped to the closest cluster centers. The solutions with the shortest projection on each cluster center will be selected as the new $S_{RP}$, while the remaining solutions will be moved to the next ordinal level. Such artificial ordinal relations greatly reduce the ratio of $S_{RP}$ in $S_A$. In LORA-MOO, we set a ratio threshold $rp\_ratio$ for $S_{RP}$, once the ratio of $S_{RP}$ is larger than $rp\_ratio$, artificial ordinal relations will be generated for surrogate modeling. Details are available in Appendix C, Alg. 2 and Fig. 5.

### 3.2.3 Surrogates for Angular Coordinates.

Given a solution $\boldsymbol{x} \in S_A$ with Cartesian coordinates $(f_1(\boldsymbol{x}), \ldots, f_M(\boldsymbol{x}))$, The angular coordinates of solution $\boldsymbol{x}$ are transformed with the following rules:

$$\varphi_i = arccos \frac{f_i(\boldsymbol{x}) - z_i^*}{\sqrt{(f_i(\boldsymbol{x}) - z_i^*)^2 + \cdots + (f_M(\boldsymbol{x}) - z_M^*)^2}}, i = 1, \ldots, M - 1, \tag{5}$$

where $\boldsymbol{z}^*$ is the ideal point. The resulting angular coordinates $(\varphi_1, \ldots, \varphi_{M-1})$ are used to fit $M - 1$ regression-based surrogates separately. In LORA-MOO, we use the Kriging model to approximate angular coordinates. The introduction and usage of Kriging model is given in Appendix B.

## 3.3 Sampling: Search and Selection

In this subsection, we describe how to use surrogate $h_o$ to search for candidate solutions and how to use surrogates $h_o$ and $h_{ai}$ to select optimal ones from candidate solutions for expensive evaluations.

### 3.3.1 Search: Generation of Candidate Solutions.

An advantage of LORA-MOO is that it searches for candidate solutions on ordinal surrogate $h_o$ only, leaving all angular surrogates $h_{ai}$ idle in this search procedure. This saves a lot of time from predicting with all surrogates. LORA-MOO employs an optimizer (e.g. PSO [13]) to generate a

population of candidate solutions $P$ (Detailed pseudo-code is available in Appendix C, Alg. 3). The initial population for optimization search consists of two parts. The first half initial solutions are generated randomly from the decision space, while the remaining initial solutions are mutants of current reference points $S_{RP}$. To ensure the diversity of initial candidate solutions, a KNN clustering method is applied to divide $S_{RP}$ into several different clusters, from each cluster, an equal number of mutants are generated as initial candidate solutions. The global optimal population $P$ produced by PSO is the candidate solutions for further environmental selection.

### 3.3.2 Selection Criteria.

To take both convergence and diversity into consideration, in each iteration, LORA-MOO selects two optimal candidate solutions $\boldsymbol{x}_1^*, \boldsymbol{x}_2^*$ from $P$ for objective function evaluations. $\boldsymbol{x}_1^*, \boldsymbol{x}_2^*$ are sampled on the basis of convergence and diversity, respectively.

**Convergence Criterion** for environmental selection is the expected improvement (EI) [14] of ordinal values, which is similar to many MOBO methods [24, 43]. Since the output of our ordinal surrogate $h_o(\boldsymbol{x})$ is an 1-D numerical value, the solution with maximal 1-D EI in $P$ is selected as $\boldsymbol{x}_1^*$.
**Diversity Criterion** to sample $\boldsymbol{x}_2^*$ from $P$ is defined as angles $d_{ang}$ between candidate solutions and reference points $S_{RP}$. Firstly, the minimal degree between each candidate solution and $S_{RP}$ is measured. Among these minimal degrees $md_{ang}$, the solution with $\max(md_{ang})$ is selected as $\boldsymbol{x}_2^*$ (Detailed pseudo-code is available in Appendix C, Alg. 4).

## 4 Experiments

To evaluate the optimization performance of LORA-MOO on expensive MOOPs, we conduct experiments to compare LORA-MOO with other SAEAs on different MOOPs, including a series of scalable multi-/many-objective benchmark optimization problems DTLZ [11], WFG [19], and a real-world network architecture search (NAS) problem.

### 4.1 Experimental Setups

**Optimization Problem Setup.** To ensure a fair comparison, the following optimization problem setup is the same as the setup that has been widely used in the literature [5, 31, 34, 17]. In our experiments, initial datasets of size $FE_{init}$ = 11 $D$ - 1 are used to initialize surrogates, while the maximum number of allowed evaluations $FE_{max}$ is 300. The statistical results are obtained from 30 independent runs. For each run, different comparison algorithms share the same initial dataset.
**Comparison Algorithms.** We compare LORA-MOO with 6 state-of-the-art SAEAs, some of them also known as MOBO methods. These comparison algorithms can be classified into three categories:

- Regression-based MOO methods: ParEGO [24], K-RVEA [5], and KTA2 [34]. ParEGO is a classic regression-based SAEA and also a MOBO, which serves as a baseline. K-RVEA is a typical SAEA which uses reference vector to guide the diversity maintenance. KTA2 is a newly proposed algorithm to use an independent archive to keep solution diversity.

- Classification-based MOO methods: CSEA [31], REMO [17]. CSEA is a classic classification-based SAEA which serves as a baseline. REMO is a newly proposed SAEA which represents the state-of-the-art performance of classification-based SAEAs.

- Ordinal-regression-based MOO method: OREA [40] is a new category of SAEA that is different from common regression-based and classification-based SAEAs. We compare with it since it is directly related to our radial surrogate.

Note that some classic SAEAs and MOBO methods such as MOEA/D-EGO [43] and CPS-MOEA [44] are not compared in our experiments as they failed to outperform other comparison algorithms on any DTLZ problem [17]. Some HV-based MOBO methods are not compared as they are failed to solve many objectives.
**Parameter Setup.** For the surrogate modeling, the Kriging models used in all comparison algorithms are implemented using DACE [32], just as [24] suggested. For regression-based Kriging surrogates, the range of hyper-parameter $\theta \in [10^{-5}, 100]$. And for the neural networks in CSEA and REMO, the parameters are the same as suggested in the literature. In the sampling strategy, the mutation operator used to initialize candidate solutions is polynomial mutation [9], the mutation probability $p_m = 1/d$

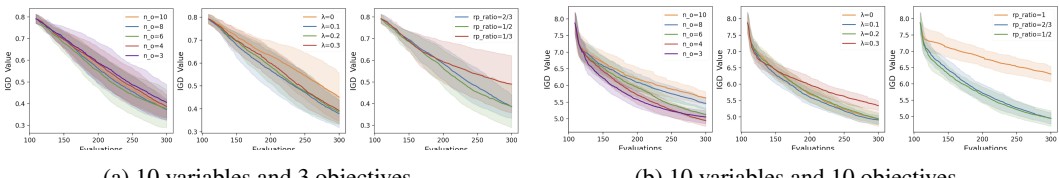

(a) 10 variables and 3 objectives.        (b) 10 variables and 10 objectives.

Figure 1: IGD curves averaged over 15 runs on the WFG5 problem instances for LORA-MOO with different parameter setups (shaded area is $\pm$ std of the mean).

and mutation index $\eta_m = 20$, as recommended in [34, 17]. The size of offspring population is 100. The settings of the PSO optimizer are the range of hyper-parameter in the ordinal-regression-based surrogate are the same as suggested in [40].

For the specific parameters exist in LORA-MOO, such as the dominance coefficient $\lambda$ and the threshold ratio of reference points to introduce clustering-based ordinal relations $rp\_ratio$. As there is no relevant study in the literature for their setups, we conducted ablation studies to investigate the effect of these parameters on the performance of LORA-MOO. The results are summarized in Section 4.2 and reported in Appendix F. The source code of LORA-MOO [2] will be available online.

**Performance Indicator.** To have a comprehensive estimation of optimization performance, we use three different performance indicators in our experiments: The inverted generational distance (IGD) [4], the inverted generational distance plus (IGD+) [20], and the Hypervolume (HV) [45]. IGD and IGD+ use a set of truth Pareto front to measure the quality of a set of non-dominated solutions in terms of convergence and diversity. A smaller IGD or IGD+ value indicates better MOO performance. HV use a reference point to calculate the area covered by a set of non-dominated solutions, a large HV value is preferable to MOO. See Appendix D for details and setups about performance indicators.

## 4.2 Ablation Studies

We conduct ablation studies on DTLZ and WFG benchmark problems with $D$ = 10 variables and $M$={3, 6, 10} objectives. LHS [30] is used to sample initial dataset. The effects of four parameters are investigated: They are the minimal number of ordinal levels $n_o$, the dominance coefficient $\lambda$, the ratio threshold of reference points $rp\_ratio$, and the clustering number for reproduction $n_c$. Three representative results obtained on the WFG5 problem with 3 and 10 objectives are depicted in Fig. 1. Complete results and statistical analysis of ablation studies are reported in Appendix F.

As shown in Fig. 1 (left), when $M$ = 10, a large $n_o$ results in poor optimization performance. This is because the ratio of non-dominated solutions in the archive tends to be large when $M$ is large, hence, setting a large $n_o$ will lead to a lack of training samples in each dominated ordinal levels, which is detrimental to the performance of surrogate modeling. As such, $n_o$ in LORA-MOO is set to 4.

The result in Fig. 1 (middle) shows that using $\lambda$-dominance to sightly modify the original dominance relations is beneficial to the effectiveness of LORA-MOO. When $\lambda = 0$, no $\lambda$-dominance would be used and the corresponding LORA-MOO variant has the worst performance among all the variants. In addition, setting a large $\lambda$ could cause severe damage to the original dominance relations. Therefore, we set $\lambda$ to 0.2.

The effect of introducing artificial ordinal relations via clustering is demonstrated in Fig. 1 (right). When the ratio threshold of reference points $rp\_ratio$ is 1 and $M$ = 10, no artificial ordinal relations are introduced to further divide ordinal levels for plenty of non-dominated solutions in the archive. Consequently, the imbalance of sample numbers in different ordinal levels leads to poor optimization performance. However, dominance relations are preferable to artificial ordinal relations when $M$ = 3 and the size of ordinal levels are well balanced. Hence, we set $rp\_ratio$ = 0.5.

## 4.3 Optimization on Benchmark Problems

The optimization performance of LORA-MOO is evaluated on DTLZ and WFG benchmark problems with $D$ = 10 variables and $M$={3, 4, 6, 8, 10} objectives. The IGD values obtained on DTLZ

---

[2] The link of code and data will be released here once the paper is accepted.

Table 1: Statistical results of the IGD value obtained by the comparison algorithms on the 35 DTLZ optimization problems over 30 runs. Symbols '+', '≈', '−' denote LORA-MOO is statistically significantly superior to, equivalent to, and inferior to the compared algorithms in the Wilcoxon rank sum test (significance level is 0.05), respectively. The last three rows are the total win/tie/loss results on DTLZ, WFG, and both of them, respectively.

| Problems | M | ParEGO | KRVEA | KTA2 | CSEA | REMO | OREA | LORA-MOO (ours) |
|---|---|---|---|---|---|---|---|---|
| DTLZ1 | 3 | 5.98e+1(3.81e+0)+ | 8.88e+1(2.16e+1)+ | 4.75e+1(1.55e+1)≈ | 6.30e+1(1.69e+1)+ | 5.06e+1(1.49e+1)+ | 4.44e+1(1.38e+1)≈ | 4.35e+1(1.80e+1) |
| | 4 | 4.68e+1(3.71e+0)+ | 6.45e+1(1.47e+1)+ | 4.08e+1(1.60e+1)+ | 3.69e+1(1.08e+1)≈ | 3.92e+1(1.11e+1)+ | 3.80e+1(1.23e+1)+ | 4.06e+1(1.34e+1) |
| | 6 | 3.04e+1(2.74e+0)+ | 3.22e+1(7.66e+0)+ | 2.03e+1(8.12e+0)≈ | 1.56e+1(4.96e+0)≈ | 1.22e+1(4.65e+0)− | 1.74e+1(3.98e+0)≈ | 1.58e+1(6.17e+0) |
| | 8 | 1.23e+1(2.99e+0)+ | 8.52e+0(2.97e+0)+ | 4.54e+0(2.66e+0)+ | 5.08e+0(2.47e+0)≈ | 3.33e+0(1.93e+0)≈ | 5.87e+0(2.91e+0)+ | 3.83e+0(2.35e+0) |
| | 10 | 4.37e-1(1.63e-1)+ | 3.32e-1(9.91e-2)+ | 3.00e-1(8.76e-2)+ | 2.90e-1(7.13e-2)+ | 2.42e-1(6.97e-2)≈ | 2.58e-1(6.33e-2)≈ | 2.31e-1(3.89e-2) |
| DTLZ2 | 3 | 3.38e-1(2.84e-2)+ | 1.32e-1(2.77e-2)+ | 6.17e-2(3.13e-3)+ | 2.26e-1(2.61e-2)+ | 1.65e-1(2.18e-2)+ | 8.59e-2(8.51e-3)+ | 6.19e-2(3.48e-3) |
| | 4 | 4.23e-1(2.79e-2)+ | 2.06e-1(2.95e-2)+ | 1.41e-1(5.45e-3)+ | 2.92e-1(1.89e-2)+ | 2.43e-1(2.33e-2)+ | 1.83e-1(1.37e-2)+ | 1.38e-1(9.86e-3) |
| | 6 | 5.53e-1(2.17e-2)+ | 3.40e-1(1.20e-2)+ | 3.24e-1(2.63e-2)+ | 4.42e-1(3.37e-2)+ | 3.77e-1(3.16e-2)+ | 3.96e-1(2.57e-2)+ | 2.67e-1(8.78e-3) |
| | 8 | 6.53e-1(1.86e-2)+ | 4.19e-1(2.65e-2)+ | 4.44e-1(1.86e-2)+ | 5.95e-1(2.77e-2)+ | 5.10e-1(3.90e-2)+ | 5.56e-1(2.19e-2)+ | 3.80e-1(1.46e-2) |
| | 10 | 6.95e-1(2.23e-2)+ | 5.92e-1(4.25e-2)+ | 4.50e-1(1.00e-2)≈ | 6.76e-1(2.52e-2)+ | 5.85e-1(3.72e-2)+ | 6.55e-1(2.66e-2)+ | 4.54e-1(1.41e-2) |
| DTLZ3 | 3 | 1.66e+2(1.31e+1)+ | 2.43e+2(4.61e+1)+ | 1.52e+2(4.73e+1)≈ | 1.62e+2(4.84e+1)≈ | 1.49e+2(3.88e+1)≈ | 1.26e+2(3.18e+1)− | 1.57e+2(3.83e+1) |
| | 4 | 1.42e+2(1.57e+1)+ | 1.83e+2(4.00e+1)+ | 1.18e+2(3.49e+1)+ | 1.29e+2(3.58e+1)≈ | 1.16e+2(3.00e+1)+ | 1.22e+2(4.13e+1)+ | 1.25e+2(4.20e+1) |
| | 6 | 9.17e+1(1.59e+1)+ | 1.06e+2(2.96e+1)+ | 6.65e+1(2.63e+1)+ | 5.27e+1(1.56e+1)≈ | 5.23e+1(1.71e+1)+ | 5.24e+1(1.68e+1)≈ | 5.96e+1(2.05e+1) |
| | 8 | 4.13e+1(9.84e+0)+ | 2.96e+1(1.15e+1)+ | 1.74e+1(1.10e+1)≈ | 1.60e+1(9.76e+0)≈ | 1.60e+1(7.70e+0)≈ | 1.50e+1(6.27e+0)≈ | 1.27e+1(8.33e+0) |
| | 10 | 1.36e+0(3.15e-1)+ | 1.23e+0(4.27e-1)+ | 9.95e-1(2.25e-1)+ | 1.01e+0(2.45e-1)+ | 9.53e-1(2.74e-1)+ | 8.77e-1(1.08e-1)+ | 8.14e-1(1.33e-1) |
| DTLZ4 | 3 | 6.70e-1(7.61e-2)+ | 3.32e-1(1.11e-1)+ | 3.49e-1(1.09e-1)+ | 4.62e-1(1.36e-1)+ | 2.31e-1(1.15e-1)+ | 2.39e-1(1.65e-1)+ | 1.89e-1(2.34e-1) |
| | 4 | 7.18e-1(6.40e-2)+ | 4.07e-1(8.73e-2)+ | 4.77e-1(9.70e-2)+ | 4.31e-1(6.36e-2)+ | 3.36e-1(7.02e-2)≈ | 3.45e-1(1.52e-1)≈ | 3.48e-1(1.60e-1) |
| | 6 | 7.06e-1(3.07e-2)+ | 5.04e-1(5.42e-2)+ | 6.05e-1(8.43e-2)+ | 4.94e-1(5.55e-2)+ | 4.97e-1(4.95e-2)+ | 4.47e-1(4.89e-2)≈ | 4.55e-1(6.53e-2) |
| | 8 | 6.81e-1(1.48e-2)+ | 5.49e-1(3.42e-2)+ | 6.24e-1(5.48e-2)+ | 5.85e-1(4.20e-2)+ | 6.16e-1(4.03e-2)+ | 5.29e-1(3.79e-2)≈ | 5.32e-1(2.38e-2) |
| | 10 | 6.77e-1(1.26e-2)+ | 6.07e-1(2.42e-2)+ | 6.36e-1(3.58e-2)+ | 6.38e-1(2.38e-2)+ | 6.71e-1(2.69e-2)+ | 5.90e-1(1.94e-2)≈ | 5.90e-1(2.51e-2) |
| DTLZ5 | 3 | 2.16e-1(4.45e-2)+ | 1.19e-1(3.38e-2)+ | 1.34e-2(2.83e-3)≈ | 1.18e-1(2.56e-2)+ | 7.36e-2(2.03e-2)+ | 2.02e-2(4.77e-3)+ | 1.26e-2(2.55e-3) |
| | 4 | 1.89e-1(3.70e-2)+ | 7.05e-2(2.25e-2)+ | 4.24e-2(8.84e-3)+ | 1.16e-1(2.23e-2)+ | 9.02e-2(2.48e-2)+ | 3.48e-2(7.82e-3)+ | 2.85e-2(9.37e-3) |
| | 6 | 1.41e-1(2.32e-2)+ | 3.53e-2(1.02e-2)− | 8.87e-2(1.91e-2)+ | 7.72e-2(2.57e-2)+ | 5.53e-2(1.90e-2)+ | 4.62e-2(1.50e-2)+ | 4.26e-2(1.11e-2) |
| | 8 | 7.72e-2(1.22e-2)+ | 1.99e-2(4.92e-3)− | 6.43e-2(8.60e-3)+ | 3.81e-2(1.03e-2)+ | 3.10e-2(7.33e-3)≈ | 2.59e-2(6.96e-3)− | 2.84e-2(4.88e-3) |
| | 10 | 2.25e-2(1.87e-3)+ | 1.25e-2(1.90e-3)+ | 2.04e-2(2.55e-3)+ | 1.27e-2(1.46e-3)+ | 9.35e-3(2.00e-3)− | 1.03e-2(1.62e-3)≈ | 1.06e-2(2.36e-3) |
| DTLZ6 | 3 | 3.15e-1(1.62e-1)+ | 3.06e+0(5.21e-1)+ | 1.83e+0(4.37e-1)+ | 4.86e+0(6.30e-1)+ | 4.27e+0(5.49e-1)+ | 3.09e-1(3.99e-1)+ | 1.18e-1(1.57e-1) |
| | 4 | 3.56e-1(2.12e-1)+ | 2.46e+0(3.84e-1)+ | 1.85e+0(5.06e-1)+ | 5.13e+0(4.23e-1)+ | 4.08e+0(6.16e-1)+ | 1.43e+0(8.89e-1)+ | 3.29e-1(2.22e-1) |
| | 6 | 2.66e-1(1.37e-1)− | 1.36e+0(2.73e-1)+ | 1.51e+0(5.85e-1)+ | 3.15e+0(4.35e-1)+ | 2.33e+0(5.70e-1)+ | 2.05e+0(6.16e-1)+ | 9.89e-1(1.02e+0) |
| | 8 | 1.61e-1(6.17e-2)+ | 5.28e-1(1.50e-1)+ | 8.64e-1(3.88e-1)+ | 1.56e+0(4.28e-1)+ | 9.64e-1(4.38e-1)+ | 1.06e+0(3.95e-1)+ | 3.56e-1(4.31e-1) |
| | 10 | 1.72e-1(1.45e-1)+ | 7.73e-2(3.13e-2)≈ | 1.01e-1(4.97e-2)+ | 2.09e-1(1.28e-1)+ | 7.91e-2(2.11e-1)+ | 1.50e-1(7.37e-2)+ | 7.05e-2(3.25e-2) |
| DTLZ7 | 3 | 2.45e-1(4.80e-2)+ | 1.35e-1(2.37e-2)+ | 2.19e-1(2.40e-1)+ | 1.75e+0(6.32e-1)+ | 1.27e+0(5.65e-1)+ | 2.73e-1(1.58e-1)+ | 2.01e-1(1.93e-1) |
| | 4 | 6.59e-1(1.02e-1)+ | 3.38e-1(7.61e-2)≈ | 3.73e-1(1.68e-1)+ | 2.94e+0(6.59e-1)+ | 2.06e+0(7.31e-1)+ | 8.92e-1(4.27e-1)+ | 4.20e-1(2.21e-1) |
| | 6 | 1.21e+0(1.58e-1)− | 6.04e-1(4.57e-2)− | 6.46e-1(1.68e-1)− | 4.92e+0(9.92e-1)+ | 3.09e+0(6.71e-1)+ | 4.03e+0(1.84e+0)+ | 1.71e+0(6.54e-1) |
| | 8 | 1.45e+0(1.24e-1)− | 8.71e-1(7.01e-2)− | 1.02e+0(1.65e-1)− | 6.12e+0(1.85e+0)+ | 3.82e+0(5.39e-1)+ | 4.55e+0(2.63e+0)+ | 2.44e+0(6.78e-1) |
| | 10 | 1.67e+0(1.24e-1)+ | 1.12e+0(4.25e-2)− | 1.30e+0(2.04e-1)+ | 1.99e+0(3.05e-1)+ | 1.99e+0(3.36e-1)+ | 1.63e+0(2.42e-1)+ | 1.34e+0(9.19e-2) |
| +/≈/− | on DTLZ | 30/2/3 | 27/3/5 | 19/13/3 | 28/7/0 | 23/10/2 | 20/13/2 | |
| +/≈/− | on WFG | 39/4/2 | 21/10/14 | 23/6/16 | 41/1/3 | 38/3/4 | 43/1/1 | |
| +/≈/− | on both | 69/6/5 | 48/13/19 | 42/19/19 | 69/8/3 | 61/13/6 | 63/14/3 | |

problems with different $M$ are reported in Table 1. It shows that LORA-MOO achieves the best optimization results among all the comparison algorithms in terms of IGD values, followed by KTA2 and KRVEA. The IGD values obtained on the WFG problems, the IGD+ and HV results, and the results obtained under different scales ($D$= 5 or 20) are reported in Appendix H. A consistent result can be concluded from the IGD+ and HV values. The results on the 3- and 10-objective problems are plotted in Fig. 2.

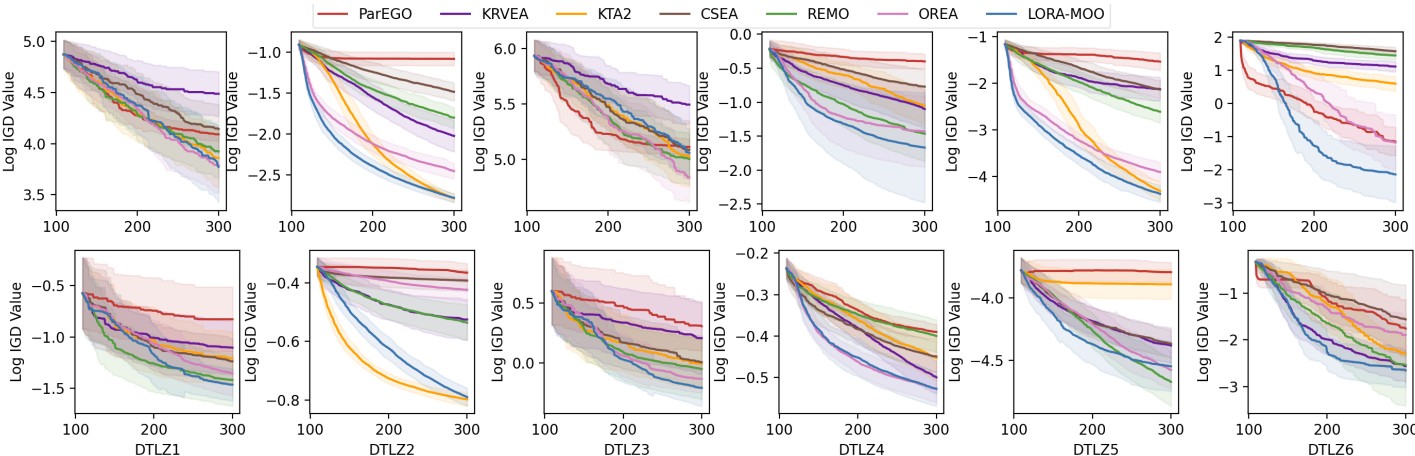

Figure 2: IGD(log) curves averaged over 30 runs on the DTLZ problems for the comparison algorithms (shaded area is ± std of the mean). **Top**: 10 variables and 3 objectives. **Bottom**: 10 variables and 10 objectives. More figures are displayed in Appendices G and H.

### 4.4 Real-World Network Architecture Search Problem

Further comparison is conducted on a real-world network architecture search (NAS) problem, the best three algorithms listed in Table 1 are compared: LORA-MOO, KTA2, and KRVEA. The NAS problem tested is the NASbench201 implemented in EvoXBench [29], it has 6 variables and 5 objectives. Details of this NAS problem is provided in Appendix E. Considering NASbench201 is a real-world application and we do not know its exact PF, we use HV to evaluate optimization performance since HV can be calculated without the exact PF. In practice, $log(HV_{\mathrm{diff}})$ is employed to amplify the visual difference of the obtained HV values:

$$log(HV_{\mathrm{diff}}) = log(HV_{\mathrm{max}} - HV)$$

where $HV_{\mathrm{max}}$ is the maximal HV value on this problem that is provided in EvoXBench.

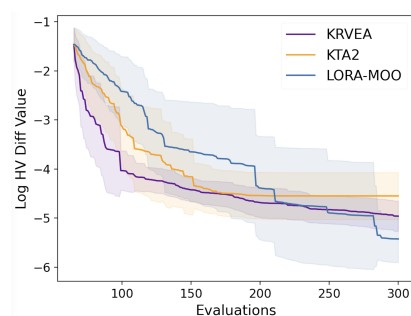

Fig. 3 plots the result. As can be seen in the figure, LORA-MOO outperforms KTA2 and KRVEA on this NAS problem. Although KTA2 and KRVEA have quicker convergence rate than LORA-MOO at the beginning of the optimization, both of them slow down their convergence speed as the number of evaluations increases. Particularly, KTA2 is trapped on local optima and thus fails to reach better results. In comparison, LORA-MOO reaches better NAS results when the evaluation number is larger than 250.

Figure 3: $Log(HV_{\mathrm{diff}})$ curves averaged over 30 runs on the NAS problem for the comparison algorithms.

### 4.5 Runtime Comparison

We compare the runtime on benchmark problems for all the comparison algorithms to investigate the relation between their optimization efficiency and the number of objectives $M$.

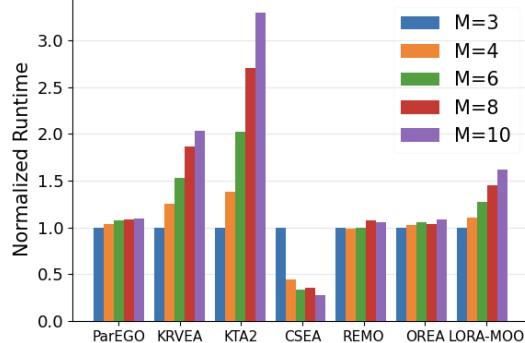

Fig. 4 illustrates how the runtime of each comparison algorithm varies as the $M$ increases. It can be observed that the runtime of KTA2 increases exactly in the same rate as $M$ increases. In comparison, the runtime of LORA-MOO increases slightly when $M$ increases. This demonstrates that using angular surrogates only at the end of environmental selection process is beneficial to the optimization efficiency of LORA-MOO. In addition, the runtimes of ParEGO, CSEA, REMO, and OREA do not increase significantly with $M$ since they do not maintain specific surrogates to manage the diversity of non-dominated solutions. Consequently, their overall performance reported in Table 1 is not desirable. Overall, LORA-MOO finds a good trade-off between optimization efficiency and optimization results.

Figure 4: Comparison of runtime averaged over 30 runs on benchmark problems $D = 10$ variables and $M = 3, 4, 6, 8,$ and 10 objectives for the comparison algorithms. For each algorithm, its runtimes are normalized by the runtime it costed on 3-objective problems.

## 5 Conclusion

In this paper, we propose an efficient MOO method, LORA-MOO, to solve expensive MOOPs. Different from existing surrogate modeling approaches, our LORA-MOO learns surrogate models from ordinal relations and spherical coordinates. Only one ordinal surrogate is used in the model-based search, which hugely improve the efficiency of optimization. Our empirical studies have demonstrated that our LORA-MOO significantly outperforms other state-of-the-art efficient MOO methods, including SAEAs and MOBO methods.

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

## A  Background of Many-Objective Optimization

We consider minimization problems and many-objective optimization problems (MOOPs) can be formulated as follows:

**Definition 2.** *(Expensive Many-Objective Optimization Problem)*
*Given $M$ expensive objective functions $f_1, \ldots, f_M$ and an evaluation budget $FE_{max}$, obtain the Pareto set for the following many-objective optimization problem:*

$$\operatorname*{argmin}_{\boldsymbol{x} \in X} f(\boldsymbol{x}) = (f_1(\boldsymbol{x}), \ldots, f_M(\boldsymbol{x}))$$

*where $X \subseteq \mathbb{R}^D$ is the decision space of the problem.*

The Pareto set is defined through the following definitions: Pareto set and Pareto front are defined as follows:

**Definition 3.** *Pareto dominance:*
*A solution $\boldsymbol{x}^1$ is said to dominate another solution $\boldsymbol{x}^2$ (denoted by $\boldsymbol{x}^1 \prec \boldsymbol{x}^2$) if and only if:*

$$\forall k \in \{1, 2, \ldots, M\} : f_k(\boldsymbol{x}^1) \leq f_k(\boldsymbol{x}^2) \wedge$$
$$\exists k \in \{1, 2, \ldots, M\} : f_k(\boldsymbol{x}^1) < f_k(\boldsymbol{x}^2)$$

**Definition 4.** *Non-dominated solution:*
*A non-dominated solution $\boldsymbol{x}^\star$ in the decision space $X$ is a solution that cannot be dominated by any other solutions in $X$:*

$$\nexists \boldsymbol{x} \in X : \boldsymbol{x} \prec \boldsymbol{x}^\star$$

**Definition 5.** *Pareto set:*
*Pareto set $S_{ps}$ is the set of all non-dominated solutions in the decision space $X$:*

$$S_{ps} = \{\boldsymbol{x}^\star \in X | \nexists \boldsymbol{x} \in X : \boldsymbol{x} \prec \boldsymbol{x}^\star\}$$

**Definition 6.** *Pareto front:*
*Pareto front $S_{pf}$ is the corresponding unique set of the Pareto set in the objective space:*

$$S_{pf} = \{f(\boldsymbol{x}) | \boldsymbol{x} \in S_{ps}\}$$

## B  Kriging Model

Kriging model, also known as Gaussian process model [23] or design and analysis of computer experiments (DACE) model [32], is a stochastic process model used to approximate an unknown objective function. LORA-MOO uses Kriging models to implement angular surrogates and the radial surrogate, to avoid potential confusion and help the understanding of our algorithm, the working mechanism of the Kriging model is described below.

A common way to approximate an unknown objective function with $n$ observations is linear regression:

$$y(\boldsymbol{x}^i) = \sum_{k=1}^{N} \beta_k f_k(\boldsymbol{x}^i) + \epsilon^i, \tag{6}$$

where $\boldsymbol{x}^i$ is the $i^{th}$ sample point observed from the objective function. $f_k(\boldsymbol{x}^i)$, $\beta_k$ are a linear or nonlinear function of $\boldsymbol{x}^i$ and its coefficient, respectively. $N$ is the number of functions $f(\boldsymbol{x})$. $\epsilon^i$ is an independent error term, which is normally distributed with mean zero and variance $\sigma^2$.

However, a stochastic process model such as Kriging does not assume that the error terms $\epsilon$ are independent. Hence, an error term $\epsilon^i$ is rewritten as $\epsilon(\boldsymbol{x}^i)$. Moreover, these error terms are assumed to be related or correlated to each other. The correlation between two error terms $\epsilon(\boldsymbol{x}^i)$ and $\epsilon(\boldsymbol{x}^j)$ is inversely proportional to the distance between the corresponding points [23]. The correlation function in the Kriging model is defined as:

$$Corr(\epsilon(\boldsymbol{x}^i), \epsilon(\boldsymbol{x}^j)) = exp[-dis(\boldsymbol{x}^i, \boldsymbol{x}^j)], \tag{7}$$

where the distance between two points $\boldsymbol{x}^i$ and $\boldsymbol{x}^j$ are measured using the special weighted distance formula shown below:

$$dis(\boldsymbol{x}^i, \boldsymbol{x}^j) = \sum_{k=1}^{D} \theta_i |x_k^i - x_k^j|^{p_k}, \tag{8}$$

where $D$ is the number of decision variables, $\boldsymbol{\theta} \in \mathbb{R}_{\geq 0}^D$ and $\mathbf{p} \in [1,2]^D$ are parameters of the Kriging model. It can be seen from Eq.(7) that the correlation is ranged within $(0,1]$ and is increasing as the distance between two points decreases. Particularly, in Eq.(8), the parameter $\theta_k$ can be explained as the importance of the decision variable $x_k$, and the parameter $p_k$ can be interpreted as the smoothness of the correlation function in the $k^{th}$ coordinate direction.

Due to the effectiveness of correlation modelling, the regression model in Eq.(6) can be simplified without degrading modelling performance [23]. Clearly, all regression terms are replaced with a constant term, thus the Kriging regression model can be rewritten as follows:

$$y(\boldsymbol{x}^i) = \mu + \epsilon(\boldsymbol{x}^i), \tag{9}$$

where $\mu$ is the mean of this stochastic process, $\epsilon(\boldsymbol{x}^i) \sim \mathcal{N}(0, \sigma^2)$.

## B.1 Training the Kriging model

To train the Kriging model and estimate the parameters $\boldsymbol{\theta}, \mathbf{p}$ in Eq.(8), the following likelihood function is maximised:

$$\frac{1}{(2\pi)^{n/2}(\sigma^2)^{n/2}|\mathbf{R}|^{1/2}} exp[-\frac{(\mathbf{y}-\mathbf{1}\mu)^T\mathbf{R}^{-1}(\mathbf{y}-\mathbf{1}\mu)}{2\sigma^2}], \tag{10}$$

where $|\mathbf{R}|$ is the determinant of the correlation matrix, each element in the matrix is obtained using Eq.(7). $\mathbf{y}$ is the $n$-dimensional vector of dependent variables that observed from the objective function. The mean value $\mu$ and variance $\sigma^2$ in Eq.(9) and Eq.(10) can be estimated by:

$$\hat{\mu} = \frac{\mathbf{1}^T\mathbf{R}^{-1}\mathbf{y}}{\mathbf{1}^T\mathbf{R}^{-1}\mathbf{1}}, \tag{11}$$

$$\hat{\sigma} = \frac{1}{n}(\mathbf{y}-\mathbf{1}\hat{\mu})^T\mathbf{R}^{-1}(\mathbf{y}-\mathbf{1}\hat{\mu}). \tag{12}$$

## B.2 Prediction with the Kriging model

For a new solution $\boldsymbol{x}^*$, the Kriging model predicts the approximation of $\hat{y}(\boldsymbol{x}^*)$ and the uncertainty $\hat{s}^2(\boldsymbol{x}^*)$ as follows:

$$\hat{y}(\boldsymbol{x}^*) = \hat{\mu} + \mathbf{r}'\mathbf{R}^{-1}(\mathbf{y}-\mathbf{1}\hat{\mu}), \tag{13}$$

$$\hat{s}^2(\boldsymbol{x}^*) = \hat{\sigma}^2(1 - \mathbf{r}'\mathbf{R}^{-1}\mathbf{r}), \tag{14}$$

where $\mathbf{r}$ is a $n$-dimensional vector of correlations between $\epsilon(\boldsymbol{x}^*)$ and the error terms at the training data, which can be calculated via Eq.(7).

Further details and a comprehensive description of the Kriging model and Gaussian Process can be found in [39]. In this paper, all regression-based Kriging models have $\boldsymbol{\theta} \in [10^{-5}, 100]^D$, $\mathbf{p} = 2^D$.

# C Additional Description of LORA-MOO

This section describes LORA-MOO with more details.

## C.1 Quantification of Ordinal Relations

In order to learn the ordinal landscape of MOOPs, we need to quantify the ordinal relations between solutions into numerical values. Alg. 2 illustrates the pseudocode of quantifying ordinal relations[3], it describes line 4 in Alg. 1 of the main file. It can be seen that Alg. 2 is mainly working on the quantification of dominance-based ordinal relations. Artificial ordinal relations will not be added unless the ratio of reference points is larger than ratio threshold $rp_{ratio}$ (line 5).

An illustration of artificial clustering-based ordinal relations is given in Fig. 5. By using clustering methods, artificial ordinal relations are generated for training ordinal regression surrogates. Picking one solution from each cluster ensures the diversity of non-dominated solutions in the first ordinal level $L_1$. Meanwhile, the selection within each cluster is based on the projection length on cluster center, which is beneficial to the convergence of non-dominated solutions.

---

[3]Symbol '←' indicates the result of a function, Symbol '=' indicates an assignment operation.

---

**Algorithm 2** Quantify Ordinal Relations for LORA-MOO

---

**Input:**
    $S_A$: Archive of evaluated solutions;
    $rp\_ratio$: Ratio threshold of reference points in $S_A$;
    $n_o$: Minimal number of ordinal levels.

**Procedure:**
  1: $S_{RP} \leftarrow$ Non-dominated solutions in $S_A$ that are non-$\lambda$-dominated to any other solution in $S_A$.
  2: Non-dominated level (The first ordinal level) $L_1 \leftarrow S_{RP}$.
  3: The number of non-dominated ordinal levels $n_{ndl} = 1$.
  4: Ratio of reference points $ratio = \frac{|S_{RP}|}{|S_A|}$.
  5: **if** $ratio > rp_{ratio}$ **then**
  6:     $n_{ndl} = n_{ndl} + 1$.
      /* Add Artificial Ordinal Relations. */
  7:     Divide $S_{RP}$ into $\frac{|S_{RP}|}{2}$ clusters via KNN clustering.
  8:     For $\boldsymbol{x}$ in each cluster, calculate the projection length of $\boldsymbol{x}$ on the corresponding cluster center.
  9:     $L_1 \leftarrow$ Solutions $\boldsymbol{x}$ with the shortest projection on each cluster.
10:     $L_2 \leftarrow$ Remaining $\frac{|S_{RP}|}{2}$ solutions in $S_{RP}$.
11: **end if**
12: Calculate extension coefficient $ec(\boldsymbol{x})$ for all $\boldsymbol{x} \in S_A$.
13: The number of ordinal levels $N_o = \max(n_o, \frac{|S_A|}{|S_{RP}|})$.
14: $L_i \leftarrow$ According to the order of $ec(\boldsymbol{x})$, uniformly divide solutions $\boldsymbol{x} \in (S_A - S_{RP})$ into $N_o$ - $n_{ndl}$ levels.
15: Ordinal relation value $v_i = 1 - \frac{i-1}{N_o-1}$ for $\boldsymbol{x} \in L_i$.

**Output:** An ordinal training set $S_o$ consisting of ordinal relation values $v_i$.

---

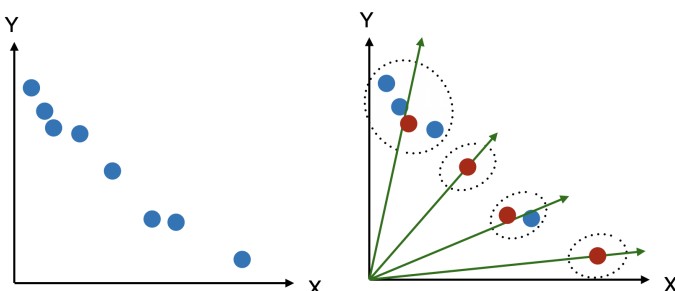

Figure 5: Illustration of artificial clustering-based ordinal relations. **Left**: Non-dominated solutions without artificial ordinal relations. **Right**: Non-dominated solutions with artificial ordinal relations. Red solutions are new non-dominated solutions in $L_1$, remaining blue solutions are moved to next ordinal level $L_2$. Dash circles are clusters, green vectors are cluster centers.

### C.2    Generation of candidate solutions

Algo. 3 gives the pseudocode of generating candidate solutions, it is the implementation of line 6 in Alg. 1 of the main file. In lines 1-9, a population $P_0$ is generated. Since reference points $S_{RP}$ are the optimal solutions in $S_A$ in terms of convergence, a half initial solutions are generated from $S_{RP}$ (lines 2-8). To obtain a diverse subset of $S_{RP}$, LORA-MOO divides $S_{RP}$ into $n_c$ clusters before sampling solutions (line 2). Once population initialization is completed (line 9), a normal PSO is conducted to produce candidate solutions (lines 11-16). Please be noted that, although we are solving expensive MOOPs, only a single ordinal surrogate $h_o$ is used in the reproduction process (line 14). This is a great advantage of LORA-MOO since existing regression-based SAEAs involve all $M$ surrogates in the reproduction process. Hence, LORA-MOO is more efficient than these regression-based SAEAs.

### C.3    Angle-Based Diversity Selection

Alg. 4 gives the pseudocode of selecting the second optimal solution $\boldsymbol{x}_2^*$ from $P$ via our angle-based diversity criterion, it is the implementation of line 11 in Alg. 1 of the main file. This angle-based

---
**Algorithm 3** Generation of candidate solutions in LORA-MOO
---
**Input:**
$S_{RP}$: Reference points used in the ordinal regression;
$h_o$: Ordinal regression surrogate;
$n_c$: The number of clusters to initialize population $P$;
$|P|$: The size of population $P$;
$G_{max}$: The number of generations for reproduction.
**Procedure:**
1: $P_r \leftarrow$ Randomly sample $\frac{|P|}{2}$ solutions from the decision space.
2: Divide $S_{RP}$ into $n_c$ clusters via KNN clustering.
3: $P_c = \emptyset$.
4: **for** $i = 1$ to $n_c$ **do**
5: $\quad P_{ci} \leftarrow$ Randomly sample $\frac{|P|}{2n_c}$ solutions from $i^{th}$ cluster.
6: $\quad P_{ci} \leftarrow$ Mutation ( $P_{ci}$).
7: $\quad P_c = P_c \cup P_{ci}$.
8: **end for**
9: Initial population $P_0 = P_r \cup P_c$.
10: $h_o(P_0) \leftarrow$ Evaluate $P_0$ on ordinal surrogate $h_o$.
11: Global Optimal Population $P_{global} = P_0$.
12: **for** $i = 1$ to $G_{max}$ **do**
13: $\quad P_i \leftarrow$ PSO operation on $P_{i-1}$ and $P_{global}$.
14: $\quad h_o(P_i) \leftarrow$ Evaluate $P_i$ on ordinal surrogate $h_o$.
15: $\quad$ Update $P_{global}$ using $h_o(P_i)$ and $h_o(P_{i-1})$.
16: **end for**
**Output:** A generation of candidate solutions $P = P_{global}$.

---
**Algorithm 4** Angle-Based Diversity Selection in LORA-MOO
---
**Input:**
$S_{RP}$: Reference points used in the ordinal regression;
$P$: Population of candidate solutions;
$h_{a1}, \ldots, h_{a(M-1)}$: $M$-1 angular surrogates;

**Procedure:**
1: $h(ai)(P) \leftarrow$ Evaluate $P$ on angular surrogates $h_{ai}$, i $= 1, \ldots, M - 1$.
2: **for** $j = 2$ to $|P|$ **do**
3: $\quad \boldsymbol{x}_j \leftarrow$ The $j^{th}$ solution in $P$. /* Assume the first solution in $P$ is selected as $\boldsymbol{x}_1^*$ already. */
4: $\quad d_{ang} \leftarrow$ Calculate the angles between $\boldsymbol{x}_j$ and all reference points in $S_{RP}$.
5: $\quad md_{ang} \leftarrow$ The angle between $\boldsymbol{x}_j$ and its nearest reference point.
6: **end for**
7: $\boldsymbol{x}_2^* \leftarrow$ The candidate solution in $P$ with maximal $md_{ang}$.
**Output:** The second candidate solution $\boldsymbol{x}_2^*$.

---

diversity selection does not require extra parameters for generating guidance vectors, it selects the candidate solution that is mostly deviate from solutions in $S_{RP}$. Note that all angular surrogates are only used to evaluate one population $P$ during the whole reproduction and environmental selection procedures. Therefore, although LORA-MOO fits $M$ surrogates in total (one ordinal surrogate and $M$-1 angular surrogates), its runtime cost is less than other SAEAs which fit $M$ surrogates from Cartesian coordinates.

# D  Details of Performance Indicators Used in Our Experiments

In our experiments, we use IGD [4], IGD+ [20], and HV [45] to measure the performance of many objective optimization. Both IGD and IGD+ require a subset of Pareto front as reference points. In our experiments, the number of IGD/IGD+ reference points is set to 5000 for 3-, 4-, and 6-objective optimization problems, as widely used in the literature [40]. Considering the large objective space,

Table 2: The HV reference points for all problems in this work.

| Problem | Reference Points |
|---------|------------------|
| DTLZ | $(1,0,\ldots,1.0)\in\mathbb{R}^M$ |
| WFG | $(1,0,\ldots,1.0)\in\mathbb{R}^M$ |
| NASBench201 | $(1.0, 1.0, 1.0, 1.0, 1.0)$ |

we set the number of IGD/IGD+ reference points to 10000 for 8- and 10-objective optimization problems to achieve a more accurate estimation of optimization performance. The method proposed in [25] is employed to generate well-distributed IGD/IGD+ reference points.

In comparison, the calculation of HV values does not require a subset of Pareto front as reference points. For a set of non-dominated solutions, its HV is the volume in the objective space it dominates from the set to a single reference point. Table 2 lists the reference point used for calculating HV values. All HV values are calculated using the reference point and the normalized solutions. A solution $\boldsymbol{x}$ is normalize by the upper bound and lower bound of Pareto front:

$$\frac{\boldsymbol{x} - lb_{pf}}{ub_{pf} - lb_{pf}}, \tag{15}$$

where $ub_{pf}$, $lb_{pf}$ are the upper bound and lower bound of Pareto front, respectively.

# E  Details of the NASbench201 Problem

NASbench201 [12] are discrete optimization problems that aim to identify the optimal architecture for neural networks. The search space is defined by a cell with 4 nodes inside, forming a directed acyclic graph as illustrated in Fig. 6. The decision variables are 6 edges, each edge is associated

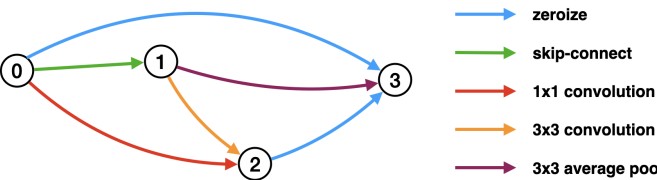

Figure 6: Diagram of a network architecture in NASbench201.

with an operation selected from a predefined operation set {zeroize, skip-connect, 1x1 convolution, 3x3 convolution, 3x3 average pool}. Therefore, a network architecture can be encoded into a 6-dimensional decision vector with 5 discrete numbers. In total, there are $5^6$=15,625 different candidates for neural architecture search.

The optimization objectives in NASbench201 varies in different optimization problems. In this paper, our NASbench201 problem consider 5 objectives, including the accuracy in CI-FAR10 dataset, groundtruth floating point operations (FLOPs), the number of parameters, latency, and energy cost. All these objectives are normalized to [0, 1] in the optimization. The optimization problem can be formulated as

$$F(\boldsymbol{x}) = \{f_{acc}(\boldsymbol{x}), f_{FLOPs}(\boldsymbol{x}), f_{param}(\boldsymbol{x}), f_{latency}(\boldsymbol{x}), f_{energy}(\boldsymbol{x})\}, \tag{16}$$

where decision vector $\boldsymbol{x} \in \{0, 1, 2, 3, 4\}^6$.

# F  Complete Results of Ablation Studies

In this section, we report complete results of our ablation studies that are not displayed in the main paper. We conduct four ablation studies to investigate the effect of the following four parameters on the optimization performance of LORA-MOO.

    1. $n_o$: The minimal number of ordinal levels. A parameter in the modeling of our ordinal-regression-based surrogate $h_o$.

2. $\lambda$: The dominance coefficient. A parameter in the modeling of our ordinal-regression-based surrogate $h_o$.

3. $rp_{ratio}$: The ratio threshold of reference points $S_{RP}$. A parameter to determine whether to introduce artificial ordinal relations via clustering.

4. $n_c$: The number of clusters generated from reference points $S_{RP}$ to initialize PSO population. A parameter in the generation of candidate solutions.

**Setup of Ablation Studies.** Our ablation studies are conducted on 7 DTLZ and 9 WFG benchmark optimization problems. These benchmark problems have different features, such as unimodal, multi-modal, scaled, degenerated, and discontinuous. Therefore, the effect of four parameters can be investigated comprehensively. Considering our paper focuses on many-objective optimization instead of scalable optimization, we are interested in the optimization performance under different numbers of objectives $M$ rather than the performance under different numbers of decision variables $D$. Hence, we set $D = 10$ for all benchmark optimization problems, as suggested in literature [5, 31, 34, 17]. In comparison, we set $M = \{3, 6, 10\}$ to observe the optimization performance with different objectives. Other setups are the same as described in Section 4.1 of the main file.

## F.1   Influence of Minimal Number of Ordinal Levels $n_o$.

This subsection investigates the influence of minimal number of ordinal levels $n_o$ on the optimization performance. We set $n_o = \{10, 8, 6, 4, 3\}$ to generate five LORA-MOO variants. For all variants, in this ablation study, we tentatively set $\lambda = 0.2$, $rp_{ratio} = 2/3$, $n_c = 5$ for a fair comparison. The IGD+ values obtained by five LORA-MOO variants with different $n_o$ are reported in Table 3.

In the last five rows of Table 3, the summary of statistical test results shows that $n_o = 4$ is the optimal parameter setup for LORA-MOO, because it is the only variant that is significantly superior to or equivalent to all other variants. In comparison, the LORA-MOO variant with $n_o = 10, 8, 6, 3$ are significantly inferior to other 4, 1, 1, 2 LORA-MOO variants, respectively.

## F.2   Influence of Dominance Coefficient $\lambda$.

In this subsection, we analyze the influence of $\lambda$-dominance coefficient $\lambda$ on the optimization performance. We set $\lambda = \{0, 0.1, 0.2, 0.3\}$ to generate four LORA-MOO variants. As determined in the previous ablation study, we set $n_o = 4$ for all variants. The remaining two parameters $rp_{ratio}$ and $n_c$ are set to 2/3 and 5, respectively. The IGD+ values obtained by four LORA-MOO variants with different $\lambda$ are reported in Table 4.

The last four rows of Table 4 shows that $\lambda = 0.2$ is the optimal parameter setup for LORA-MOO. The variant of $\lambda = 0.2$ is significantly superior to both the variants of $\lambda = 0$ and $\lambda = 0.1$, and it is equivalent to the variant of $\lambda = 0.3$. We note that the variant of $\lambda = 0.3$ is also significantly superior to both the variants of $\lambda = 0$ and $\lambda = 0.1$. However, this variant wins/ties/losses 30/105/9 statistical tests in total, while the variant of $\lambda = 0.2$ wins/ties/losses 32/109/3 statistical tests in total. Therefore, setting $\lambda = 0.2$ is preferable to setting $\lambda = 0.3$.

Note that all other LORA-MOO variants outperform the variant of $\lambda = 0$, this implies that excluding some samples from the set of non-dominated solutions is beneficial to the performance of ordinal regression. The effectiveness of using our $\lambda$-dominance approach in LORA-MOO is demonstrated.

## F.3   Influence of Ratio Threshold $rp_{ratio}$.

In this subsection, we investigate the influence of ratio threshold $rp_{ratio}$ on the optimization performance. $rp_{ratio}$ is the threshold to determine when to add artificial ordinal relations for the training of ordinal surrogate $h_o$. We set $rp_{ratio} = \{1, 2/3, 1/2, 1/3\}$ to generate four LORA-MOO variants. For all variants, we set $n_o$, $\lambda$ to 4, 0.2, respectively, which are consistent with our conclusions in previous ablation studies. Parameter $n_c$ is tentatively set to 5. The IGD+ values obtained by four LORA-MOO variants with different $rp_{ratio}$ are reported in Table 5. It should be noted that, when the number of objectives $M = 3$, the results of $rp_{ratio} = 1$ are the same as the results of $rp_{ratio} = 2/3$, because the ratio of reference points in archive $S_A$ is always lower than 2/3. Consequently, when $M = 3$, setting ratio threshold $rp_{ratio}$ to either 1 or 2/3 makes no difference to the optimization process of LORA-MOO. Similarly, the results of $rp_{ratio} = 1/3$ on some problems are the same as the results

Table 3: Statistical results of the IGD+ value obtained by LORA-MOO with different $n_o$ on 48 benchmark optimization problems over 15 runs. The last five rows count the total results of Wilcoxon rank sum tests (significance level is 0.05). '+', '$\approx$', and '$-$' denote the corresponding LORA-MOO variant is statistically significantly superior to, almost equivalent to, and inferior to the compared variants in Wilcoxon tests, respectively.

| Problems | M | $n_o$=10 | $n_o$=8 | $n_o$=6 | $n_o$=4 | $n_o$=3 |
|---|---|---|---|---|---|---|
| DTLZ1 | 3 | 4.63e+1(1.60e+1) | 4.64e+1(1.23e+1) | 5.61e+1(2.04e+1) | 4.84e+1(1.34e+1) | 4.58e+1(1.85e+1) |
| | 6 | 1.35e+1(7.10e+0) | 1.77e+1(5.08e+0) | 1.87e+1(6.85e+0) | 1.64e+1(3.24e+0) | 1.50e+1(7.84e+0) |
| | 10 | 1.56e-1(3.58e-2) | 1.60e-1(3.60e-2) | 1.63e-1(6.95e-2) | 1.60e-1(2.67e-2) | 1.63e-1(3.51e-2) |
| DTLZ2 | 3 | 4.50e-2(3.90e-3) | 4.54e-2(4.16e-3) | 4.38e-2(2.61e-3) | 4.45e-2(4.72e-3) | 4.39e-2(3.88e-3) |
| | 6 | 2.67e-1(1.47e-2) | 2.73e-1(1.93e-2) | 2.64e-1(1.67e-2) | 2.57e-1(1.91e-2) | 2.51e-1(2.20e-2) |
| | 10 | 3.04e-1(1.55e-2) | 2.97e-1(1.63e-2) | 2.94e-1(1.24e-2) | 3.00e-1(1.31e-2) | 3.11e-1(1.78e-2) |
| DTLZ3 | 3 | 1.50e+2(4.72e+1) | 1.60e+2(4.92e+1) | 1.55e+2(5.03e+1) | 1.48e+2(4.92e+1) | 1.45e+2(4.10e+1) |
| | 6 | 5.43e+1(1.85e+1) | 5.65e+1(1.99e+1) | 6.92e+1(2.39e+1) | 6.68e+1(1.64e+1) | 6.24e+1(2.34e+1) |
| | 10 | 4.51e-1(4.40e-2) | 4.68e-1(6.10e-2) | 4.35e-1(3.71e-2) | 4.72e-1(5.45e-2) | 4.85e-1(7.87e-2) |
| DTLZ4 | 3 | 1.03e-1(1.28e-1) | 8.77e-2(1.30e-1) | 9.16e-2(1.25e-1) | 1.05e-1(1.27e-1) | 1.15e-1(1.33e-1) |
| | 6 | 1.74e-1(3.63e-2) | 1.60e-1(3.35e-2) | 1.84e-1(3.79e-2) | 1.75e-1(3.57e-2) | 1.68e-1(2.11e-2) |
| | 10 | 2.29e-1(1.05e-2) | 2.29e-1(9.43e-3) | 2.36e-1(1.27e-2) | 2.38e-1(1.35e-2) | 2.42e-1(1.71e-2) |
| DTLZ5 | 3 | 8.65e-3(1.39e-3) | 8.76e-3(1.53e-3) | 9.03e-3(1.67e-3) | 9.26e-3(1.22e-3) | 9.26e-3(2.23e-3) |
| | 6 | 3.43e-2(7.07e-3) | 3.28e-2(7.74e-3) | 3.24e-2(7.73e-3) | 3.25e-2(8.25e-3) | 3.33e-2(9.38e-3) |
| | 10 | 4.06e-3(6.52e-4) | 3.99e-3(4.47e-4) | 3.94e-3(4.04e-4) | 3.97e-3(9.34e-4) | 4.02e-3(1.10e-3) |
| DTLZ6 | 3 | 5.09e-2(5.72e-2) | 1.05e-1(2.57e-1) | 2.45e-2(8.80e-3) | 4.67e-2(4.92e-2) | 3.12e-2(1.58e-2) |
| | 6 | 9.45e-1(1.13e+0) | 5.16e-1(6.72e-1) | 5.42e-1(8.28e-1) | 7.52e-1(9.50e-1) | 1.34e+0(1.04e+0) |
| | 10 | 4.48e-2(3.90e-2) | 2.50e-2(7.37e-3) | 5.14e-2(4.26e-2) | 4.18e-2(4.66e-2) | 4.72e-2(4.57e-2) |
| DTLZ7 | 3 | 1.19e-1(1.00e-1) | 9.47e-2(1.15e-1) | 1.16e-1(7.80e-2) | 1.61e-1(2.77e-1) | 1.46e-1(1.27e-1) |
| | 6 | 1.90e+0(9.89e-1) | 1.72e+0(6.52e-1) | 1.77e+0(7.63e-1) | 1.25e+0(4.72e-1) | 1.54e+0(8.80e-1) |
| | 10 | 1.19e+0(9.00e-2) | 1.18e+0(9.13e-2) | 1.17e+0(8.41e-2) | 1.17e+0(8.97e-2) | 1.22e+0(1.13e-1) |
| WFG1 | 3 | 1.65e+0(5.78e-2) | 1.65e+0(3.73e-2) | 1.64e+0(3.86e-2) | 1.67e+0(4.67e-2) | 1.65e+0(5.96e-2) |
| | 6 | 2.24e+0(5.47e-2) | 2.20e+0(6.93e-2) | 2.23e+0(4.37e-2) | 2.22e+0(6.80e-2) | 2.21e+0(5.52e-2) |
| | 10 | 2.62e+0(8.72e-2) | 2.58e+0(7.39e-2) | 2.59e+0(7.81e-2) | 2.62e+0(8.93e-2) | 2.58e+0(1.16e-1) |
| WFG2 | 3 | 2.39e-1(3.16e-2) | 2.49e-1(4.94e-2) | 2.68e-1(4.81e-2) | 2.52e-1(4.94e-2) | 2.66e-1(4.58e-2) |
| | 6 | 5.91e-1(1.79e-1) | 5.85e-1(9.10e-2) | 5.61e-1(1.29e-1) | 5.43e-1(1.51e-1) | 5.67e-1(1.07e-1) |
| | 10 | 1.50e+0(3.53e-1) | 1.41e+0(2.62e-1) | 1.42e+0(3.21e-1) | 1.47e+0(4.49e-1) | 1.39e+0(2.82e-1) |
| WFG3 | 3 | 2.42e-1(4.10e-2) | 2.66e-1(3.75e-2) | 2.57e-1(3.28e-2) | 2.41e-1(3.21e-2) | 2.56e-1(5.04e-2) |
| | 6 | 6.19e-1(8.08e-2) | 6.28e-1(6.58e-2) | 6.15e-1(9.32e-2) | 5.92e-1(7.43e-2) | 6.19e-1(1.22e-1) |
| | 10 | 6.24e-1(9.78e-2) | 6.07e-1(8.67e-2) | 6.18e-1(8.74e-2) | 6.60e-1(8.00e-2) | 6.61e-1(8.80e-2) |
| WFG4 | 3 | 2.62e-1(5.18e-2) | 2.52e-1(1.99e-2) | 2.51e-1(1.27e-2) | 2.48e-1(1.04e-2) | 2.38e-1(8.69e-3) |
| | 6 | 1.41e+0(2.17e-1) | 1.34e+0(1.96e-1) | 1.27e+0(2.31e-1) | 1.30e+0(2.41e-1) | 1.58e+0(4.08e-1) |
| | 10 | 4.12e+0(5.64e-1) | 3.63e+0(6.43e-1) | 3.55e+0(5.77e-1) | 3.99e+0(7.21e-1) | 4.08e+0(7.57e-1) |
| WFG5 | 3 | 2.93e-1(4.46e-2) | 2.89e-1(5.58e-2) | 3.01e-1(9.11e-2) | 3.10e-1(5.46e-2) | 3.19e-1(9.97e-2) |
| | 6 | 1.69e+0(8.33e-2) | 1.72e+0(8.16e-2) | 1.66e+0(9.57e-2) | 1.69e+0(1.53e-1) | 1.83e+0(1.34e-1) |
| | 10 | 4.76e+0(2.87e-1) | 4.57e+0(3.19e-1) | 4.10e+0(3.07e-1) | 3.71e+0(3.87e-1) | 3.71e+0(4.39e-1) |
| WFG6 | 3 | 4.66e-1(4.13e-2) | 4.91e-1(4.44e-2) | 4.51e-1(4.36e-2) | 4.76e-1(6.61e-2) | 4.58e-1(8.29e-2) |
| | 6 | 1.70e+0(1.48e-1) | 1.65e+0(9.89e-2) | 1.61e+0(1.10e-1) | 1.67e+0(1.35e-1) | 1.81e+0(2.71e-1) |
| | 10 | 3.88e+0(6.68e-1) | 3.60e+0(3.51e-1) | 3.64e+0(2.96e-1) | 3.45e+0(4.44e-1) | 3.72e+0(5.21e-1) |
| WFG7 | 3 | 3.12e-1(2.16e-2) | 3.02e-1(2.17e-2) | 3.00e-1(2.68e-2) | 3.02e-1(2.75e-2) | 2.99e-1(2.96e-2) |
| | 6 | 1.78e+0(1.05e-1) | 1.69e+0(1.27e-1) | 1.73e+0(1.38e-1) | 1.67e+0(1.85e-1) | 1.74e+0(2.32e-1) |
| | 10 | 5.15e+0(3.94e-1) | 5.11e+0(2.97e-1) | 4.89e+0(2.62e-1) | 4.97e+0(3.07e-1) | 4.94e+0(4.00e-1) |
| WFG8 | 3 | 5.84e-1(5.34e-2) | 6.09e-1(5.54e-2) | 6.07e-1(4.89e-2) | 5.68e-1(4.78e-2) | 5.70e-1(4.15e-2) |
| | 6 | 2.19e+0(1.08e-1) | 2.11e+0(9.97e-2) | 2.15e+0(1.22e-1) | 2.25e+0(1.12e-1) | 2.37e+0(1.76e-1) |
| | 10 | 5.22e+0(4.43e-1) | 5.31e+0(3.08e-1) | 4.99e+0(3.75e-1) | 5.16e+0(5.37e-1) | 5.37e+0(4.82e-1) |
| WFG9 | 3 | 3.79e-1(7.28e-2) | 3.85e-1(1.20e-1) | 3.73e-1(8.90e-2) | 4.12e-1(1.17e-1) | 4.17e-1(1.11e-1) |
| | 6 | 1.87e+0(1.95e-1) | 1.73e+0(2.02e-1) | 1.78e+0(2.45e-1) | 1.77e+0(2.57e-1) | 1.76e+0(1.35e-1) |
| | 10 | 5.03e+0(2.28e-1) | 4.63e+0(4.11e-1) | 4.44e+0(4.68e-1) | 3.96e+0(3.83e-1) | 3.73e+0(2.50e-1) |
| $+/\approx/-$ | $n_o$=10 | -/-/- | 1/41/6 | 2/40/6 | 0/44/4 | 3/41/4 |
| $+/\approx/-$ | $n_o$=8 | 6/41/1 | -/-/- | 2/43/3 | 3/42/3 | 4/40/4 |
| $+/\approx/-$ | $n_o$=6 | 6/40/2 | 3/43/2 | -/-/- | 3/41/4 | 7/38/3 |
| $+/\approx/-$ | $n_o$=4 | 4/44/0 | 3/42/3 | 4/41/3 | -/-/- | 2/45/1 |
| $+/\approx/-$ | $n_o$=3 | 4/41/3 | 4/40/4 | 3/38/7 | 1/45/2 | -/-/- |

obtained by setting $rp_{ratio}$ to 1/2, because on these problems, the ratio of reference points in $S_A$ is always higher than 1/2.

As shown in Table 5, the variant of $rp_{ratio}$ = 1/2 outperforms other variants and achieves the optimal behavior. Therefore, we set $rp_{ratio}$ = 1/2 for LORA-MOO. In comparison, the variants of $rp_{ratio}$ = 2/3 and $rp_{ratio}$ = 1/3 have competitive performance, both of them are inferior to the variant of $rp_{ratio}$ = 1/2 but significantly superior to the variant of $rp_{ratio}$ = 1.

Setting $rp_{ratio}$ = 1 indicates this LORA-MOO variant will never introduce artificial ordinal relations for the learning of the ordinal surrogate. The ordinal surrogate in this variant is trained completely on

Table 4: Statistical results of the IGD+ value obtained by LORA-MOO with different $\lambda$ on 48 benchmark optimization problems over 15 runs. The last four rows count the total results of Wilcoxon rank sum tests (significance level is 0.05). '+', '≈', and '−' denote the corresponding LORA-MOO variant is statistically significantly superior to, almost equivalent to, and inferior to the compared variants in Wilcoxon tests, respectively.

| Problems | M | $\lambda = 0$ | $\lambda = 0.1$ | $\lambda = 0.2$ | $\lambda = 0.3$ |
|---|---|---|---|---|---|
| DTLZ1 | 3 | 7.51e+1(1.74e+1) | 6.88e+1(1.28e+1) | 4.84e+1(1.34e+1) | 4.96e+1(1.56e+1) |
| | 6 | 2.74e+1(5.30e+0) | 1.73e+1(3.80e+0) | 1.64e+1(3.24e+0) | 1.41e+1(7.02e+0) |
| | 10 | 1.62e-1(5.15e-2) | 1.43e-1(2.33e-2) | 1.60e-1(2.67e-2) | 1.53e-1(2.28e-2) |
| DTLZ2 | 3 | 4.95e-2(3.32e-3) | 4.89e-2(5.80e-3) | 4.45e-2(4.72e-3) | 4.81e-2(4.10e-3) |
| | 6 | 2.51e-1(2.91e-2) | 2.56e-1(2.48e-2) | 2.57e-1(1.91e-2) | 2.67e-1(1.34e-2) |
| | 10 | 2.97e-1(1.72e-2) | 2.94e-1(1.54e-2) | 3.00e-1(1.31e-2) | 2.92e-1(1.35e-2) |
| DTLZ3 | 3 | 1.91e+2(6.02e+1) | 1.80e+2(2.31e+1) | 1.48e+2(4.92e+1) | 1.57e+2(4.54e+1) |
| | 6 | 9.01e+1(3.13e+1) | 8.06e+1(2.18e+1) | 6.68e+1(1.64e+1) | 6.05e+1(2.03e+1) |
| | 10 | 5.74e-1(2.57e-1) | 4.60e-1(5.69e-2) | 4.72e-1(5.45e-2) | 4.48e-1(4.14e-2) |
| DTLZ4 | 3 | 9.37e-2(1.30e-1) | 1.16e-1(1.35e-1) | 1.05e-1(1.27e-1) | 1.02e-1(1.28e-1) |
| | 6 | 1.72e-1(2.91e-2) | 1.63e-1(3.51e-2) | 1.75e-1(1.96e-2) | 1.61e-1(1.96e-2) |
| | 10 | 2.36e-1(1.29e-2) | 2.37e-1(1.77e-2) | 2.38e-1(1.35e-2) | 2.28e-1(1.05e-2) |
| DTLZ5 | 3 | 1.40e-2(2.50e-3) | 1.13e-2(3.34e-3) | 9.26e-3(1.22e-3) | 7.96e-3(1.58e-3) |
| | 6 | 5.00e-2(9.20e-3) | 4.52e-2(1.60e-2) | 3.25e-2(8.25e-3) | 3.48e-2(5.12e-3) |
| | 10 | 5.16e-3(9.20e-4) | 4.44e-3(1.43e-3) | 3.97e-3(9.34e-4) | 4.10e-3(3.97e-4) |
| DTLZ6 | 3 | 1.54e-1(1.65e-1) | 4.14e-2(1.61e-2) | 4.67e-2(4.92e-2) | 4.13e-2(2.30e-2) |
| | 6 | 1.72e+0(7.66e-1) | 1.52e+0(1.08e+0) | 7.52e-1(9.50e-1) | 2.45e-1(4.79e-1) |
| | 10 | 9.60e-2(7.76e-2) | 6.08e-2(5.26e-2) | 4.18e-2(4.66e-2) | 2.99e-2(9.13e-3) |
| DTLZ7 | 3 | 6.57e-2(1.85e-2) | 1.25e-1(1.06e-1) | 1.61e-1(2.77e-1) | 1.05e-1(1.80e-1) |
| | 6 | 2.74e+0(1.22e+0) | 1.53e+0(8.21e-1) | 1.25e+0(4.72e-1) | 1.66e+0(1.06e+0) |
| | 10 | 1.19e+0(9.70e-2) | 1.18e+0(8.58e-2) | 1.17e+0(8.97e-2) | 1.27e+0(1.61e-1) |
| WFG1 | 3 | 1.74e+0(4.92e-2) | 1.67e+0(4.82e-2) | 1.67e+0(4.67e-2) | 1.64e+0(3.52e-2) |
| | 6 | 2.30e+0(3.54e-2) | 2.22e+0(8.09e-2) | 2.22e+0(6.80e-2) | 2.23e+0(7.54e-2) |
| | 10 | 2.71e+0(6.98e-2) | 2.63e+0(7.80e-2) | 2.62e+0(8.93e-2) | 2.63e+0(7.71e-2) |
| WFG2 | 3 | 2.94e-1(5.47e-2) | 2.69e-1(5.46e-1) | 2.52e-1(4.94e-2) | 2.55e-1(3.46e-2) |
| | 6 | 6.84e-1(1.47e-1) | 5.38e-1(1.05e-1) | 5.43e-1(1.51e-1) | 6.65e-1(2.55e-1) |
| | 10 | 1.67e+0(5.02e-1) | 1.27e+0(2.80e-1) | 1.47e+0(4.49e-1) | 1.37e+0(3.46e-1) |
| WFG3 | 3 | 4.08e-1(4.84e-2) | 3.25e-1(3.53e-2) | 2.41e-1(3.21e-2) | 2.70e-1(5.19e-2) |
| | 6 | 8.23e-1(6.96e-2) | 7.51e-1(9.15e-2) | 5.92e-1(7.43e-2) | 4.94e-1(6.55e-2) |
| | 10 | 7.58e-1(7.71e-2) | 7.71e-1(1.08e-1) | 6.60e-1(8.00e-2) | 6.35e-1(1.04e-1) |
| WFG4 | 3 | 2.55e-1(1.63e-2) | 2.56e-1(1.48e-2) | 2.48e-1(1.04e-2) | 2.57e-1(1.44e-2) |
| | 6 | 1.28e+0(2.24e-1) | 1.31e+0(2.39e-1) | 1.30e+0(2.41e-1) | 1.37e+0(2.50e-1) |
| | 10 | 3.85e+0(5.45e-1) | 3.84e+0(5.48e-1) | 3.99e+0(7.21e-1) | 3.79e+0(4.91e-1) |
| WFG5 | 3 | 3.84e-1(1.18e-1) | 2.89e-1(6.47e-2) | 3.10e-1(5.46e-2) | 3.11e-1(6.94e-2) |
| | 6 | 1.77e+0(1.36e-1) | 1.72e+0(1.43e-1) | 1.69e+0(1.53e-1) | 1.72e+0(1.20e-1) |
| | 10 | 3.70e+0(4.80e-1) | 3.58e+0(2.79e-1) | 3.71e+0(3.87e-1) | 4.38e+0(2.67e-1) |
| WFG6 | 3 | 4.78e-1(7.23e-2) | 4.63e-1(5.50e-2) | 4.76e-1(6.61e-2) | 4.74e-1(4.87e-2) |
| | 6 | 1.62e+0(1.67e-1) | 1.59e+0(1.21e-1) | 1.67e+0(1.35e-1) | 1.60e+0(1.52e-1) |
| | 10 | 3.48e+0(2.80e-1) | 3.43e+0(3.18e-1) | 3.45e+0(4.44e-1) | 3.70e+0(3.85e-1) |
| WFG7 | 3 | 3.16e-1(2.20e-2) | 3.13e-1(3.79e-2) | 3.02e-1(2.75e-2) | 3.17e-1(4.42e-2) |
| | 6 | 1.62e+0(1.57e-1) | 1.68e+0(1.80e-1) | 1.67e+0(1.85e-1) | 1.69e+0(1.88e-1) |
| | 10 | 4.88e+0(4.14e-1) | 4.99e+0(3.94e-1) | 4.97e+0(3.07e-1) | 4.98e+0(2.87e-1) |
| WFG8 | 3 | 5.96e-1(4.58e-2) | 6.09e-1(3.63e-2) | 5.68e-1(4.78e-2) | 5.96e-1(3.58e-2) |
| | 6 | 2.21e+0(1.49e-1) | 2.20e+0(1.18e-1) | 2.25e+0(1.12e-1) | 2.20e+0(7.76e-2) |
| | 10 | 5.07e+0(4.48e-1) | 4.96e+0(4.84e-1) | 5.16e+0(5.37e-1) | 5.09e+0(3.92e-1) |
| WFG9 | 3 | 3.72e-1(3.91e-2) | 3.82e-1(9.02e-2) | 4.12e-1(1.17e-1) | 3.80e-1(1.00e-1) |
| | 6 | 1.76e+0(2.07e-1) | 1.67e+0(1.86e-1) | 1.77e+0(2.57e-1) | 1.81e+0(1.69e-1) |
| | 10 | 3.87e+0(3.66e-1) | 4.13e+0(3.55e-1) | 3.96e+0(3.83e-1) | 4.76e+0(2.31e-1) |
| $+/ \approx /-$ | $\lambda=0$ | -/-/- | 0/35/13 | 0/29/19 | 3/27/18 |
| $+/ \approx /-$ | $\lambda=0.1$ | 13/35/0 | -/-/- | 0/38/10 | 3/36/9 |
| $+/ \approx /-$ | $\lambda=0.2$ | 19/29/0 | 10/38/0 | -/-/- | 3/42/3 |
| $+/ \approx /-$ | $\lambda=0.3$ | 18/27/3 | 9/36/3 | 3/42/3 | -/-/- |

the basis of dominance ordinal relations. When the number of objectives $M$ is large, a majority of evaluated solutions in archive $S_A$ are non-dominated, leading to a large ratio of reference points $S_{RP}$ in $S_A$. As a result, there would be a significant imbalance between the number of evaluated solutions in each ordinal level, which causes a poor performance on ordinal surrogate and LORA-MOO. In particular, on most 10-objective WFG problems, the variant of $rp_{ratio} = 1$ performs worse than all other variants. This observation shows the detrimental effect of imbalance solutions in ordinal levels on the optimization performance, which also demonstrates the effectiveness of using artificial ordinal relations in LORA-MOO to address many-objective optimization problems.

Table 5: Statistical results of the IGD+ value obtained by LORA-MOO with different $rp_{ratio}$ on 48 benchmark optimization problems over 15 runs. The last four rows count the total results of Wilcoxon rank sum tests (significance level is 0.05). '+', '≈', and '−' denote the corresponding LORA-MOO variant is statistically significantly superior to, almost equivalent to, and inferior to the compared variants in Wilcoxon tests, respectively.

| Problems | M | $rp_{ratio}$=1 | $rp_{ratio}$=2/3 | $rp_{ratio}$=1/2 | $rp_{ratio}$=1/3 |
|---|---|---|---|---|---|
| DTLZ1 | 3 | 4.84e+1(1.34e+1) | 4.84e+1(1.34e+1) | 4.75e+1(1.54e+1) | 4.75e+1(1.54e+1) |
|  | 6 | 1.83e+1(1.06e+1) | 1.64e+1(3.24e+0) | 1.35e+1(6.23e+0) | 1.35e+1(6.23e+0) |
|  | 10 | 1.63e-1(2.74e-2) | 1.60e-1(2.67e-2) | 1.58e-1(2.81e-2) | 1.58e-1(2.81e-2) |
| DTLZ2 | 3 | 4.45e-2(4.72e-3) | 4.45e-2(4.72e-3) | 4.37e-2(3.41e-3) | 3.60e-2(3.69e-3) |
|  | 6 | 2.57e-1(1.93e-2) | 2.57e-1(1.91e-2) | 1.80e-1(1.17e-2) | 1.80e-1(7.34e-3) |
|  | 10 | 3.74e-1(8.09e-3) | 3.00e-1(1.31e-2) | 2.87e-1(1.71e-2) | 2.87e-1(1.71e-2) |
| DTLZ3 | 3 | 1.48e+2(4.92e+1) | 1.48e+2(4.92e+1) | 1.54e+2(4.89e+1) | 1.54e+2(4.89e+1) |
|  | 6 | 6.52e+1(2.87e+1) | 6.68e+1(1.64e+1) | 6.01e+1(2.61e+1) | 6.01e+1(2.61e+1) |
|  | 10 | 4.23e-1(5.63e-2) | 4.72e-1(5.45e-2) | 4.84e-1(5.71e-2) | 4.84e-1(5.71e-2) |
| DTLZ4 | 3 | 1.05e-1(1.27e-1) | 1.05e-1(1.27e-1) | 1.06e-1(1.32e-1) | 1.06e-1(1.32e-1) |
|  | 6 | 1.70e-1(3.56e-2) | 1.75e-1(3.57e-2) | 1.79e-1(4.06e-2) | 1.79e-1(4.06e-2) |
|  | 10 | 2.33e-1(1.26e-2) | 2.38e-1(1.35e-2) | 2.38e-1(1.56e-2) | 2.49e-1(1.46e-2) |
| DTLZ5 | 3 | 9.26e-3(1.22e-3) | 9.26e-3(1.22e-3) | 8.98e-3(1.67e-3) | 8.71e-3(1.89e-3) |
|  | 6 | 3.40e-2(9.35e-3) | 3.25e-2(8.25e-3) | 3.31e-2(7.84e-3) | 2.81e-2(1.15e-2) |
|  | 10 | 3.83e-3(6.08e-4) | 3.97e-3(9.34e-4) | 4.85e-3(1.78e-3) | 4.92e-3(1.54e-3) |
| DTLZ6 | 3 | 4.67e-2(4.92e-2) | 4.67e-2(4.92e-2) | 6.38e-2(7.62e-2) | 2.56e-2(6.58e-3) |
|  | 6 | 4.70e-1(7.64e-1) | 7.52e-1(9.50e-1) | 7.28e-1(1.00e+0) | 1.25e+0(1.13e+0) |
|  | 10 | 3.38e-2(1.18e-2) | 4.18e-2(4.66e-2) | 3.92e-2(3.62e-2) | 3.27e-2(2.08e-2) |
| DTLZ7 | 3 | 1.61e-1(2.77e-1) | 1.61e-1(2.77e-1) | 1.36e-1(1.32e-1) | 7.58e-2(2.50e-2) |
|  | 6 | 1.41e+0(9.24e-1) | 1.25e+0(4.72e-1) | 1.21e+0(7.32e-1) | 1.28e+0(6.69e-1) |
|  | 10 | 1.17e+0(8.28e-2) | 1.17e+0(8.97e-2) | 1.23e+0(1.33e-1) | 1.23e+0(1.33e-1) |
| WFG1 | 3 | 1.67e+0(4.67e-2) | 1.67e+0(4.67e-2) | 1.67e+0(4.86e-2) | 1.67e+0(4.86e-2) |
|  | 6 | 2.20e+0(6.03e-2) | 2.22e+0(6.80e-2) | 2.21e+0(5.69e-2) | 2.21e+0(5.69e-2) |
|  | 10 | 2.61e+0(1.15e-1) | 2.62e+0(8.93e-2) | 2.55e+0(1.15e-1) | 2.55e+0(1.15e-1) |
| WFG2 | 3 | 2.52e-1(4.94e-2) | 2.52e-1(4.94e-2) | 2.48e-1(5.57e-2) | 2.48e-1(5.57e-2) |
|  | 6 | 5.73e-1(1.75e-1) | 5.43e-1(1.51e-1) | 5.35e-1(9.94e-2) | 5.35e-1(9.94e-2) |
|  | 10 | 1.37e+0(3.08e-1) | 1.47e+0(4.49e-1) | 1.36e+0(3.13e-1) | 1.25e+0(3.81e-1) |
| WFG3 | 3 | 2.41e-1(3.21e-2) | 2.41e-1(3.21e-2) | 2.51e-1(3.82e-2) | 2.51e-1(3.26e-2) |
|  | 6 | 5.82e-1(4.97e-2) | 5.92e-1(7.43e-2) | 5.83e-1(8.20e-2) | 6.05e-1(9.65e-2) |
|  | 10 | 6.09e-1(4.65e-2) | 6.60e-1(8.00e-2) | 6.93e-1(1.22e-1) | 6.63e-1(1.05e-1) |
| WFG4 | 3 | 2.48e-1(1.04e-2) | 2.48e-1(1.04e-2) | 2.49e-1(2.61e-2) | 2.96e-1(9.20e-2) |
|  | 6 | 2.06e+0(4.21e-1) | 1.30e+0(2.41e-1) | 1.35e+0(3.15e-1) | 1.35e+0(3.15e-1) |
|  | 10 | 5.51e+0(6.14e-1) | 3.99e+0(7.21e-1) | 3.86e+0(6.03e-1) | 3.86e+0(6.03e-1) |
| WFG5 | 3 | 3.10e-1(5.46e-2) | 3.10e-1(5.46e-2) | 3.06e-1(1.05e-1) | 4.28e-1(1.46e-1) |
|  | 6 | 1.93e+0(1.20e-1) | 1.69e+0(1.53e-1) | 1.72e+0(1.26e-1) | 1.72e+0(1.26e-1) |
|  | 10 | 5.50e+0(3.80e-1) | 3.71e+0(3.87e-1) | 3.63e+0(4.80e-1) | 3.63e+0(4.80e-1) |
| WFG6 | 3 | 4.76e-1(6.61e-2) | 4.76e-1(6.61e-2) | 4.87e-1(1.00e-1) | 6.26e-1(1.19e-1) |
|  | 6 | 2.21e+0(2.26e-1) | 1.67e+0(1.35e-1) | 1.62e+0(1.85e-1) | 1.62e+0(1.85e-1) |
|  | 10 | 5.43e+0(4.78e-1) | 3.45e+0(4.44e-1) | 3.19e+0(2.14e-1) | 3.19e+0(2.14e-1) |
| WFG7 | 3 | 3.02e-1(2.75e-2) | 3.02e-1(2.75e-2) | 2.95e-1(2.76e-2) | 2.98e-1(3.12e-2) |
|  | 6 | 2.10e+0(2.12e-1) | 1.67e+0(1.85e-1) | 1.58e+0(1.47e-1) | 1.58e+0(1.47e-1) |
|  | 10 | 5.85e+0(5.16e-1) | 4.97e+0(3.07e-1) | 4.76e+0(4.89e-1) | 4.76e+0(4.89e-1) |
| WFG8 | 3 | 5.68e-1(4.78e-2) | 5.68e-1(4.78e-2) | 5.71e-1(4.02e-2) | 5.83e-1(4.65e-2) |
|  | 6 | 2.61e+0(2.09e-1) | 2.25e+0(1.12e-1) | 2.21e+0(1.21e-1) | 2.21e+0(1.21e-1) |
|  | 10 | 6.41e+0(4.20e-1) | 5.16e+0(5.37e-1) | 5.06e+0(5.80e-1) | 5.06e+0(5.80e-1) |
| WFG9 | 3 | 4.12e-1(1.17e-1) | 4.12e-1(1.17e-1) | 3.81e-1(1.02e-1) | 3.66e-1(8.95e-2) |
|  | 6 | 1.86e+0(2.00e-1) | 1.77e+0(2.57e-1) | 1.48e+0(2.27e-1) | 1.45e+0(1.77e-1) |
|  | 10 | 5.57e+0(2.73e-1) | 3.96e+0(3.83e-1) | 4.02e+0(4.62e-1) | 4.02e+0(4.62e-1) |
| $+/\approx/-$ | $rp_{ratio}$=1 | -/-/- | 2/34/12 | 2/32/14 | 5/28/15 |
| $+/\approx/-$ | $rp_{ratio}$=2/3 | 12/34/2 | -/-/- | 0/46/2 | 3/42/3 |
| $+/\approx/-$ | $rp_{ratio}$=1/2 | 14/32/2 | 2/46/0 | -/-/- | 2/45/1 |
| $+/\approx/-$ | $rp_{ratio}$=1/3 | 15/28/5 | 3/42/3 | 1/45/2 | -/-/- |

**F.4 Influence of Clustering Number for Reproduction $n_c$.**

This subsection analyzes the influence of clustering number $n_c$ on the optimization performance. $n_c$ is used in the reproduction process to initialize the PSO population. We set $n_c = \{1, 3, 5, 7, 10\}$ to generate five LORA-MOO variants. According to the conclusions of previous ablation studies, in this ablation study, we set $n_o = 4$, $\lambda = 0.2$, $rp_{ratio} = 1/2$ for all variants. The IGD+ values obtained by five LORA-MOO variants with different $n_c$ are reported in Table 6.

It can be observed that both the variants of $n_c = 5$ and $n_c = 7$ outperform three other variants and are inferior to one variant, showing the optimal performance over other variants in this ablation study. In comparison, the variants of $n_c = 3$ and $n_c = 10$ are significantly superior to two variants but are

Table 6: Statistical results of the IGD+ value obtained by LORA-MOO with different $n_c$ on 48 benchmark optimization problems over 15 runs. The last five rows count the total results of Wilcoxon rank sum tests (significance level is 0.05). '+', '≈', and '−' denote the corresponding LORA-MOO variant is statistically significantly superior to, almost equivalent to, and inferior to the compared variants in Wilcoxon tests, respectively.

| Problems | M | $n_c$=1 | $n_c$=3 | $n_c$=5 | $n_c$=7 | $n_c$=10 |
|---|---|---|---|---|---|---|
| DTLZ1 | 3 | 6.45e+1(1.31e+1) | 5.77e+1(2.13e+1) | 4.75e+1(1.54e+1) | 4.02e+1(1.46e+1) | 3.91e+1(1.53e+1) |
| | 6 | 2.22e+1(5.99e+0) | 1.67e+1(4.35e+0) | 1.35e+1(6.23e+0) | 1.55e+1(5.29e+0) | 1.56e+1(7.51e+0) |
| | 10 | 1.52e-1(3.01e-2) | 1.67e-1(4.03e-2) | 1.58e-1(2.81e-2) | 1.58e-1(3.11e-2) | 1.64e-1(3.19e-2) |
| DTLZ2 | 3 | 4.40e-2(3.06e-3) | 4.38e-2(4.17e-3) | 4.37e-2(3.41e-3) | 4.48e-2(3.51e-3) | 4.29e-2(4.38e-3) |
| | 6 | 1.84e-1(1.50e-2) | 1.79e-1(1.02e-2) | 1.80e-1(1.17e-2) | 1.79e-1(9.20e-3) | 1.80e-1(1.49e-2) |
| | 10 | 2.89e-1(1.00e-2) | 2.97e-1(1.40e-2) | 2.87e-1(1.71e-2) | 2.90e-1(1.22e-2) | 2.85e-1(1.09e-2) |
| DTLZ3 | 3 | 1.89e+2(4.68e+1) | 1.61e+2(3.71e+1) | 1.54e+2(4.89e+1) | 1.58e+2(3.45e+1) | 1.57e+2(3.17e+1) |
| | 6 | 7.44e+1(2.34e+1) | 6.06e+1(1.32e+1) | 6.01e+1(2.61e+1) | 6.65e+1(2.14e+1) | 6.44e+1(2.63e+1) |
| | 10 | 4.65e-1(1.12e-1) | 4.70e-1(8.67e-2) | 4.84e-1(5.71e-2) | 4.92e-1(1.38e-1) | 4.61e-1(4.94e-2) |
| DTLZ4 | 3 | 8.66e-2(1.25e-1) | 1.35e-1(1.64e-1) | 1.06e-1(1.32e-1) | 8.82e-2(1.26e-1) | 1.04e-1(1.28e-1) |
| | 6 | 1.69e-1(2.20e-2) | 1.80e-1(3.27e-2) | 1.79e-1(4.06e-2) | 1.81e-1(4.77e-2) | 1.79e-1(2.78e-2) |
| | 10 | 2.29e-1(1.15e-2) | 2.30e-1(1.06e-2) | 2.38e-1(1.56e-2) | 2.37e-1(2.00e-2) | 2.37e-1(1.88e-2) |
| DTLZ5 | 3 | 9.75e-3(2.19e-3) | 8.93e-3(1.67e-3) | 8.98e-3(1.67e-3) | 9.15e-3(1.58e-3) | 8.80e-3(1.44e-3) |
| | 6 | 3.12e-2(9.30e-3) | 2.98e-2(1.02e-2) | 3.31e-2(7.84e-3) | 2.72e-2(7.30e-3) | 3.00e-2(1.05e-2) |
| | 10 | 5.60e-3(1.76e-3) | 3.92e-3(6.78e-4) | 4.85e-3(1.78e-3) | 5.65e-3(2.12e-3) | 6.02e-3(1.70e-3) |
| DTLZ6 | 3 | 4.87e-2(2.65e-2) | 4.28e-2(2.73e-2) | 6.38e-2(7.62e-2) | 9.93e-2(2.14e-1) | 5.04e-2(3.71e-2) |
| | 6 | 1.09e+0(1.19e+0) | 1.11e+0(1.07e+0) | 7.28e-1(1.00e+0) | 1.01e+0(1.13e+0) | 8.36e-1(1.16e+0) |
| | 10 | 2.25e-2(7.14e-3) | 6.20e-2(5.11e-2) | 3.92e-2(3.62e-2) | 3.51e-2(3.23e-2) | 4.42e-2(4.00e-2) |
| DTLZ7 | 3 | 6.96e-2(3.03e-2) | 7.83e-2(5.28e-2) | 1.36e-1(1.32e-1) | 1.28e-1(1.31e-1) | 9.71e-2(5.24e-2) |
| | 6 | 6.96e-1(2.65e-1) | 1.68e+0(8.29e-1) | 1.21e+0(7.32e-1) | 1.16e+0(6.33e-1) | 1.74e+0(8.02e-1) |
| | 10 | 1.24e+0(1.54e-1) | 1.20e+0(9.84e-2) | 1.23e+0(1.33e-1) | 1.20e+0(8.92e-2) | 1.25e+0(1.08e-1) |
| WFG1 | 3 | 1.67e+0(4.91e-2) | 1.64e+0(5.90e-2) | 1.67e+0(4.86e-2) | 1.62e+0(3.43e-2) | 1.61e+0(4.98e-2) |
| | 6 | 2.27e+0(5.70e-2) | 2.24e+0(5.05e-2) | 2.21e+0(5.69e-2) | 2.21e+0(7.43e-2) | 2.20e+0(6.16e-2) |
| | 10 | 2.67e+0(8.46e-2) | 2.56e+0(1.07e-1) | 2.55e+0(1.15e-1) | 2.64e+0(7.62e-2) | 2.61e+0(8.36e-2) |
| WFG2 | 3 | 2.63e-1(3.41e-2) | 2.63e-1(3.89e-2) | 2.48e-1(5.57e-2) | 2.47e-1(4.40e-2) | 2.44e-1(5.40e-2) |
| | 6 | 5.17e-1(1.03e-1) | 5.43e-1(1.35e-1) | 5.35e-1(9.94e-2) | 5.24e-1(1.26e-1) | 5.09e-1(1.49e-1) |
| | 10 | 1.39e+0(4.37e-1) | 1.39e+0(3.77e-1) | 1.36e+0(3.13e-1) | 1.40e+0(2.71e-1) | 1.38e+0(3.83e-1) |
| WFG3 | 3 | 2.57e-1(3.61e-2) | 2.64e-1(7.85e-2) | 2.51e-1(3.82e-2) | 2.78e-1(5.66e-2) | 2.48e-1(2.96e-2) |
| | 6 | 6.25e-1(1.13e-1) | 5.89e-1(6.72e-2) | 5.83e-1(8.20e-2) | 5.80e-1(7.49e-2) | 6.56e-1(1.04e-1) |
| | 10 | 6.67e-1(8.95e-2) | 6.93e-1(9.45e-2) | 6.93e-1(1.22e-1) | 7.03e-1(9.06e-2) | 7.47e-1(8.54e-2) |
| WFG4 | 3 | 2.56e-1(3.27e-2) | 2.49e-1(2.04e-2) | 2.49e-1(2.61e-2) | 2.48e-1(1.75e-2) | 2.41e-1(1.77e-2) |
| | 6 | 1.30e+0(1.91e-1) | 1.34e+0(2.28e-1) | 1.35e+0(3.15e-1) | 1.20e+0(2.23e-1) | 1.38e+0(2.88e-1) |
| | 10 | 3.68e+0(6.78e-1) | 3.87e+0(7.96e-1) | 3.86e+0(6.03e-1) | 3.83e+0(7.38e-1) | 3.65e+0(3.90e-1) |
| WFG5 | 3 | 3.17e-1(1.22e-1) | 3.50e-1(1.07e-1) | 3.06e-1(1.05e-1) | 3.12e-1(1.25e-1) | 2.92e-1(1.28e-1) |
| | 6 | 1.78e+0(9.49e-2) | 1.76e+0(1.11e-1) | 1.72e+0(1.26e-1) | 1.73e+0(9.61e-2) | 1.74e+0(1.33e-1) |
| | 10 | 3.79e+0(2.92e-1) | 3.59e+0(2.81e-1) | 3.63e+0(4.80e-1) | 3.87e+0(3.19e-1) | 3.79e+0(2.71e-1) |
| WFG6 | 3 | 4.48e-1(1.00e-1) | 5.24e-1(1.08e-1) | 4.87e-1(1.00e-1) | 4.86e-1(9.23e-2) | 4.64e-1(9.08e-2) |
| | 6 | 1.65e+0(1.84e-1) | 1.63e+0(8.15e-2) | 1.62e+0(1.85e-1) | 1.61e+0(1.48e-1) | 1.59e+0(2.47e-1) |
| | 10 | 3.35e+0(4.95e-1) | 3.51e+0(3.14e-1) | 3.19e+0(2.14e-1) | 3.33e+0(3.76e-1) | 3.14e+0(5.76e-1) |
| WFG7 | 3 | 2.90e-1(3.37e-2) | 3.14e-1(3.26e-2) | 2.95e-1(2.76e-2) | 2.95e-1(2.68e-2) | 2.90e-1(3.27e-2) |
| | 6 | 1.62e+0(2.02e-1) | 1.72e+0(1.37e-1) | 1.58e+0(1.47e-1) | 1.61e+0(1.63e-1) | 1.64e+0(1.85e-1) |
| | 10 | 4.55e+0(3.72e-1) | 4.81e+0(3.13e-1) | 4.76e+0(4.89e-1) | 4.82e+0(3.93e-1) | 4.51e+0(2.58e-1) |
| WFG8 | 3 | 5.91e-1(6.73e-2) | 6.06e-1(5.44e-2) | 5.71e-1(4.02e-2) | 5.77e-1(3.92e-2) | 5.61e-1(3.98e-2) |
| | 6 | 2.20e+0(1.50e-1) | 2.20e+0(1.48e-1) | 2.21e+0(1.21e-1) | 2.24e+0(1.57e-1) | 2.16e+0(1.06e-1) |
| | 10 | 4.99e+0(4.45e-1) | 5.15e+0(4.48e-1) | 5.06e+0(5.80e-1) | 5.00e+0(3.93e-1) | 4.90e+0(5.04e-1) |
| WFG9 | 3 | 3.68e-1(1.03e-1) | 4.43e-1(1.41e-1) | 3.81e-1(1.02e-1) | 3.85e-1(9.50e-2) | 3.56e-1(6.48e-2) |
| | 6 | 1.54e+0(1.81e-1) | 1.51e+0(1.73e-1) | 1.48e+0(2.27e-1) | 1.45e+0(1.19e-1) | 1.48e+0(1.75e-1) |
| | 10 | 4.02e+0(2.34e-1) | 3.97e+0(4.11e-1) | 4.02e+0(4.62e-1) | 3.94e+0(3.94e-1) | 3.96e+0(3.20e-1) |
| +/ ≈ /− | $n_c$=1 | -/-/- | 2/43/3 | 1/41/6 | 1/42/5 | 3/41/4 |
| +/ ≈ /− | $n_c$=3 | 3/43/2 | -/-/- | 0/46/2 | 2/45/1 | 1/41/6 |
| +/ ≈ /− | $n_c$=5 | 6/41/1 | 2/46/0 | -/-/- | 1/45/2 | 2/45/1 |
| +/ ≈ /− | $n_c$=7 | 5/42/1 | 1/45/2 | 2/45/1 | -/-/- | 2/45/1 |
| +/ ≈ /− | $n_c$=10 | 4/41/3 | 6/41/1 | 1/45/2 | 1/45/2 | -/-/- |

also significantly inferior to two other variants. The variant of $n_c = 1$ reaches the worst optimization results as it is significantly inferior to all other variants. In addition, considering that the variant of $n_c = 7$ wins/ties/losses 2/45/1 statistical tests when compared with the variant of $n_c = 5$, we set $n_c = 7$ for LORA-MOO.

The result of this ablation study demonstrates the influence of population initialization on the optimization results. By clustering the evaluated solutions into several clusters and sampling the same amount of initial solutions from each cluster, the solutions in the initial population are distributed in a more diverse way than the solutions sampled from the set of reference points $S_{RP}$ directly.

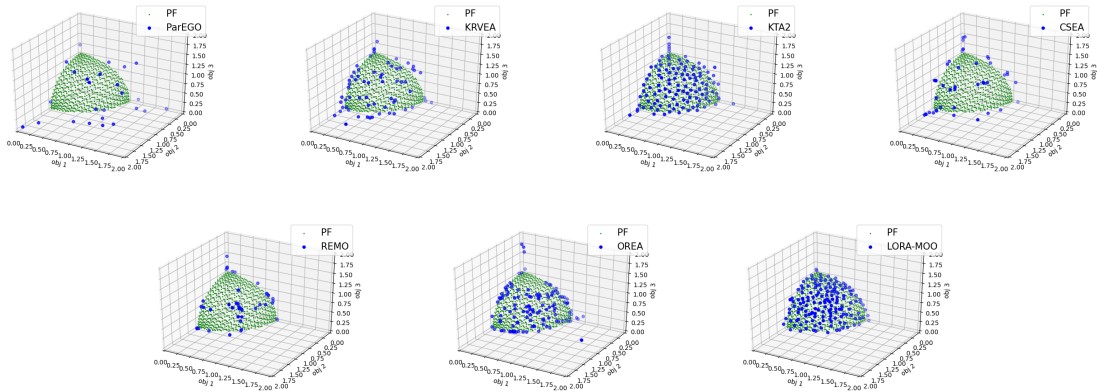

Figure 7: Distribution of obtained non-dominated solutions on DTLZ2 with 10 variables and 3 objectives.

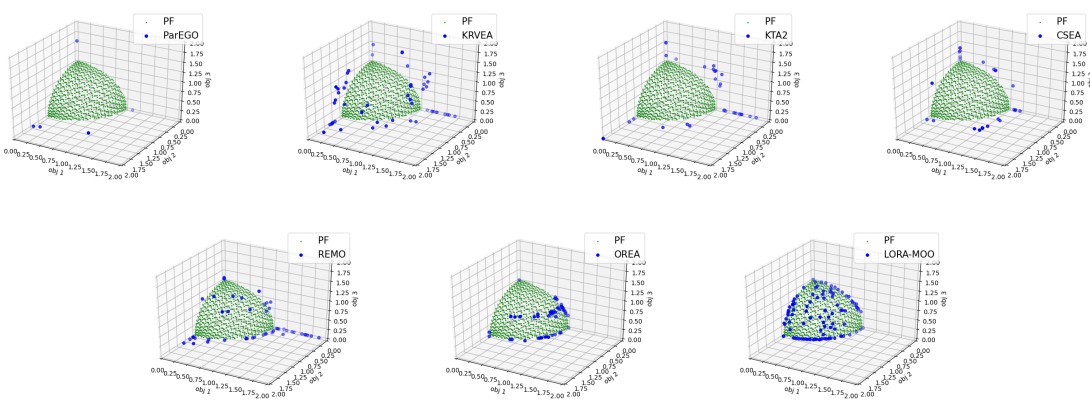

Figure 8: Distribution of obtained non-dominated solutions on DTLZ4 with 10 variables and 3 objectives.

Consequently, all variants of $n_c > 1$ have achieved better optimization results than the variant of $n_c = 1$.

# G    Solution Distribution

The solution distribution we obtained on some 3-objective DTLZ problems are plotted.

# H    Complete Results of Benchmark Optimization

In Section 4.3 of the main file, we display the optimization results of comparison algorithms on DTLZ problems in terms of IGD values. In this section, we provide detailed IGD results on WFG problems and more results on IGD+ and HV values. In addition, the optimization results on DTLZ problems with different scales, such as $D = 5$ and 20, are reported.

## H.1    IGD Results on WFG Optimization Problems

Table 7 shows the optimization results on WFG problems in terms of IGD values. The last row summarizes the results of statistical tests, which has reported at the end of Table 1 in the main file. It can be seen that LORA-MOO outperforms all comparison algorithms, followed by KTA2 and

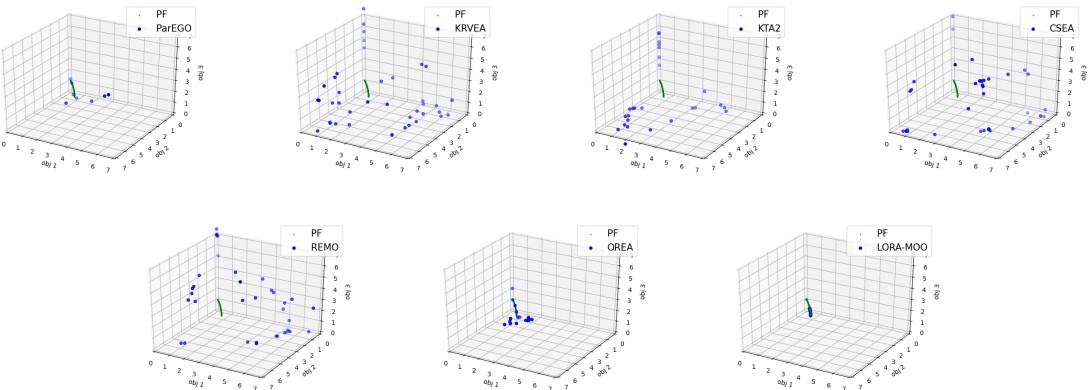

Figure 9: Distribution of obtained non-dominated solutions on DTLZ6 with 10 variables and 3 objectives.

Table 7: Statistical results of the IGD value obtained by comparison algorithms on 45 WFG optimization problems over 30 runs. Symbols '+', '≈', '−' denote LORA-MOO is statistically significantly superior to, almost equivalent to, and inferior to the compared algorithms in the Wilcoxon rank sum test (significance level is 0.05), respectively. The last row counts the total win/tie/loss results.

| Problems | M | ParEGO | KRVEA | KTA2 | CSEA | REMO | OREA | LORA-MOO |
|---|---|---|---|---|---|---|---|---|
| WFG1 | 3 | 1.65e+0(8.08e-2)− | 1.74e+0(9.91e-2)≈ | 1.87e+0(1.27e-1)+ | 1.74e+0(8.60e-2)≈ | 1.73e+0(1.12e-1)≈ | 2.03e+0(1.16e-1)+ | 1.71e+0(9.26e-2) |
| | 4 | 1.94e+0(7.04e-2)≈ | 2.07e+0(9.03e-2)+ | 2.18e+0(1.43e-1)+ | 2.05e+0(1.05e-1)+ | 1.96e+0(8.19e-2)≈ | 2.22e+0(9.54e-2)+ | 1.95e+0(7.52e-2) |
| | 6 | 2.38e+0(5.53e-2)≈ | 2.49e+0(6.57e-2)+ | 2.56e+0(9.95e-2)+ | 2.52e+0(9.89e-2)+ | 2.42e+0(5.34e-2)+ | 2.53e+0(1.04e-1)+ | 2.36e+0(5.07e-2) |
| | 8 | 2.75e+0(5.21e-2)≈ | 2.86e+0(7.05e-2)+ | 2.85e+0(1.06e-1)+ | 2.89e+0(5.19e-2)+ | 2.80e+0(7.44e-2)+ | 2.82e+0(7.56e-2)+ | 2.72e+0(6.21e-2) |
| | 10 | 3.08e+0(5.70e-2)+ | 3.11e+0(9.16e-2)+ | 2.99e+0(9.77e-2)+ | 3.09e+0(1.03e-1)+ | 3.04e+0(1.12e-1)+ | 3.10e+0(9.11e-2)+ | 2.93e+0(6.20e-2) |
| WFG2 | 3 | 7.66e-1(7.11e-2)+ | 3.61e-1(3.87e-2)+ | 4.24e-1(6.65e-2)+ | 5.48e-1(3.75e-2)+ | 5.22e-1(7.67e-2)+ | 4.88e-1(6.53e-2)+ | 3.72e-1(4.87e-2) |
| | 4 | 1.05e+0(1.40e-1)+ | 5.00e-1(3.97e-2)− | 5.66e-1(3.80e-2)+ | 7.61e-1(1.21e-1)+ | 7.48e-1(1.23e-1)+ | 7.45e-1(1.45e-1)+ | 5.46e-1(3.53e-2) |
| | 6 | 1.90e+0(3.51e-1)+ | 7.77e-1(5.25e-2)− | 9.00e-1(5.39e-2)+ | 1.28e+0(4.02e-1)+ | 1.28e+0(3.75e-1)+ | 1.49e+0(3.76e-1)+ | 8.55e-1(7.00e-2) |
| | 8 | 2.74e+0(6.68e-1)+ | 1.06e+0(5.98e-2)− | 1.18e+0(1.14e-1)− | 2.10e+0(6.97e-1)+ | 1.90e+0(5.25e-1)+ | 2.06e+0(4.58e-1)+ | 1.24e+0(1.23e-1) |
| | 10 | 3.73e+0(9.41e-1)+ | 1.18e+0(9.32e-2)− | 1.37e+0(1.03e-1)− | 2.84e+0(8.61e-1)+ | 2.59e+0(9.91e-1)+ | 2.95e+0(7.55e-1)+ | 1.83e+0(2.27e-1) |
| WFG3 | 3 | 5.82e-1(3.86e-2)+ | 5.39e-1(5.81e-2)+ | 3.29e-1(5.99e-2)+ | 5.04e-1(6.26e-2)+ | 4.60e-1(5.94e-2)+ | 3.85e-1(4.76e-2)+ | 2.83e-1(5.99e-2) |
| | 4 | 7.30e-1(6.25e-2)+ | 6.66e-1(7.02e-2)+ | 5.63e-1(6.47e-2)+ | 6.05e-1(7.26e-2)+ | 5.64e-1(6.43e-2)+ | 5.68e-1(5.92e-2)+ | 4.13e-1(5.98e-2) |
| | 6 | 7.75e-1(9.36e-2)+ | 6.76e-1(1.32e-1)+ | 7.94e-1(6.73e-2)+ | 7.41e-1(8.33e-2)+ | 6.37e-1(9.55e-2)≈ | 7.96e-1(6.68e-2)+ | 6.51e-1(9.20e-2) |
| | 8 | 8.38e-1(1.63e-1)≈ | 8.27e-1(9.79e-2)+ | 9.45e-1(7.42e-2)+ | 7.63e-1(1.06e-1)− | 6.25e-1(1.18e-1)− | 8.92e-1(9.90e-2)+ | 8.54e-1(9.98e-2) |
| | 10 | 6.85e-1(1.02e-1)+ | 6.87e-1(8.79e-2)− | 9.16e-1(8.20e-2)+ | 5.91e-1(9.34e-2)− | 5.19e-1(1.04e-1)− | 7.28e-1(1.10e-1)− | 8.23e-1(1.14e-1) |
| WFG4 | 3 | 6.21e-1(3.68e-2)+ | 4.67e-1(2.33e-2)+ | 4.21e-1(2.21e-2)+ | 4.57e-1(2.88e-2)+ | 4.23e-1(2.53e-2)+ | 4.34e-1(5.63e-2)+ | 3.36e-1(2.95e-2) |
| | 4 | 1.11e+0(3.45e-2)+ | 7.86e-1(2.45e-2)+ | 7.78e-1(4.50e-2)+ | 9.83e-1(1.22e-1)+ | 8.46e-1(8.32e-2)+ | 1.07e+0(1.18e-1)+ | 6.82e-1(4.97e-2) |
| | 6 | 2.75e+0(2.36e-1)+ | 1.87e+0(8.92e-2)≈ | 1.78e+0(7.66e-2)− | 3.13e+0(3.86e-1)+ | 2.69e+0(3.61e-1)+ | 2.92e+0(3.04e-1)+ | 1.86e+0(1.30e-1) |
| | 8 | 5.09e+0(9.78e-1)+ | 3.47e+0(2.96e-1)+ | 3.26e+0(1.67e-1)− | 5.81e+0(5.38e-1)+ | 4.99e+0(4.67e-1)+ | 5.76e+0(4.34e-1)+ | 3.62e+0(3.31e-1) |
| | 10 | 7.18e+0(1.21e+0)+ | 5.60e+0(6.92e-1)+ | 4.97e+0(1.72e-1)+ | 8.58e+0(8.39e-1)+ | 7.78e+0(8.13e-1)+ | 8.03e+0(5.03e-1)+ | 5.47e+0(4.14e-1) |
| WFG5 | 3 | 4.21e-1(3.05e-2)+ | 3.91e-1(4.22e-2)+ | 3.30e-1(9.56e-2)− | 5.50e-1(3.05e-2)+ | 5.30e-1(4.46e-2)+ | 4.51e-1(6.51e-2)+ | 4.21e-1(1.35e-1) |
| | 4 | 9.98e-1(8.09e-2)≈ | 7.65e-1(2.86e-2)− | 7.20e-1(6.23e-2)− | 8.87e-1(3.98e-2)− | 8.61e-1(4.68e-2)− | 1.02e+0(4.57e-2)+ | 9.81e-1(5.76e-2) |
| | 6 | 2.82e+0(1.65e-1)+ | 1.78e+0(6.23e-2)− | 1.92e+0(1.03e-1)− | 2.35e+0(1.86e-1)+ | 2.04e+0(1.29e-1)− | 2.44e+0(1.08e-1)+ | 2.11e+0(9.10e-2) |
| | 8 | 5.25e+0(2.55e-1)+ | 3.30e+0(2.61e-1)− | 3.62e+0(2.64e-1)+ | 4.75e+0(3.77e-1)+ | 3.95e+0(2.83e-1)+ | 4.57e+0(1.82e-1)+ | 3.66e+0(9.43e-2) |
| | 10 | 7.64e+0(3.23e-1)+ | 4.67e+0(4.78e-1)− | 4.76e+0(1.99e-1)− | 6.88e+0(4.23e-1)+ | 6.11e+0(4.62e-1)+ | 6.68e+0(3.49e-1)+ | 4.98e+0(1.57e-1) |
| WFG6 | 3 | 7.96e-1(5.50e-2)+ | 7.05e-1(5.10e-2)+ | 6.22e-1(8.49e-2)+ | 7.19e-1(4.80e-2)+ | 7.09e-1(4.61e-2)+ | 5.79e-1(4.68e-2)+ | 5.67e-1(1.09e-1) |
| | 4 | 1.14e+0(3.47e-2)+ | 1.02e+0(4.96e-2)+ | 9.62e-1(4.46e-2)+ | 1.08e+0(4.82e-2)+ | 1.04e+0(4.53e-2)+ | 1.17e+0(4.94e-2)+ | 9.51e-1(9.85e-2) |
| | 6 | 2.81e+0(2.60e-1)+ | 2.18e+0(7.41e-2)+ | 1.96e+0(4.17e-2)− | 2.56e+0(2.16e-1)+ | 2.20e+0(1.61e-1)+ | 2.77e+0(1.81e-1)+ | 2.04e+0(9.86e-2) |
| | 8 | 4.70e+0(5.78e-1)+ | 3.60e+0(1.17e-1)+ | 3.54e+0(1.85e-1)≈ | 4.70e+0(5.18e-1)+ | 4.13e+0(3.06e-1)+ | 5.06e+0(3.20e-1)+ | 3.52e+0(1.52e-1) |
| | 10 | 7.66e+0(5.36e-1)+ | 5.00e+0(1.33e-1)+ | 5.09e+0(1.58e-1)+ | 6.73e+0(5.98e-1)+ | 5.83e+0(4.69e-1)+ | 7.00e+0(4.90e-1)+ | 4.76e+0(1.94e-1) |
| WFG7 | 3 | 6.69e-1(2.70e-2)+ | 6.28e-1(2.45e-2)+ | 5.73e-1(2.76e-2)+ | 5.78e-1(3.23e-2)+ | 5.38e-1(3.58e-2)+ | 4.43e-1(4.15e-2)+ | 3.52e-1(2.22e-2) |
| | 4 | 1.13e+0(4.94e-2)+ | 9.48e-1(2.66e-2)+ | 9.04e-1(2.51e-2)+ | 9.92e-1(8.75e-2)+ | 8.81e-1(3.49e-2)+ | 9.72e-1(7.29e-2)+ | 7.07e-1(4.29e-2) |
| | 6 | 3.17e+0(2.89e-1)+ | 2.00e+0(5.61e-2)≈ | 1.96e+0(5.97e-2)≈ | 2.71e+0(3.18e-1)+ | 2.18e+0(1.49e-1)+ | 2.71e+0(1.91e-1)+ | 1.96e+0(1.06e-1) |
| | 8 | 5.93e+0(3.95e-1)+ | 3.64e+0(1.23e-1)− | 3.37e+0(1.16e-1)− | 5.19e+0(5.20e-1)+ | 4.28e+0(4.59e-1)+ | 5.19e+0(3.07e-1)+ | 3.82e+0(1.63e-1) |
| | 10 | 8.78e+0(4.70e-1)+ | 5.31e+0(3.01e-1)− | 4.88e+0(1.76e-1)− | 8.07e+0(5.07e-1)+ | 6.77e+0(5.93e-1)+ | 7.57e+0(4.12e-1)+ | 5.73e+0(3.07e-1) |
| WFG8 | 3 | 8.45e-1(2.87e-2)+ | 6.42e-1(2.49e-2)+ | 5.09e-1(4.39e-2)− | 7.49e-1(4.33e-2)+ | 7.13e-1(3.87e-2)+ | 7.01e-1(4.35e-2)+ | 6.02e-1(3.64e-2) |
| | 4 | 1.33e+0(4.61e-2)+ | 1.14e+0(3.89e-2)≈ | 1.02e+0(3.96e-2)− | 1.26e+0(6.23e-2)+ | 1.20e+0(5.28e-2)+ | 1.36e+0(6.94e-2)+ | 1.13e+0(7.12e-2) |
| | 6 | 3.11e+0(2.82e-1)+ | 2.43e+0(7.15e-2)≈ | 2.28e+0(5.05e-2)− | 3.00e+0(1.53e-1)+ | 2.80e+0(1.90e-1)+ | 3.07e+0(1.74e-1)+ | 2.45e+0(9.73e-2) |
| | 8 | 5.74e+0(3.56e-1)+ | 4.01e+0(2.28e-1)− | 3.92e+0(1.28e-1)− | 5.56e+0(3.24e-1)+ | 5.11e+0(4.10e-1)+ | 5.34e+0(2.72e-1)+ | 4.22e+0(2.75e-1) |
| | 10 | 8.30e+0(4.83e-1)+ | 5.56e+0(5.40e-1)− | 5.71e+0(3.80e-1)+ | 5.71e+0(3.80e-1)+ | 7.32e+0(4.74e-1)+ | 7.54e+0(4.88e-1)+ | 5.82e+0(2.95e-1) |
| WFG9 | 3 | 7.14e-1(5.09e-2)+ | 6.75e-1(6.73e-2)+ | 6.37e-1(8.35e-2)+ | 6.74e-1(8.53e-2)+ | 6.11e-1(9.76e-2)+ | 5.12e-1(7.74e-2)+ | 4.34e-1(8.18e-2) |
| | 4 | 1.24e+0(1.41e-1)+ | 1.06e+0(8.72e-2)+ | 1.07e+0(9.28e-2)+ | 1.16e+0(1.18e-1)+ | 1.05e+0(1.61e-1)+ | 1.02e+0(7.89e-2)+ | 8.43e-1(9.25e-2) |
| | 6 | 3.14e+0(2.96e-1)+ | 2.22e+0(1.94e-1)+ | 2.19e+0(1.52e-1)+ | 2.83e+0(2.36e-1)+ | 2.30e+0(1.82e-1)+ | 2.55e+0(1.21e-1)+ | 1.97e+0(9.18e-2) |
| | 8 | 5.78e+0(4.51e-1)+ | 3.93e+0(3.00e-1)+ | 3.77e+0(2.23e-1)+ | 5.43e+0(3.68e-1)+ | 4.60e+0(3.92e-1)+ | 4.73e+0(3.07e-1)+ | 3.61e+0(2.05e-1) |
| | 10 | 8.41e+0(4.80e-1)+ | 5.69e+0(6.42e-1)+ | 5.26e+0(3.13e-1)≈ | 7.77e+0(5.05e-1)+ | 6.48e+0(5.60e-1)+ | 6.74e+0(4.17e-1)+ | 5.16e+0(2.60e-1) |
| +/≈/− | | 39/4/2 | 21/10/14 | 23/6/16 | 41/1/3 | 38/3/4 | 43/1/1 | |

KRVEA. This is consistent with the results we observed from Table 1. The results on six 3- and 10-objective WFG problems are plotted in Fig. 10.

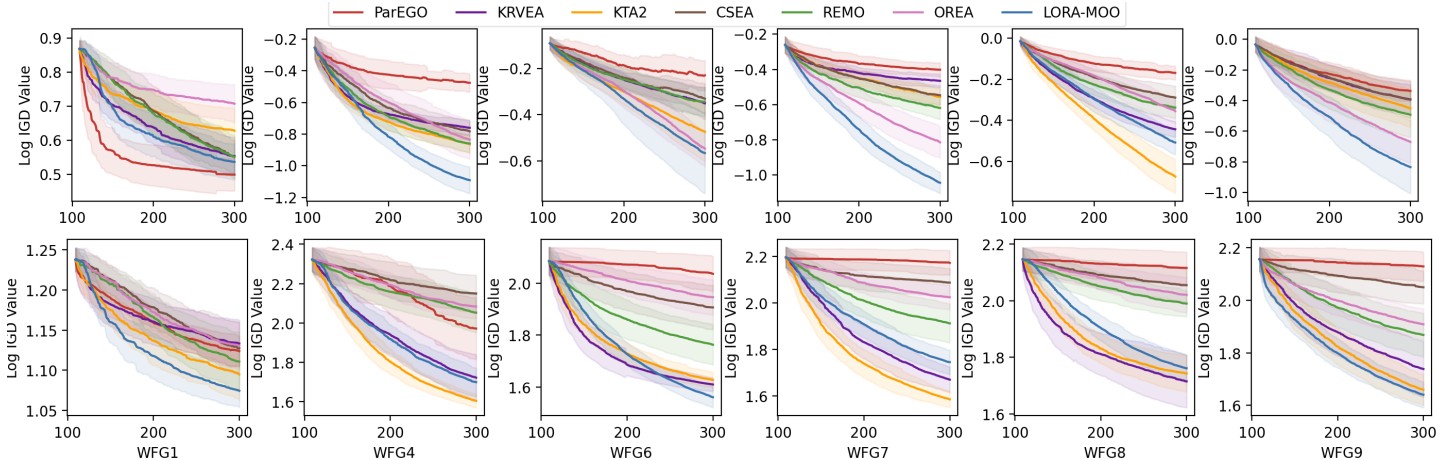

Figure 10: Log (IGD) curves averaged over 30 runs on six WFG problems for comparison algorithms (shaded area is ± std of the mean). **Top**: 10 variables and 3 objectives. **Bottom**: 10 variables and 10 objectives.

Table 8: Statistical results of the IGD+ value obtained by comparison algorithms on 35 DTLZ optimization problems over 30 runs. Symbols '+', '≈', '−' denote LORA-MOO is statistically significantly superior to, almost equivalent to, and inferior to the compared algorithms in the Wilcoxon rank sum test (significance level is 0.05), respectively. The last row counts the total win/tie/loss results.

| Problems | M | ParEGO | KRVEA | KTA2 | CSEA | REMO | OREA | LORA-MOO |
|---|---|---|---|---|---|---|---|---|
| DTLZ1 | 3 | 5.98e+1(3.81e+0)+ | 8.88e+1(2.16e+1)+ | 4.75e+1(1.55e+1)≈ | 6.30e+1(1.69e+1)+ | 5.06e+1(1.49e+1)+ | 4.44e+1(1.38e+1)≈ | 4.35e+1(1.80e+1) |
| | 4 | 4.68e+1(3.71e+0)+ | 6.45e+1(1.47e+1)+ | 4.08e+1(1.60e+1)+ | 3.69e+1(1.08e+1)≈ | 3.92e+1(1.11e+1)≈ | 3.80e+1(1.23e+1)≈ | 4.06e+1(1.34e+1) |
| | 6 | 3.04e+1(2.74e+0)+ | 3.22e+1(7.66e+0)+ | 2.03e+1(8.12e+0)+ | 1.56e+1(4.96e+0)≈ | 1.22e+1(4.65e+0)− | 1.74e+1(3.98e+0)≈ | 1.58e+1(6.17e+0) |
| | 8 | 1.23e+1(2.99e+0)+ | 8.52e+0(2.98e+0)+ | 4.54e+0(2.66e+0)+ | 5.08e+0(2.47e+0)+ | 3.33e+0(1.93e+0)+ | 5.87e+0(2.91e+0)+ | 3.82e+0(2.35e+0) |
| | 10 | 3.82e-1(1.79e-1)+ | 2.76e-1(1.14e-1)+ | 2.33e-1(9.65e-2)+ | 2.22e-1(8.29e-2)+ | 1.75e-1(7.84e-2)≈ | 1.83e-1(6.73e-2)+ | 1.56e-1(3.41e-2) |
| DTLZ2 | 3 | 2.61e-1(3.63e-2)+ | 9.22e-2(2.57e-2)+ | 3.82e-2(3.29e-3)+ | 1.60e-1(2.76e-2)+ | 1.01e-1(1.75e-2)+ | 5.86e-2(8.28e-3)+ | 4.47e-2(3.35e-3) |
| | 4 | 3.55e-1(4.11e-2)+ | 1.30e-1(3.08e-2)+ | 9.05e-2(6.95e-3)− | 2.05e-1(2.43e-2)+ | 1.60e-1(3.01e-2)+ | 1.37e-1(1.61e-2)+ | 9.74e-2(1.14e-2) |
| | 6 | 4.47e-1(2.32e-2)+ | 1.82e-1(1.49e-2)+ | 2.36e-1(3.71e-2)+ | 3.15e-1(4.24e-2)+ | 2.64e-1(3.18e-2)+ | 3.21e-1(2.78e-2)+ | 1.82e-1(1.15e-2) |
| | 8 | 4.68e-1(1.49e-2)+ | 2.34e-1(1.90e-2)− | 3.43e-1(2.37e-2)+ | 3.95e-1(2.66e-2)+ | 3.42e-1(2.91e-2)+ | 4.19e-1(1.86e-2)+ | 2.58e-1(1.88e-2) |
| | 10 | 4.33e-1(2.26e-2)+ | 2.92e-1(3.09e-2)+ | 3.15e-1(1.47e-2)+ | 4.17e-1(2.03e-2)+ | 3.61e-1(2.70e-2)+ | 4.28e-1(1.61e-2)+ | 2.88e-1(1.27e-2) |
| DTLZ3 | 3 | 1.66e+2(1.31e+1)+ | 2.43e+2(4.61e+1)+ | 1.52e+2(4.73e+1)≈ | 1.62e+2(4.84e+1)+ | 1.49e+2(3.88e+1)≈ | 1.26e+2(3.18e+1)− | 1.57e+2(3.83e+1) |
| | 4 | 1.42e+2(1.57e+1)+ | 1.83e+2(4.00e+1)+ | 1.18e+2(3.49e+1)≈ | 1.29e+2(3.58e+1)≈ | 1.16e+2(3.00e+1)≈ | 1.22e+2(4.13e+1)≈ | 1.25e+2(4.20e+1) |
| | 6 | 9.17e+1(1.59e+1)+ | 1.06e+2(2.96e+1)+ | 6.65e+1(2.63e+1)≈ | 5.27e+1(1.56e+1)≈ | 5.23e+1(1.71e+1)≈ | 5.24e+1(1.68e+1)≈ | 5.96e+1(2.05e+1) |
| | 8 | 4.13e+1(9.84e+0)+ | 2.96e+1(1.15e+1)+ | 1.73e+1(1.10e+1)+ | 1.59e+1(9.77e+0)+ | 1.60e+1(7.71e+0)+ | 1.49e+1(6.28e+0)+ | 1.26e+1(8.35e+0) |
| | 10 | 1.08e+0(3.73e-1)+ | 9.96e-1(4.96e-1)+ | 7.29e-1(2.75e-1)+ | 6.94e-1(2.89e-1)+ | 6.89e-1(3.18e-1)+ | 5.27e-1(6.34e-2)+ | 4.75e-1(1.13e-1) |
| DTLZ4 | 3 | 4.57e-1(7.52e-2)+ | 2.66e-1(1.02e-1)+ | 2.33e-1(8.36e-2)+ | 2.34e-1(7.76e-2)+ | 1.32e-1(6.41e-2)+ | 1.07e-1(9.68e-2)+ | 8.96e-2(1.25e-1) |
| | 4 | 4.86e-1(5.76e-2)+ | 2.84e-1(7.44e-2)+ | 2.95e-1(6.34e-2)+ | 2.03e-1(3.78e-2)+ | 1.66e-1(3.40e-2)+ | 1.35e-1(9.87e-2)≈ | 1.37e-1(9.79e-2) |
| | 6 | 4.24e-1(4.26e-2)+ | 2.94e-1(5.11e-2)+ | 3.61e-1(7.84e-2)+ | 2.41e-1(3.82e-2)+ | 2.27e-1(3.26e-2)+ | 1.67e-1(2.62e-2)+ | 1.78e-1(4.02e-2) |
| | 8 | 3.53e-1(2.66e-2)+ | 2.67e-1(3.51e-2)+ | 3.33e-1(4.56e-2)+ | 2.78e-1(3.65e-2)+ | 2.93e-1(3.63e-2)+ | 2.09e-1(2.55e-2)+ | 2.08e-1(1.89e-2) |
| | 10 | 2.86e-1(1.61e-2)+ | 2.58e-1(2.11e-2)+ | 2.88e-1(3.27e-2)+ | 2.92e-1(2.16e-2)+ | 3.06e-1(2.71e-2)+ | 2.29e-1(1.41e-2)+ | 2.30e-1(1.70e-2) |
| DTLZ5 | 3 | 1.60e-1(4.40e-2)+ | 9.18e-2(2.76e-2)+ | 8.66e-3(1.96e-3)≈ | 9.58e-2(2.60e-2)+ | 5.78e-2(1.81e-2)+ | 1.59e-2(5.12e-3)+ | 9.40e-3(1.93e-3) |
| | 4 | 1.47e-1(3.58e-2)+ | 4.96e-2(1.98e-2)+ | 3.25e-2(9.50e-3)+ | 9.78e-2(2.16e-2)+ | 7.51e-2(2.55e-2)+ | 2.88e-2(7.46e-3)+ | 2.21e-2(7.30e-3) |
| | 6 | 1.08e-1(2.44e-2)+ | 2.24e-2(7.50e-3)− | 8.02e-2(2.16e-2)+ | 6.16e-2(2.49e-2)+ | 4.14e-2(1.76e-2)+ | 3.89e-2(1.47e-2)+ | 3.20e-2(1.14e-2) |
| | 8 | 5.11e-2(7.70e-3)+ | 1.44e-2(5.17e-3)− | 5.35e-2(1.14e-2)+ | 2.49e-2(6.87e-3)+ | 2.01e-2(5.56e-3)≈ | 1.89e-2(5.87e-3)+ | 1.87e-2(3.21e-3) |
| | 10 | 1.19e-2(1.01e-3)+ | 6.26e-3(9.09e-4)+ | 1.19e-2(1.80e-3)+ | 7.45e-3(9.85e-4)+ | 4.80e-3(1.09e-3)− | 5.48e-3(9.49e-4)+ | 5.62e-3(1.75e-3) |
| DTLZ6 | 3 | 2.42e-1(1.07e-1)+ | 3.05e+0(5.23e-1)+ | 1.82e+0(4.48e-1)+ | 4.85e+0(6.38e-1)+ | 4.27e+0(5.48e-1)+ | 2.35e-1(4.14e-1)+ | 6.74e-2(1.55e-1) |
| | 4 | 2.64e-1(1.83e-1)+ | 2.44e+0(3.90e-1)+ | 1.84e+0(5.17e-1)+ | 5.12e+0(4.31e-1)+ | 4.07e+0(6.25e-1)+ | 1.35e+0(9.45e-1)+ | 2.07e-1(2.06e-1) |
| | 6 | 1.78e-1(1.07e-1)− | 1.33e+0(2.80e-1)+ | 1.49e+0(5.98e-1)+ | 3.14e+0(4.44e-1)+ | 2.32e+0(5.72e-1)+ | 2.04e+0(6.34e-1)+ | 9.00e-1(1.07e+0) |
| | 8 | 8.31e-2(2.90e-2)≈ | 4.48e-1(1.88e-1)+ | 8.28e-1(4.14e-1)+ | 1.53e+0(4.64e-1)+ | 9.18e-1(4.68e-1)+ | 1.03e+0(4.26e-1)+ | 2.96e-1(4.46e-1) |
| | 10 | 8.21e-2(9.39e-2)+ | 3.08e-2(1.03e-2)+ | 6.59e-2(5.61e-2)+ | 1.63e-1(2.40e-1)+ | 5.12e-2(1.09e-1)≈ | 1.15e-1(7.35e-2)+ | 3.30e-2(2.86e-2) |
| DTLZ7 | 3 | 1.10e-1(3.57e-2)+ | 7.39e-2(1.52e-2)+ | 1.54e-1(1.97e-1)− | 1.65e-1(6.43e-1)+ | 1.20e+0(5.73e-1)+ | 1.79e-1(1.20e-1)+ | 1.38e-1(1.53e-1) |
| | 4 | 4.98e-1(1.02e-1)+ | 2.20e-1(5.76e-2)≈ | 2.31e-1(1.27e-1)+ | 2.82e+0(6.75e-1)+ | 1.96e+0(7.49e-1)+ | 7.18e-1(4.34e-1)+ | 2.80e-1(1.73e-1) |
| | 6 | 1.07e+0(1.62e-1)≈ | 4.31e-1(3.82e-2)+ | 4.39e-1(1.48e-1)+ | 4.80e+0(1.01e+0)+ | 2.93e+0(7.01e-1)+ | 3.96e+0(1.88e+0)+ | 1.46e+0(6.89e-1) |
| | 8 | 1.28e+0(1.27e-1)− | 6.29e-1(7.74e-2)− | 7.72e-1(1.53e-1)− | 6.03e+0(1.87e+0)+ | 3.63e+0(5.55e-1)+ | 4.40e+0(2.74e+0)+ | 2.25e+0(6.88e-1) |
| | 10 | 1.51e+0(1.37e-1)+ | 9.42e-1(4.54e-2)+ | 1.11e+0(1.99e-1)− | 1.80e+0(3.39e-1)+ | 1.79e+0(3.78e-1)+ | 1.46e+0(2.55e-1)+ | 1.19e+0(8.31e-2) |
| +/≈/− | | 31/2/2 | 24/5/6 | 20/9/6 | 28/7/0 | 24/9/2 | 20/14/1 | |

## H.2 IGD+ Results on DTLZ and WFG Optimization Problems

Tables 8 and 9 display the IGD+ optimization results of comparison algorithms on DTLZ and WFG optimization problems, respectively. Different from IGD results, although LORA-MOO achieves the smallest IGD+ values on most DTLZ problems, its perform is competitive to KRVEA and KTA2 on WFG problems. However, from the perspective of overall performance, we can still conclude that our LORA-MOO outperforms all comparison algorithms on benchmark optimization problems in terms of IGD+ values. Such a observation is consistent with the results we observed from IGD values.

Table 9: Statistical results of the IGD+ value obtained by comparison algorithms on 45 WFG optimization problems over 30 runs. Symbols '+', '≈', '−' denote LORA-MOO is statistically significantly superior to, almost equivalent to, and inferior to the compared algorithms in the Wilcoxon rank sum test (significance level is 0.05), respectively. The last row counts the total win/tie/loss results.

| Problems | M | ParEGO | KRVEA | KTA2 | CSEA | REMO | OREA | LORA-MOO |
|---|---|---|---|---|---|---|---|---|
| WFG1 | 3 | 1.62e+0(3.90e-2)≈ | 1.68e+0(9.09e-2)+ | 1.78e+0(1.38e-1)+ | 1.68e+0(7.59e-2)+ | 1.69e+0(1.08e-1)+ | 1.92e+0(1.27e-1)+ | 1.63e+0(3.69e-2) |
| | 4 | 1.90e+0(6.54e-2)+ | 1.99e+0(1.02e-1)+ | 2.07e+0(1.47e-1)+ | 1.98e+0(1.06e-1)+ | 1.90e+0(8.14e-2)+ | 2.12e+0(8.95e-2)+ | 1.85e+0(7.27e-2) |
| | 6 | 2.30e+0(4.35e-2)+ | 2.36e+0(7.09e-2)+ | 2.41e+0(1.08e-1)+ | 2.37e+0(9.06e-2)+ | 2.29e+0(7.24e-2)+ | 2.39e+0(8.81e-2)+ | 2.22e+0(6.71e-2) |
| | 8 | 2.64e+0(4.48e-2)+ | 2.66e+0(7.65e-2)+ | 2.60e+0(1.15e-1)+ | 2.62e+0(6.34e-2)+ | 2.55e+0(6.82e-2)+ | 2.59e+0(4.96e-2)+ | 2.49e+0(7.00e-2) |
| | 10 | 2.88e+0(6.44e-2)+ | 2.78e+0(9.91e-2)+ | 2.65e+0(1.26e-1)≈ | 2.71e+0(1.27e-1)+ | 2.71e+0(1.22e-1)+ | 2.78e+0(1.04e-1)+ | 2.62e+0(7.81e-2) |
| WFG2 | 3 | 6.99e-1(9.48e-2)+ | 2.58e-1(4.09e-2)≈ | 2.39e-1(7.01e-2)≈ | 4.68e-1(5.12e-2)+ | 4.30e-1(9.29e-2)+ | 3.95e-1(7.73e-2)+ | 2.47e-1(4.89e-2) |
| | 4 | 9.74e-1(1.65e-1)+ | 3.21e-1(4.70e-2)− | 3.52e-1(5.16e-2)+ | 6.27e-1(1.42e-1)+ | 6.22e-1(1.45e-1)+ | 6.23e-1(1.69e-1)+ | 3.52e-1(5.74e-2) |
| | 6 | 1.77e+0(4.19e-1)+ | 3.84e-1(7.38e-2)− | 5.75e-1(1.00e-1)+ | 1.02e+0(4.94e-1)+ | 1.01e+0(4.70e-1)+ | 1.33e+0(4.17e-1)+ | 5.29e-1(1.26e-1) |
| | 8 | 2.55e+0(7.48e-1)+ | 4.09e-1(1.34e-1)− | 6.82e-1(1.43e-1)+ | 1.77e+0(8.24e-1)+ | 1.52e+0(6.54e-1)+ | 1.84e+0(4.86e-1)+ | 8.28e-1(1.52e-1) |
| | 10 | 3.49e+0(1.01e+0)+ | 4.18e-1(1.81e-1)− | 8.19e-1(1.39e-1)− | 2.49e+0(9.71e-1)+ | 2.19e+0(1.13e+0)+ | 2.67e+0(8.17e-1)+ | 1.40e+0(2.64e-1) |
| WFG3 | 3 | 5.65e-1(4.14e-2)+ | 5.26e-1(5.99e-2)+ | 3.05e-1(6.02e-2)+ | 4.87e-1(6.70e-2)+ | 4.42e-1(6.58e-2)+ | 3.67e-1(4.79e-2)+ | 2.65e-1(5.63e-2) |
| | 4 | 7.12e-1(6.70e-2)+ | 6.35e-1(6.90e-2)+ | 5.33e-1(6.42e-2)+ | 5.75e-1(7.97e-2)+ | 5.24e-1(7.33e-2)+ | 5.47e-1(6.00e-2)+ | 3.88e-1(6.09e-2) |
| | 6 | 7.42e-1(9.98e-2)+ | 6.24e-1(1.35e-1)+ | 7.25e-1(7.13e-2)+ | 6.91e-1(8.44e-2)+ | 5.60e-1(9.53e-2)≈ | 7.62e-1(6.68e-2)+ | 6.04e-1(8.95e-2) |
| | 8 | 7.74e-1(1.66e-1)+ | 7.26e-1(1.06e-1)+ | 8.46e-1(7.67e-2)+ | 6.83e-1(1.06e-1)− | 5.18e-1(1.13e-1)− | 8.26e-1(1.01e-1)+ | 7.58e-1(9.00e-2) |
| | 10 | 5.78e-1(9.80e-2)− | 5.54e-1(8.05e-2)− | 7.80e-1(8.72e-2)+ | 4.91e-1(8.69e-2)− | 4.07e-1(9.40e-2)− | 6.44e-1(1.04e-1)≈ | 6.92e-1(1.07e-1) |
| WFG4 | 3 | 4.74e-1(4.21e-2)+ | 3.78e-1(2.17e-2)+ | 3.42e-1(2.35e-2)+ | 3.49e-1(3.80e-2)+ | 3.04e-1(2.99e-2)+ | 3.66e-1(6.70e-2)+ | 2.55e-1(3.20e-2) |
| | 4 | 8.04e-1(5.34e-2)+ | 5.86e-1(3.17e-2)+ | 6.00e-1(6.42e-2)+ | 7.81e-1(1.78e-1)+ | 6.15e-1(1.13e-1)+ | 9.50e-1(1.50e-1)+ | 4.85e-1(6.14e-2) |
| | 6 | 1.83e+0(3.74e-1)+ | 1.20e+0(1.52e-1)+ | 1.12e+0(1.55e-1)+ | 2.78e+0(4.35e-1)+ | 2.26e+0(4.42e-1)+ | 2.56e+0(4.05e-1)+ | 1.21e+0(2.18e-1) |
| | 8 | 3.39e+0(1.48e+0)≈ | 2.33e+0(5.25e-1)+ | 2.15e+0(3.46e-1)− | 5.15e+0(5.66e-1)+ | 4.22e+0(5.32e-1)+ | 5.19e+0(4.73e-1)+ | 2.55e+0(5.66e-1) |
| | 10 | 3.27e+0(2.29e+0)− | 4.00e+0(9.92e-1)≈ | 3.45e+0(3.75e-1)− | 7.46e+0(8.64e-1)+ | 6.61e+0(8.48e-1)+ | 7.03e+0(6.17e-1)+ | 3.92e+0(7.04e-1) |
| WFG5 | 3 | 2.07e-1(1.28e-2)− | 3.01e-1(3.82e-2)≈ | 2.38e-1(7.04e-2)− | 3.98e-1(3.16e-2)+ | 3.93e-1(5.70e-2)+ | 3.60e-1(7.41e-2)+ | 3.49e-1(1.55e-1) |
| | 4 | 7.09e-1(1.49e-1)− | 5.32e-1(4.45e-2)− | 4.97e-1(4.53e-2)− | 6.09e-1(6.70e-2)− | 6.13e-1(5.55e-2)− | 9.11e-1(6.00e-2)≈ | 8.68e-1(7.81e-2) |
| | 6 | 2.38e+0(2.47e-1)+ | 1.07e+0(1.36e-1)− | 1.38e+0(1.64e-1)− | 1.89e+0(2.56e-1)+ | 1.52e+0(2.17e-1)− | 2.13e+0(1.77e-1)+ | 1.71e+0(1.09e-1) |
| | 8 | 4.63e+0(2.89e-1)+ | 2.11e+0(5.15e-1)− | 2.74e+0(4.81e-1)≈ | 4.13e+0(4.55e-1)+ | 3.26e+0(4.42e-1)+ | 4.08e+0(2.55e-1)+ | 2.88e+0(2.00e-1) |
| | 10 | 6.67e+0(3.78e-1)+ | 2.48e+0(9.46e-1)− | 3.13e+0(5.04e-1)− | 5.90e+0(5.30e-1)+ | 5.16e+0(5.38e-1)+ | 5.84e+0(5.37e-1)+ | 3.87e+0(3.50e-1) |
| WFG6 | 3 | 5.52e-1(4.95e-2)+ | 6.19e-1(6.81e-2)+ | 5.70e-1(8.76e-2)+ | 5.71e-1(5.32e-2)+ | 5.65e-1(5.43e-2)+ | 5.09e-1(5.01e-2)+ | 5.21e-1(1.15e-1) |
| | 4 | 8.09e-1(7.65e-2)+ | 7.62e-1(9.60e-2)+ | 8.14e-1(6.51e-2)+ | 8.33e-1(7.44e-2)+ | 7.87e-1(7.30e-2)+ | 1.07e+0(7.09e-2)+ | 8.09e-1(1.12e-1) |
| | 6 | 2.25e+0(5.29e-1)+ | 1.28e+0(1.52e-1)− | 1.52e+0(9.93e-2)≈ | 2.17e+0(3.22e-1)+ | 1.74e+0(2.70e-1)+ | 2.52e+0(2.20e-1)+ | 1.60e+0(1.59e-1) |
| | 8 | 3.63e+0(9.69e-1)+ | 1.50e+0(2.46e-1)− | 2.66e+0(3.17e-1)≈ | 3.96e+0(7.85e-1)+ | 3.41e+0(4.65e-1)+ | 4.60e+0(3.93e-1)+ | 2.72e+0(2.95e-1) |
| | 10 | 6.42e+0(8.39e-1)+ | 1.27e+0(1.06e-1)− | 3.67e+0(3.06e-1)+ | 5.61e+0(7.46e-1)+ | 4.68e+0(6.46e-1)+ | 6.05e+0(7.21e-1)+ | 3.38e+0(4.60e-1) |
| WFG7 | 3 | 5.47e-1(3.21e-2)+ | 5.38e-1(3.52e-2)+ | 4.97e-1(3.13e-2)+ | 4.36e-1(3.98e-2)+ | 3.94e-1(4.46e-2)+ | 3.65e-1(5.17e-2)+ | 2.92e-1(2.42e-2) |
| | 4 | 9.25e-1(9.05e-2)+ | 7.42e-1(3.50e-2)+ | 7.47e-1(3.15e-2)+ | 7.74e-1(1.39e-1)+ | 6.29e-1(5.40e-2)+ | 8.46e-1(1.05e-1)+ | 5.38e-1(5.32e-2) |
| | 6 | 2.85e+0(3.54e-1)+ | 1.41e+0(1.08e-1)− | 1.41e+0(1.36e-1)− | 2.29e+0(4.59e-1)+ | 1.74e+0(2.09e-1)+ | 2.45e+0(2.22e-1)+ | 1.61e+0(1.56e-1) |
| | 8 | 5.37e+0(4.28e-1)+ | 2.59e+0(2.47e-1)− | 2.40e+0(3.16e-1)− | 4.51e+0(6.31e-1)+ | 3.62e+0(5.07e-1)+ | 4.68e+0(3.37e-1)+ | 3.28e+0(2.02e-1) |
| | 10 | 7.77e+0(5.41e-1)+ | 3.50e+0(4.76e-1)− | 3.47e+0(3.98e-1)− | 6.92e+0(5.90e-1)+ | 5.72e+0(6.38e-1)+ | 6.70e+0(4.31e-1)+ | 4.85e+0(3.42e-1) |
| WFG8 | 3 | 7.23e-1(3.76e-2)+ | 5.89e-1(2.95e-2)≈ | 4.72e-1(4.57e-2)− | 6.59e-1(5.09e-2)+ | 6.21e-1(1.47e-2)+ | 6.77e-1(1.74e-2)+ | 5.79e-1(4.03e-2) |
| | 4 | 1.19e+0(6.76e-2)+ | 1.01e+0(5.20e-2)− | 9.25e-1(5.15e-2)− | 1.14e+0(8.61e-2)+ | 1.07e+0(7.07e-2)+ | 1.30e+0(7.86e-2)+ | 1.07e+0(7.91e-2) |
| | 6 | 2.80e+0(3.88e-1)+ | 1.82e+0(1.29e-1)− | 1.96e+0(1.02e-1)− | 2.77e+0(1.80e-1)+ | 2.58e+0(2.23e-1)+ | 2.90e+0(2.21e-1)+ | 2.22e+0(1.47e-1) |
| | 8 | 5.23e+0(4.86e-1)+ | 2.93e+0(4.96e-1)− | 3.31e+0(2.44e-1)− | 5.13e+0(3.86e-1)+ | 4.69e+0(4.63e-1)+ | 4.98e+0(3.05e-1)+ | 3.78e+0(3.27e-1) |
| | 10 | 7.43e+0(5.62e-1)+ | 2.74e+0(1.25e+0)− | 4.75e+0(5.99e-1)− | 7.03e+0(5.46e-1)+ | 6.52e+0(3.98e-1)+ | 6.74e+0(5.72e-1)+ | 5.03e+0(3.92e-1) |
| WFG9 | 3 | 5.82e-1(7.28e-2)+ | 5.83e-1(7.77e-2)+ | 5.56e-1(9.06e-2)+ | 6.10e-1(1.00e-1)+ | 5.32e-1(1.12e-1)+ | 4.51e-1(8.67e-2)+ | 3.82e-1(8.04e-2) |
| | 4 | 1.00e+0(1.88e-1)+ | 8.56e-1(1.30e-1)+ | 8.76e-1(1.43e-1)+ | 1.00e+0(1.56e-1)+ | 8.59e-1(2.01e-1)+ | 8.50e-1(1.15e-1)+ | 6.77e-1(9.61e-2) |
| | 6 | 2.72e+0(3.83e-1)+ | 1.72e+0(2.90e-1)+ | 1.66e+0(2.48e-1)+ | 2.44e+0(3.25e-1)+ | 1.87e+0(2.59e-1)+ | 2.17e+0(1.80e-1)+ | 1.45e+0(1.42e-1) |
| | 8 | 5.14e+0(5.22e-1)+ | 3.05e+0(4.65e-1)+ | 2.82e+0(2.91e-1)≈ | 4.80e+0(4.05e-1)+ | 3.95e+0(4.95e-1)+ | 4.17e+0(3.83e-1)+ | 2.76e+0(3.72e-1) |
| | 10 | 7.30e+0(5.37e-1)+ | 4.30e+0(8.61e-1)+ | 3.81e+0(4.78e-1)+ | 6.66e+0(5.44e-1)+ | 5.47e+0(6.11e-1)+ | 5.75e+0(4.84e-1)+ | 3.98e+0(4.51e-1) |
| +/≈/− | | 37/4/4 | 16/10/19 | 18/11/16 | 41/1/3 | 38/3/4 | 42/3/0 | |

## H.3 HV Results on DTLZ and WFG Optimization Problems

Tables 10 and 11 report the HV optimization results of comparison algorithms on DTLZ and WFG optimization problems, respectively. Since the calculation of HV values on 8- and 10-obj optimization problems is very time-consuming, only the results obtained on 3-, 4-, and 6-objective optimization problems are displayed. Consistent with the IGD an IGD+ results obtained on 3-, 4-, and 6-objectives, our LORA-MOO achieves the best overall performance over all comparison algorithms, showing the effectiveness of LORA-MOO on addressing expensive many-objective optimization problems.

## H.4 Problems with Different Scales

In this subsection, we investigate the optimization performance of LORA-MOO when the number of decision variables $D$ is different. The experimental setups for all comparison algorithms are the same as the setups used in previous benchmark optimization problems, but the setup for optimization problems is different:

- The optimization problems have $D = \{5, 10, 20\}$ decision variables and $M = 3$ objectives.

- When $D = 5$ or 10, a dataset of size 11 $D$ - 1 is used for surrogate initialization. When $D$ = 20, since 11 $D$ - 1 would be greater than our evaluation budget (300), the size of initial dataset is set to 100.

Tables 12, 13, and 14 report the obtained IGD, IGD+, and HV values on benchmark optimization problems with different numbers of decision variables $D$, respectively. It can be seen from Table 12

Table 10: Statistical results of the HV value obtained by comparison algorithms on 21 DTLZ optimization problems over 30 runs. Symbols '+', '≈', '−' denote LORA-MOO is statistically significantly superior to, almost equivalent to, and inferior to the compared algorithms in the Wilcoxon rank sum test (significance level is 0.05), respectively. The last row counts the total win/tie/loss results.

| Problems | M | ParEGO | KRVEA | KTA2 | CSEA | REMO | OREA | LORA-MOO |
|---|---|---|---|---|---|---|---|---|
| DTLZ1 | 3 | 0.00e+0(0.00e+0)≈ | 0.00e+0(0.00e+0)≈ | 0.00e+0(0.00e+0)≈ | 0.00e+0(0.00e+0)≈ | 0.00e+0(0.00e+0)≈ | 0.00e+0(0.00e+0)≈ | 0.00e+0(0.00e+0) |
| | 4 | 0.00e+0(0.00e+0)≈ | 0.00e+0(0.00e+0)≈ | 0.00e+0(0.00e+0)≈ | 0.00e+0(0.00e+0)≈ | 0.00e+0(0.00e+0)≈ | 0.00e+0(0.00e+0)≈ | 0.00e+0(0.00e+0) |
| | 6 | 0.00e+0(0.00e+0)≈ | 0.00e+0(0.00e+0)≈ | 0.00e+0(0.00e+0)≈ | 0.00e+0(0.00e+0)≈ | 0.00e+0(0.00e+0)≈ | 0.00e+0(0.00e+0)≈ | 0.00e+0(0.00e+0) |
| DTLZ2 | 3 | 4.53e-2(2.22e-2)+ | 2.61e-1(4.46e-2)+ | 3.87e-1(6.59e-3)− | 1.55e-1(3.85e-2)+ | 2.49e-1(3.32e-2)+ | 3.49e-1(1.33e-2)+ | 3.77e-1(6.75e-3) |
| | 4 | 6.06e-2(2.65e-2)+ | 3.71e-1(6.43e-2)+ | 4.80e-1(1.34e-2)≈ | 1.95e-1(3.26e-2)+ | 3.09e-1(4.54e-2)+ | 3.87e-1(3.31e-2)+ | 4.75e-1(2.34e-2) |
| | 6 | 1.26e-1(1.87e-2)+ | 4.85e-1(4.22e-2)+ | 4.48e-1(7.23e-2)+ | 2.86e-1(4.80e-2)+ | 4.00e-1(4.15e-2)+ | 3.66e-1(3.09e-2)+ | 6.09e-1(2.27e-2) |
| DTLZ3 | 3 | 0.00e+0(0.00e+0)≈ | 0.00e+0(0.00e+0)≈ | 0.00e+0(0.00e+0)≈ | 0.00e+0(0.00e+0)≈ | 0.00e+0(0.00e+0)≈ | 0.00e+0(0.00e+0)≈ | 0.00e+0(0.00e+0) |
| | 4 | 0.00e+0(0.00e+0)≈ | 0.00e+0(0.00e+0)≈ | 0.00e+0(0.00e+0)≈ | 0.00e+0(0.00e+0)≈ | 0.00e+0(0.00e+0)≈ | 0.00e+0(0.00e+0)≈ | 0.00e+0(0.00e+0) |
| | 6 | 0.00e+0(0.00e+0)≈ | 0.00e+0(0.00e+0)≈ | 0.00e+0(0.00e+0)≈ | 0.00e+0(0.00e+0)≈ | 0.00e+0(0.00e+0)≈ | 0.00e+0(0.00e+0)≈ | 0.00e+0(0.00e+0) |
| DTLZ4 | 3 | 4.20e-4(2.03e-3)+ | 6.42e-2(5.54e-2)+ | 8.85e-2(7.53e-2)+ | 6.53e-2(3.42e-2)+ | 1.99e-1(6.05e-2)+ | 2.52e-1(6.75e-2)+ | 3.24e-1(9.98e-2) |
| | 4 | 3.27e-3(6.73e-3)+ | 8.79e-2(6.62e-2)+ | 8.14e-2(5.85e-2)+ | 1.46e-1(5.25e-2)+ | 2.52e-1(6.25e-2)+ | 3.66e-1(8.97e-2)+ | 3.93e-1(9.18e-2) |
| | 6 | 2.14e-2(2.69e-2)+ | 2.05e-1(9.66e-2)+ | 1.44e-1(8.78e-2)+ | 3.16e-1(6.50e-2)+ | 3.53e-1(7.16e-2)+ | 5.12e-1(5.37e-2)≈ | 5.17e-1(4.93e-2) |
| DTLZ5 | 3 | 7.49e-3(1.04e-2)+ | 2.60e-1(1.04e-2)≈ | 8.60e-1(1.99e-3)≈ | 2.54e-2(9.46e-3)+ | 4.66e-2(1.02e-2)+ | 8.48e-2(1.78e-3)≈ | 8.53e-2(2.03e-3) |
| | 4 | 4.12e-3(5.91e-3)+ | 2.35e-2(7.10e-3)+ | 3.31e-2(4.30e-3)+ | 1.10e-2(4.90e-3)+ | 1.65e-2(7.08e-3)+ | 3.55e-2(4.96e-3)+ | 3.73e-3(3.97e-3) |
| | 6 | 1.75e-3(1.88e-3)+ | 1.28e-2(2.87e-3)− | 8.26e-3(2.88e-3)≈ | 5.75e-3(3.24e-3)+ | 8.48e-3(3.87e-3)≈ | 9.99e-3(3.78e-3)≈ | 9.23e-3(3.37e-3) |
| DTLZ6 | 3 | 3.91e-3(7.22e-3)+ | 0.00e+0(0.00e+0)+ | 0.00e+0(0.00e+0)+ | 0.00e+0(0.00e+0)+ | 0.00e+0(0.00e+0)+ | 3.52e-2(2.51e-2)+ | 4.91e-2(2.38e-2) |
| | 4 | 1.78e-3(2.86e-3)+ | 0.00e+0(0.00e+0)+ | 2.07e-5(1.11e-4)+ | 0.00e+0(0.00e+0)+ | 0.00e+0(0.00e+0)+ | 2.60e-4(9.64e-4)+ | 7.45e-3(9.93e-3) |
| | 6 | 1.28e-3(2.18e-3)+ | 0.00e+0(0.00e+0)+ | 1.10e-5(5.88e-5)+ | 0.00e+0(0.00e+0)+ | 0.00e+0(0.00e+0)+ | 1.21e-0(6.50e-0)+ | 7.42e-4(2.53e-3) |
| DTLZ7 | 3 | 1.81e-1(4.40e-2)+ | 2.53e-1(9.02e-3)≈ | 2.81e-1(3.28e-2)− | 1.44e-2(2.31e-2)+ | 2.11e-2(2.95e-2)+ | 2.23e-1(3.95e-2)+ | 2.47e-1(3.63e-2) |
| | 4 | 9.45e-2(3.19e-2)+ | 1.95e-1(1.73e-2)+ | 2.36e-1(8.48e-3)− | 4.80e-4(2.04e-3)+ | 1.20e-2(2.15e-2)+ | 1.04e-1(4.79e-2)+ | 1.88e-1(1.33e-2) |
| | 6 | 3.12e-2(1.83e-2)+ | 1.02e-1(1.04e-2)≈ | 1.57e-1(1.62e-2)− | 5.56e-4(2.99e-3)+ | 1.55e-2(1.81e-2)+ | 8.81e-4(1.91e-3)+ | 1.05e-1(2.61e-2) |
| +/≈/− | | 14/7/0 | 11/9/1 | 8/9/4 | 15/6/0 | 14/7/0 | 10/11/0 | |

Table 11: Statistical results of the HV value obtained by comparison algorithms on 27 WFG optimization problems over 30 runs. Symbols '+', '≈', '−' denote LORA-MOO is statistically significantly superior to, almost equivalent to, and inferior to the compared algorithms in the Wilcoxon rank sum test (significance level is 0.05), respectively. The last row counts the total win/tie/loss results.

| Problems | M | ParEGO | KRVEA | KTA2 | CSEA | REMO | OREA | LORA-MOO |
|---|---|---|---|---|---|---|---|---|
| WFG1 | 3 | 1.92e-1(2.65e-2)− | 1.09e-1(3.15e-2)≈ | 6.25e-2(3.98e-2)+ | 8.61e-2(4.91e-2)≈ | 1.02e-1(4.70e-2)≈ | 1.57e-2(2.69e-2)+ | 1.07e-1(3.15e-2) |
| | 4 | 2.07e-1(2.96e-2)− | 1.14e-1(5.44e-2)+ | 7.27e-2(5.18e-2)+ | 1.17e-1(5.34e-2)+ | 1.66e-1(3.54e-2)+ | 2.84e-2(3.66e-2)+ | 1.70e-1(4.15e-2) |
| | 6 | 2.16e-1(8.50e-3)≈ | 1.46e-1(2.93e-2)+ | 1.11e-1(4.99e-2)+ | 1.23e-1(5.25e-2)+ | 1.76e-1(2.54e-2)+ | 1.12e-1(5.80e-2)+ | 2.11e-1(2.75e-2) |
| WFG2 | 3 | 5.76e-1(3.88e-2)+ | 7.46e-1(2.87e-2)≈ | 7.11e-1(3.38e-2)+ | 6.57e-1(2.85e-2)+ | 6.65e-1(4.44e-2)+ | 6.92e-1(2.96e-2)+ | 7.42e-1(3.11e-2) |
| | 4 | 6.14e-1(3.28e-2)+ | 8.20e-1(3.33e-2)− | 7.36e-1(3.33e-2)+ | 7.23e-1(4.35e-2)+ | 7.06e-1(4.68e-2)+ | 7.21e-1(3.81e-2)+ | 7.79e-1(3.30e-2) |
| | 6 | 6.46e-1(5.10e-2)+ | 8.51e-1(3.38e-2)≈ | 8.26e-1(3.84e-2)+ | 7.80e-1(5.00e-2)+ | 7.73e-1(5.46e-2)+ | 7.29e-1(4.17e-2)+ | 8.39e-1(3.76e-2) |
| WFG3 | 3 | 1.04e-1(1.96e-2)+ | 1.13e-1(1.80e-2)+ | 1.90e-1(2.71e-2)≈ | 1.20e-1(1.90e-2)+ | 1.27e-1(2.01e-2)+ | 1.62e-1(2.11e-2)+ | 1.91e-1(2.20e-2) |
| | 4 | 3.10e-2(2.15e-2)+ | 3.48e-2(1.41e-2)+ | 2.73e-2(1.70e-2)+ | 3.65e-2(2.01e-2)+ | 4.07e-2(1.92e-2)+ | 3.10e-2(2.15e-2)+ | 5.57e-2(1.56e-2) |
| | 6 | 1.10e-2(1.26e-2)− | 1.39e-3(2.87e-3)− | 0.00e+0(0.00e+0)≈ | 6.59e-5(2.13e-4)+ | 2.96e-3(8.32e-3)− | 0.00e+0(0.00e+0)≈ | 0.00e+0(0.00e+0) |
| WFG4 | 3 | 1.74e-1(1.18e-2)+ | 2.18e-1(1.10e-2)+ | 2.44e-1(1.30e-2)+ | 2.37e-1(1.46e-2)+ | 2.55e-1(1.52e-2)+ | 2.66e-1(2.01e-2)+ | 2.98e-1(1.58e-2) |
| | 4 | 2.12e-1(9.87e-3)+ | 2.97e-1(1.52e-2)+ | 3.18e-1(2.01e-2)+ | 2.96e-1(2.19e-2)+ | 3.33e-1(2.24e-2)+ | 2.97e-1(1.89e-2)+ | 3.91e-1(1.96e-2) |
| | 6 | 2.50e-1(1.18e-2)+ | 4.09e-1(3.09e-2)+ | 4.38e-1(2.23e-2)+ | 3.16e-1(2.50e-2)+ | 3.78e-1(2.82e-2)+ | 3.19e-1(2.08e-2)+ | 4.78e-1(2.39e-2) |
| WFG5 | 3 | 2.98e-1(1.33e-2)− | 2.55e-1(2.28e-2)≈ | 2.98e-1(4.75e-2)− | 2.03e-1(1.32e-2)+ | 2.08e-1(2.74e-2)+ | 2.45e-1(3.49e-2)+ | 2.51e-1(6.54e-2) |
| | 4 | 3.19e-1(2.64e-2)− | 3.21e-1(2.50e-2)− | 3.63e-1(3.37e-2)− | 2.92e-1(2.21e-2)− | 2.83e-1(2.44e-2)− | 2.16e-1(1.31e-2)− | 2.05e-1(3.01e-2) |
| | 6 | 3.39e-1(2.37e-2)− | 4.17e-1(3.07e-2)− | 3.72e-1(3.17e-2)− | 3.46e-1(2.51e-2)− | 3.53e-1(2.43e-2)− | 2.78e-1(1.48e-2)− | 2.66e-1(2.60e-2) |
| WFG6 | 3 | 1.15e-1(2.24e-2)+ | 1.20e-1(2.10e-2)+ | 1.29e-1(3.72e-2)+ | 1.29e-1(2.01e-2)+ | 1.31e-1(1.90e-2)+ | 1.87e-1(1.98e-2)≈ | 1.85e-1(4.25e-2) |
| | 4 | 1.83e-1(1.87e-2)+ | 2.18e-1(3.46e-2)≈ | 2.17e-1(2.49e-2)≈ | 1.87e-1(2.16e-2)+ | 2.05e-1(2.17e-2)+ | 1.96e-1(1.60e-2)+ | 2.33e-1(5.01e-2) |
| | 6 | 2.30e-1(2.14e-2)+ | 2.75e-1(4.76e-2)+ | 3.15e-1(2.12e-2)≈ | 2.49e-1(1.89e-2)+ | 2.93e-1(3.03e-2)+ | 2.42e-1(1.28e-2)+ | 3.11e-1(2.91e-2) |
| WFG7 | 3 | 1.43e-1(8.60e-3)+ | 1.44e-1(1.11e-2)+ | 1.75e-1(1.26e-2)+ | 1.91e-1(1.74e-2)+ | 2.13e-1(2.05e-2)+ | 2.53e-1(1.32e-2)+ | 2.87e-1(1.30e-2) |
| | 4 | 1.91e-1(1.45e-2)+ | 2.22e-1(1.23e-2)+ | 2.36e-1(1.09e-2)+ | 2.42e-1(1.97e-2)+ | 2.90e-1(2.08e-2)+ | 2.83e-1(1.74e-2)+ | 3.66e-1(2.21e-2) |
| | 6 | 2.25e-1(1.42e-2)+ | 3.24e-1(2.49e-2)+ | 3.38e-1(2.89e-2)+ | 3.16e-1(3.37e-2)+ | 3.77e-1(2.50e-2)+ | 3.07e-1(1.80e-2)+ | 4.06e-1(2.28e-2) |
| WFG8 | 3 | 9.39e-2(1.01e-2)+ | 1.48e-1(9.46e-3)+ | 2.14e-1(1.61e-2)− | 1.24e-1(1.35e-2)+ | 1.32e-1(1.24e-2)+ | 1.60e-1(1.44e-2)+ | 1.84e-1(9.51e-3) |
| | 4 | 1.32e-1(1.22e-2)+ | 2.03e-1(1.81e-2)≈ | 2.17e-1(1.76e-2)− | 1.57e-1(1.81e-2)+ | 1.79e-1(1.75e-2)+ | 1.80e-1(1.38e-2)+ | 1.95e-1(2.50e-2) |
| | 6 | 1.81e-1(1.26e-2)+ | 2.59e-1(2.37e-2)− | 2.58e-1(1.13e-2)− | 2.18e-1(2.14e-2)+ | 2.62e-1(1.21e-2)− | 2.17e-1(1.19e-2)+ | 2.40e-1(2.32e-2) |
| WFG9 | 3 | 1.22e-1(1.94e-2)+ | 1.28e-1(2.33e-2)+ | 1.50e-1(3.21e-2)+ | 1.39e-1(2.58e-2)+ | 1.67e-1(3.64e-2)+ | 2.23e-1(2.82e-2)+ | 2.46e-1(3.68e-2) |
| | 4 | 1.74e-1(3.27e-2)+ | 2.08e-1(3.51e-2)+ | 2.04e-1(2.90e-2)+ | 1.87e-1(3.11e-2)+ | 2.35e-1(4.04e-2)+ | 2.63e-1(2.48e-2)+ | 3.06e-1(4.82e-2) |
| | 6 | 2.14e-1(2.85e-2)+ | 3.31e-1(5.50e-2)+ | 3.65e-1(5.25e-2)≈ | 2.76e-1(3.85e-2)+ | 3.62e-1(3.76e-2)+ | 2.90e-1(2.96e-2)+ | 3.89e-1(3.60e-2) |
| +/≈/− | | 20/1/6 | 16/6/5 | 15/6/6 | 23/2/2 | 20/3/4 | 23/2/2 | |

685 that LORA-MOO outperforms all comparison algorithms on DTLZ optimization problems when $D$
686 = 5, 10, and 20. In addition, KTA2 reaches competitive optimization results on many optimization
687 problems. The observations from Tables 13 and 14 have demonstrated consistent conclusions.

Table 12: Statistical results of the IGD value obtained by comparison algorithms on $5D$, $10D$, and $20D$ DTLZ optimization problems over 30 runs. Symbols '+', '≈', '−' denote LORA-MOO is statistically significantly superior to, almost equivalent to, and inferior to the compared algorithms in the Wilcoxon rank sum test (significance level is 0.05), respectively. The last row counts the total win/tie/loss results.

| Problems | D | ParEGO | KRVEA | KTA2 | CSEA | REMO | OREA | LORA-MOO |
|---|---|---|---|---|---|---|---|---|
| DTLZ1 | 5 | 1.24e+1(4.40e+0)+ | 7.19e+0(3.77e+0)+ | 4.00e+0(2.28e+0)≈ | 5.71e+0(2.66e+0)≈ | 5.97e+0(2.98e+0)≈ | 2.27e+0(1.45e+0)− | 4.78e+0(2.80e+0) |
| | 10 | 5.98e+1(3.81e+0)+ | 8.88e+1(2.16e+1)+ | 4.75e+1(1.55e+1)+ | 6.30e+1(1.69e+1)+ | 5.06e+1(1.49e+1)+ | 4.44e+1(1.38e+1)≈ | 4.35e+1(1.80e+1) |
| | 20 | 1.59e+2(1.56e+1)− | 3.12e+2(3.79e+1)≈ | 2.48e+2(3.66e+1)+ | 2.35e+2(3.47e+1)− | 2.01e+2(3.95e+1)− | 2.94e+2(3.78e+1)+ | 2.91e+2(3.98e+1) |
| DTLZ2 | 5 | 1.81e-1(1.26e-2)+ | 6.06e-2(2.40e-3)+ | 4.39e-2(1.11e-3)≈ | 1.03e-1(7.78e-3)+ | 7.94e-2(7.71e-3)+ | 6.55e-2(6.87e-3)+ | 4.36e-2(2.15e-3) |
| | 10 | 3.38e-1(2.84e-2)+ | 1.32e-1(2.77e-2)+ | 6.17e-2(3.13e-3)≈ | 2.26e-1(2.61e-2)+ | 1.65e-1(2.18e-2)+ | 8.59e-2(8.51e-3)+ | 6.19e-2(3.48e-3) |
| | 20 | 7.15e-1(1.21e-1)+ | 6.66e-1(7.34e-2)+ | 2.85e-1(5.83e-2)+ | 5.17e-1(6.66e-2)+ | 4.00e-1(7.02e-2)+ | 1.62e-1(3.35e-2)+ | 1.02e-1(1.36e-2) |
| DTLZ3 | 5 | 3.17e+1(1.17e+1)+ | 1.91e+1(9.12e+0)+ | 1.17e+1(6.12e+0)+ | 1.58e+1(7.60e+0)≈ | 1.61e+1(9.16e+0)+ | 6.78e+0(4.79e+0)− | 1.51e+1(9.40e+0) |
| | 10 | 1.66e+2(1.31e+1)+ | 2.43e+2(4.61e+1)+ | 1.52e+2(4.73e+1)+ | 1.62e+2(4.84e+1)+ | 1.49e+2(3.88e+1)≈ | 1.26e+2(3.18e+1)− | 1.57e+2(3.83e+1) |
| | 20 | 4.32e+2(1.78e+1)− | 9.11e+2(8.72e+1)≈ | 7.23e+2(1.38e+2)− | 7.12e+2(1.10e+2)− | 5.86e+2(1.18e+2)− | 7.81e+2(1.20e+2)− | 8.58e+2(1.31e+2) |
| DTLZ4 | 5 | 4.33e-1(5.55e-2)≈ | 1.35e-1(6.05e-2)≈ | 1.68e-1(1.22e-1)≈ | 4.33e-1(1.54e-1)+ | 1.60e-1(6.12e-2)≈ | 2.91e-1(2.44e-1)≈ | 3.96e-1(3.71e-1) |
| | 10 | 6.70e-1(7.61e-2)+ | 3.32e-1(1.11e-1)+ | 3.49e-1(1.09e-1)+ | 4.62e-1(1.36e-1)+ | 2.31e-1(1.15e-1)+ | 2.39e-1(1.65e-1)+ | 1.89e-1(2.34e-1) |
| | 20 | 1.02e+0(1.04e-1)+ | 8.32e-1(1.36e-1)+ | 7.76e-1(1.29e-1)+ | 7.11e-1(1.74e-1)+ | 5.51e-1(1.18e-1)+ | 5.27e-1(2.75e-1)+ | 4.01e-1(3.28e-1) |
| DTLZ5 | 5 | 4.16e-2(9.61e-3)+ | 2.31e-2(3.02e-3)+ | 3.57e-3(2.35e-4)− | 2.18e-2(3.22e-3)+ | 1.49e-2(3.28e-3)+ | 1.12e-2(5.73e-3)+ | 4.20e-3(6.92e-4) |
| | 10 | 2.16e-1(4.45e-2)+ | 1.19e-1(3.38e-2)+ | 1.34e-2(2.83e-3)− | 1.18e-1(2.56e-2)+ | 7.36e-2(2.03e-2)+ | 2.02e-2(4.77e-3)+ | 1.26e-2(2.55e-3) |
| | 20 | 6.05e-1(1.43e-1)+ | 6.16e-1(7.41e-2)+ | 2.13e-1(5.07e-2)+ | 4.84e-1(8.14e-2)+ | 3.60e-1(8.07e-2)+ | 8.11e-2(3.39e-2)+ | 4.32e-2(1.45e-2) |
| DTLZ6 | 5 | 4.57e-2(1.11e-2)+ | 4.69e-1(1.54e-1)+ | 2.68e-1(1.01e-1)+ | 7.65e-1(4.09e-1)+ | 4.08e-1(2.59e-1)+ | 2.57e-2(2.92e-2)≈ | 2.98e-2(3.53e-2) |
| | 10 | 3.15e-1(1.62e-1)+ | 3.06e+0(5.21e-1)+ | 1.83e+0(4.37e-1)+ | 4.86e+0(6.30e-1)+ | 4.27e+0(5.49e-1)+ | 3.09e-1(3.99e-1)+ | 1.18e-1(1.57e-1) |
| | 20 | 3.54e+0(1.04e+0)≈ | 1.10e+1(7.15e-1)+ | 8.72e+0(1.01e+0)+ | 1.33e+1(8.48e-1)+ | 1.23e+1(7.84e-1)+ | 7.06e+0(3.05e+0)+ | 6.81e+0(5.11e+0) |
| DTLZ7 | 5 | 1.87e-1(2.40e-2)+ | 1.07e-1(1.50e-2)+ | 6.66e-2(4.28e-2)− | 5.67e-1(2.78e-1)+ | 2.30e-1(1.07e-1)+ | 3.05e-1(2.01e-1)+ | 1.41e-1(1.50e-2) |
| | 10 | 2.45e-1(4.80e-2)+ | 1.35e-1(2.37e-2)+ | 2.19e-1(2.40e-1)− | 1.75e+0(6.32e-1)+ | 1.27e+0(5.65e-1)+ | 2.73e-1(1.58e-1)+ | 2.01e-1(1.93e-1) |
| | 20 | 2.67e-1(4.98e-2)≈ | 4.17e-1(2.04e-1)+ | 4.69e-1(2.56e-1)+ | 3.69e+0(9.09e-1)+ | 2.62e+0(7.33e-1)+ | 4.77e-1(2.53e-1)+ | 2.99e-1(2.51e-1) |
| +/≈/− | | 16/3/2 | 16/5/0 | 7/9/5 | 16/3/2 | 15/4/2 | 12/5/4 | |

Table 13: Statistical results of the IGD+ value obtained by comparison algorithms on $5D$, $10D$, and $20D$ DTLZ optimization problems over 30 runs. Symbols '+', '≈', '−' denote LORA-MOO is statistically significantly superior to, almost equivalent to, and inferior to the compared algorithms in the Wilcoxon rank sum test (significance level is 0.05), respectively. The last row counts the total win/tie/loss results.

| Problems | D | ParEGO | KRVEA | KTA2 | CSEA | REMO | OREA | LORA-MOO |
|---|---|---|---|---|---|---|---|---|
| DTLZ1 | 5 | 1.24e+1(4.40e+0)+ | 7.19e+0(3.77e+0)+ | 4.00e+0(2.28e+0)≈ | 5.70e+0(2.67e+0)≈ | 5.97e+0(2.98e+0)≈ | 2.27e+0(1.45e+0)− | 4.78e+0(2.81e+0) |
| | 10 | 5.98e+1(3.81e+0)+ | 8.88e+1(2.16e+1)+ | 4.75e+1(1.55e+1)+ | 6.30e+1(1.69e+1)+ | 5.06e+1(1.49e+1)+ | 4.44e+1(1.38e+1)≈ | 4.35e+1(1.80e+1) |
| | 20 | 1.59e+2(1.56e+1)+ | 3.12e+2(3.79e+1)+ | 2.48e+2(3.66e+1)− | 2.35e+2(3.47e+1)− | 2.01e+2(3.95e+1)− | 2.94e+2(3.78e+1)≈ | 2.91e+2(3.98e+1) |
| DTLZ2 | 5 | 1.01e-1(7.98e-3)+ | 2.86e-2(9.66e-4)+ | 1.94e-2(6.20e-4)− | 5.24e-2(6.84e-3)+ | 3.83e-2(4.18e-3)+ | 3.92e-2(5.96e-3)+ | 2.30e-2(2.07e-3) |
| | 10 | 2.61e-1(3.63e-2)+ | 9.22e-2(2.57e-2)+ | 3.82e-2(3.29e-3)− | 1.60e-1(2.76e-2)+ | 1.01e-1(1.75e-2)+ | 5.86e-2(8.28e-3)+ | 4.47e-2(3.35e-3) |
| | 20 | 6.51e-1(1.39e-1)+ | 6.66e-1(7.19e-2)+ | 2.61e-1(5.87e-2)+ | 4.69e-1(6.69e-2)+ | 3.56e-1(8.04e-2)+ | 1.39e-1(3.02e-2)+ | 8.36e-2(1.22e-2) |
| DTLZ3 | 5 | 3.17e+1(1.17e+1)+ | 1.91e+1(9.13e+0)≈ | 1.17e+1(6.15e+0)+ | 1.58e+1(7.61e+0)≈ | 1.61e+1(9.16e+0)≈ | 6.77e+0(4.80e+0)− | 1.51e+1(9.41e+0) |
| | 10 | 1.66e+2(1.31e+1)+ | 2.43e+2(4.61e+1)+ | 1.52e+2(4.73e+1)≈ | 1.62e+2(4.84e+1)≈ | 1.49e+2(3.88e+1)≈ | 1.26e+2(3.18e+1)− | 1.57e+2(3.83e+1) |
| | 20 | 4.32e+2(1.78e+1)− | 9.11e+2(8.72e+1)≈ | 7.23e+2(1.38e+2)+ | 7.12e+2(1.10e+2)+ | 5.86e+2(1.18e+2)+ | 7.81e+2(1.20e+2)+ | 8.58e+2(1.31e+2) |
| DTLZ4 | 5 | 1.88e-1(3.03e-2)≈ | 7.41e-2(4.55e-2)≈ | 7.39e-2(5.63e-2)≈ | 1.80e-1(7.75e-2)+ | 6.02e-2(2.08e-2)≈ | 1.24e-1(1.32e-1)≈ | 1.96e-1(2.08e-1) |
| | 10 | 4.57e-1(7.52e-2)+ | 2.66e-1(1.02e-1)+ | 2.33e-1(8.36e-2)+ | 2.34e-1(7.76e-2)+ | 1.32e-1(6.41e-2)+ | 1.07e-1(9.68e-2)+ | 8.96e-2(1.25e-1) |
| | 20 | 6.79e-1(1.38e-1)+ | 7.74e-1(1.34e-1)+ | 6.65e-1(1.18e-1)+ | 5.50e-1(1.44e-1)+ | 4.63e-1(8.22e-2)+ | 3.16e-1(1.90e-1)+ | 2.27e-1(2.02e-1) |
| DTLZ5 | 5 | 2.37e-2(3.64e-3)+ | 1.30e-2(1.76e-3)+ | 1.65e-3(1.03e-4)− | 1.26e-2(2.08e-3)+ | 7.74e-3(1.49e-3)+ | 6.37e-3(2.67e-3)+ | 2.48e-3(5.73e-4) |
| | 10 | 1.60e-1(4.40e-2)+ | 9.18e-2(2.76e-2)+ | 8.66e-3(1.96e-3)− | 9.58e-2(2.60e-2)+ | 5.78e-2(1.81e-2)+ | 1.59e-2(5.12e-3)+ | 9.40e-3(1.93e-3) |
| | 20 | 5.52e-1(1.50e-1)+ | 5.91e-1(7.98e-2)+ | 2.01e-1(5.29e-2)+ | 4.67e-1(8.41e-2)+ | 3.49e-1(8.31e-2)+ | 7.69e-2(2.31e-2)+ | 3.93e-2(1.41e-2) |
| DTLZ6 | 5 | 2.47e-2(6.71e-3)+ | 3.89e-1(1.88e-1)+ | 2.13e-1(1.02e-1)+ | 7.13e-1(4.42e-1)+ | 3.64e-1(2.75e-1)+ | 9.09e-3(9.88e-3)≈ | 1.17e-2(1.30e-2) |
| | 10 | 2.42e-1(1.07e-1)+ | 3.05e+0(5.23e-1)+ | 1.82e+0(4.48e-1)+ | 4.85e+0(6.38e-1)+ | 4.27e+0(5.48e-1)+ | 2.35e-1(4.14e-1)+ | 6.74e-2(1.55e-1) |
| | 20 | 3.49e+0(1.06e+0)≈ | 1.10e+1(7.14e-1)+ | 8.71e+0(1.01e+0)+ | 1.33e+1(8.47e-1)+ | 1.23e+1(7.85e-1)+ | 7.04e+0(3.06e+0)+ | 6.77e+0(5.15e+0) |
| DTLZ7 | 5 | 7.68e-2(1.31e-2)+ | 4.68e-2(4.64e-3)+ | 3.52e-2(2.90e-2)≈ | 4.46e-1(2.51e-1)+ | 1.55e-1(8.32e-2)+ | 2.04e-1(1.80e-1)+ | 8.42e-2(1.14e-1) |
| | 10 | 1.10e-1(3.57e-2)+ | 7.39e-2(1.52e-2)≈ | 1.54e-1(1.97e-1)− | 1.65e+0(6.43e-1)+ | 1.20e+0(5.73e-1)+ | 1.79e-1(1.20e-1)+ | 1.38e-1(1.53e-1) |
| | 20 | 1.38e-1(4.67e-2)+ | 3.30e-1(1.80e-1)+ | 3.60e-1(2.27e-1)+ | 3.65e+0(9.08e-1)+ | 2.61e+0(7.28e-1)+ | 4.15e-1(2.30e-1)+ | 2.28e-1(2.10e-1) |
| +/≈/− | | 16/3/2 | 16/5/0 | 7/8/6 | 16/3/2 | 15/4/2 | 12/5/4 | |

688

Table 14: Statistical results of the HV value obtained by comparison algorithms on $5D$, $10D$, and $20D$ DTLZ optimization problems over 30 runs. Symbols '+', '≈', '−' denote LORA-MOO is statistically significantly superior to, almost equivalent to, and inferior to the compared algorithms in the Wilcoxon rank sum test (significance level is 0.05), respectively. The last row counts the total win/tie/loss results.

| Problems | D | ParEGO | KRVEA | KTA2 | CSEA | REMO | OREA | LORA-MOO |
|---|---|---|---|---|---|---|---|---|
| DTLZ1 | 5 | 0.00e+0(0.00e+0)≈ | 0.00e+0(0.00e+0)≈ | 0.00e+0(0.00e+0)≈ | 0.00e+0(0.00e+0)≈ | 0.00e+0(0.00e+0)≈ | 6.38e-4(3.44e-3)≈ | 1.10e-2(5.92e-2) |
| | 10 | 0.00e+0(0.00e+0)≈ | 0.00e+0(0.00e+0)≈ | 0.00e+0(0.00e+0)≈ | 0.00e+0(0.00e+0)≈ | 0.00e+0(0.00e+0)≈ | 0.00e+0(0.00e+0)≈ | 0.00e+0(0.00e+0) |
| | 20 | 0.00e+0(0.00e+0)≈ | 0.00e+0(0.00e+0)≈ | 0.00e+0(0.00e+0)≈ | 0.00e+0(0.00e+0)≈ | 0.00e+0(0.00e+0)≈ | 0.00e+0(0.00e+0)≈ | 0.00e+0(0.00e+0) |
| DTLZ2 | 5 | 2.15e-1(1.98e-2)+ | 4.00e-1(2.88e-3)+ | 4.26e-1(1.70e-3)− | 3.39e-1(1.61e-2)+ | 3.78e-1(1.08e-2)+ | 3.83e-1(1.22e-2)+ | 4.21e-1(4.35e-3) |
| | 10 | 4.53e-2(2.22e-2)+ | 2.61e-1(4.46e-2)+ | 3.87e-1(6.59e-3)− | 1.55e-1(3.85e-2)+ | 2.49e-1(3.32e-2)+ | 3.49e-1(1.33e-2)+ | 3.77e-1(6.75e-3) |
| | 20 | 1.02e-3(3.44e-3)+ | 7.41e-5(3.74e-4)+ | 8.31e-2(4.46e-2)+ | 5.91e-3(9.22e-3)+ | 3.81e-2(2.47e-2)+ | 2.38e-1(2.81e-2)+ | 3.01e-1(2.25e-2) |
| DTLZ3 | 5 | 0.00e+0(0.00e+0)≈ | 0.00e+0(0.00e+0)≈ | 0.00e+0(0.00e+0)≈ | 0.00e+0(0.00e+0)≈ | 0.00e+0(0.00e+0)≈ | 0.00e+0(0.00e+0)≈ | 0.00e+0(0.00e+0) |
| | 10 | 0.00e+0(0.00e+0)≈ | 0.00e+0(0.00e+0)≈ | 0.00e+0(0.00e+0)≈ | 0.00e+0(0.00e+0)≈ | 0.00e+0(0.00e+0)≈ | 0.00e+0(0.00e+0)≈ | 0.00e+0(0.00e+0) |
| | 20 | 0.00e+0(0.00e+0)≈ | 0.00e+0(0.00e+0)≈ | 0.00e+0(0.00e+0)≈ | 0.00e+0(0.00e+0)≈ | 0.00e+0(0.00e+0)≈ | 0.00e+0(0.00e+0)≈ | 0.00e+0(0.00e+0) |
| DTLZ4 | 5 | 2.28e-2(2.65e-2)+ | 2.93e-1(7.80e-2)≈ | 3.02e-1(8.32e-2)≈ | 1.87e-1(5.36e-2)+ | 3.07e-1(5.76e-2)≈ | 2.65e-1(1.11e-1)≈ | 2.49e-1(1.66e-1) |
| | 10 | 4.20e-4(2.03e-3)+ | 6.42e-2(5.54e-2)+ | 8.85e-2(7.53e-2)+ | 6.53e-2(3.42e-2)+ | 1.99e-1(6.05e-2)+ | 2.52e-1(6.75e-2)+ | 3.24e-1(9.98e-2) |
| | 20 | 0.00e+0(0.00e+0)+ | 0.00e+0(0.00e+0)+ | 8.09e-4(2.67e-3)+ | 1.20e-3(5.76e-3)+ | 6.38e-3(8.46e-3)+ | 8.86e-2(6.97e-2)+ | 1.97e-1(1.08e-1) |
| DTLZ5 | 5 | 7.09e-2(2.85e-3)+ | 7.93e-2(2.59e-3)+ | 9.36e-2(1.60e-4)− | 8.00e-2(2.29e-3)+ | 8.58e-2(2.49e-3)+ | 9.14e-2(6.46e-4)+ | 9.27e-2(5.11e-4) |
| | 10 | 7.49e-3(1.04e-2)+ | 2.60e-2(1.04e-2)+ | 8.60e-2(1.99e-3)≈ | 2.54e-2(9.46e-3)+ | 4.66e-2(1.02e-2)+ | 8.48e-2(1.78e-3)≈ | 8.53e-2(2.03e-3) |
| | 20 | 4.12e-5(2.22e-4)+ | 0.00e+0(0.00e+0)+ | 1.00e-2(1.02e-2)+ | 0.00e+0(0.00e+0)+ | 9.09e-4(2.11e-3)+ | 5.09e-2(7.32e-3)+ | 6.15e-2(7.35e-3) |
| DTLZ6 | 5 | 6.52e-2(7.55e-3)+ | 6.06e-3(1.28e-2)+ | 3.10e-2(1.98e-2)+ | 3.56e-3(1.03e-2)+ | 1.93e-2(2.10e-2)+ | 8.70e-2(8.64e-3)− | 7.68e-2(1.94e-2) |
| | 10 | 3.91e-3(7.22e-3)+ | 0.00e+0(0.00e+0)+ | 0.00e+0(0.00e+0)+ | 0.00e+0(0.00e+0)+ | 0.00e+0(0.00e+0)+ | 3.52e-2(2.51e-2)+ | 4.91e-2(2.38e-2) |
| | 20 | 0.00e+0(0.00e+0)≈ | 0.00e+0(0.00e+0)≈ | 0.00e+0(0.00e+0)≈ | 0.00e+0(0.00e+0)≈ | 0.00e+0(0.00e+0)≈ | 0.00e+0(0.00e+0)≈ | 2.06e-3(7.33e-3) |
| DTLZ7 | 5 | 2.29e-1(2.23e-2)+ | 2.82e-1(5.98e-3)+ | 3.08e-1(7.28e-3)− | 1.90e-1(3.80e-2)+ | 2.24e-1(2.41e-2)+ | 2.49e-1(4.23e-2)+ | 2.84e-1(3.96e-2) |
| | 10 | 1.81e-1(4.40e-2)+ | 2.53e-1(9.02e-3)≈ | 2.81e-1(3.28e-2)− | 1.44e-2(2.31e-2)+ | 2.11e-2(2.95e-2)+ | 2.23e-1(3.95e-2)+ | 2.47e-1(3.63e-2) |
| | 20 | 1.59e-1(4.85e-2)+ | 1.56e-1(4.53e-2)+ | 2.21e-1(3.02e-2)≈ | 0.00e+0(0.00e+0)+ | 1.56e-6(8.40e-6)+ | 1.15e-1(4.03e-2)+ | 2.03e-1(4.17e-2) |
| $+/≈/−$ | | 14/7/0 | 12/9/0 | 6/10/5 | 14/7/0 | 13/8/0 | 11/9/1 | |

