# OpenReview forum: "LORA-MOO: Learning Ordinal Relations and Angles for Expensive Many-Objective Optimization"
_NeurIPS.cc/2024/Conference — Submitted to NeurIPS 2024_

### Official Review · Reviewer_97CT · 2024-06-30

**Soundness:** 3
**Presentation:** 3
**Contribution:** 2
**Rating:** 4
**Confidence:** 5

**Summary:**

This paper proposed the LORA-MOO framework, a surrogate-assisted MOO algorithm that learns surrogates from spherical coordinates. This includes an ordinal-regression-based surrogate for 10 convergence and M −1 regression-based surrogates for diversity.

**Strengths:**

The considered problem is pretty important.

**Weaknesses:**

Major:
1. Line 113-116, a bit too repetitive.

2. Line 121, an initial dataset size of 11D-1, too specific.

3. Lines 121-134: the algorithms are described too specifically. It is more suitable for EC journals rather than NeurIPS.

4. Selection Criteria, too simple.

5. What does LORA-MOO means?

6. Some HV-based MOBO methods are not compared as they are failed to solve many objectives. This argument is not accurate, please consider to run "https://github.com/xzhang2523/libmoon/blob/main/libmoon/solver/mobo/run_dirhvego.py", which supports more than ten objectives problems.


Minor:
There are too many grammar errors in this paper.
(1) Line 249, they are failed -> they failed. .

(2) Line 270, HV use -> HV uses .

(3) Line 175, consider using \max.

(4) line 21, are widely exist -> widely exist.

(5) line 64, an non-pa .. -> a non-pa..

**Questions:**

See weakness.

**Limitations:**

See weakness.

---

> ### Author Rebuttal · Authors · 2024-08-06
>
> Weakness 1: Line 113-116, a bit too repetitive.
>
> Response: Thanks for your comment.
>
> We explain the connection between SAEAs and BO in lines 113 - 116 since some less-skilled readers may not be knowledgeable enough about this. Actually, we met many reviewers who do not know the connection between SAEAs and BO.
>
> ---
>
> Weakness 2: Line 121, an initial dataset size of 11D-1, too specific.
>
> Response: Thanks for your comment.
>
> 11D - 1 is a widely used conventional setting for initial datasets in the literature (such as the study recommended by the reviewer in weakness 6 below, line 11 in the linked code page). We use this setting to ensure a fair comparison and also make our work consistent with existing studies.
>
> ---
>
> Weakness 3: Lines 121-134: the algorithms are described too specifically. It is more suitable for EC journals rather than NeurIPS.
>
> Response: Thanks for your comment.
>
> We cannot grasp the point of this comment. As far as we know, many specific papers on our topic have been published in NeurIPS.
>
> ---
>
> Weakness 4: Selection Criteria, too simple.
>
> Response: Thanks for your comment.
>
> 1. We argue that the simplicity of a method can be a weakness. Some famous algorithms are extremely simple but working, e.g. ReLU.
>
> 2. In addition, we would like to explain our selection criteria here.
> - For convergence criterion, the selection criterion is simple since the modeling of dominance relation is relatively complex. Many operations have been done during the modeling procedure, making the selection criterion pretty intuitive and simple.
> - For diversity criterion, we do not think it is simple, we actually have a separate pseudo-code (see Alg. 4 in Appendix C.3) to describe its details.
>
> ---
>
> Weakness 5: What does LORA-MOO means?
>
> Response: Thanks for your comment.
>
> LORA-MOO is the abbreviation of our algorithm, Learning Ordinal Relations and Angles for expensive Many-Objective Optimization (The title of this submission).
>
> ---
>
> Weakness 6: Some HV-based MOBO methods are not compared as they are failed to solve many objectives. This argument is not accurate, please consider to run (we hided the link given by the reviewer because any external link is not allowed to appear in our rebuttals.), which supports more than ten objectives problems.
>
> Response: Thanks for your comment.
>
> 1. We would like to revise the inaccurate statement in our paper ''Some HV-based MOBO methods are not compared as they failed to solve many objectives'' to ''Some HV-based MOBO methods are not compared due to their long runtimes when solving many-objective problems".
>
> 2. We have attempted to run many HV-based MOBO methods for comparison, but it turned out that they were unable to complete a single run of our 10-objective optimization test within one day.
>
> 3. For the recommended link, we have looked into this link and get two questions:
> - We have read the paper associated with this link (Hypervolume Maximization: A Geometric View of Pareto Set Learning, NeurIPS 2023), the paper was proposed for multi-objective optimization and it did not claim its effectiveness on many-objective optimization. In addition, the experiment section conducted only multi-objective optimization experiments.
> - We have checked the library associated with this link (LibMOON), we found that it was a Python library for multi-objective optimization. We have downloaded LibMOON and configured for it, but when we attempted to run the reviewer recommended code (with a new setting of n\_obj = 10), the program reported errors. We have identified that the error was caused due to the optimization problem (e.g. ZDT, DTLZ) files were coded for only multi-objective optimization. In a word, for now, LibMOON did not support many-objective optimization.
>
> Hope our answer meets your expectation, if you have any further suggestions about how to run the recommended code for many-objective optimization problems, please let us know. We are really interested in this.
>
> ---
>
> Weakness Minor:
>
> Response: Thanks for your comment.
>
> We have looked thorough our paper and corrected all grammar errors.

---

> > ### Comment · Reviewer_97CT · 2024-08-11
> > **Thanks for your response**
> >
> > Thanks for your response. I have no questions for this paper.

---

### Official Review · Reviewer_pAi2 · 2024-07-07

**Soundness:** 2
**Presentation:** 3
**Contribution:** 2
**Rating:** 4
**Confidence:** 4

**Summary:**

This paper proposes a surrogate-assisted evolutionary many-objective optimization algorithm, named LORA-MOO. LORA-MOO is composed of a surrogate for ordinal modeling, which focuses on convergence, and m-1 surrogates for distribution modeling, which focus on diversity. Empirical study demonstrates the effectiveness of the proposed algorithm.

**Strengths:**

This paper is well-written and easy to follow. Although the proposed algorithm seems somewhat complicated, all the technical details are clearly presented, and the motivations behind them are explained. The empirical study is generally solid, with all the major parameters included in the ablation study, and most of the commonly used test instances are covered. The results show that LORA-MOO outperforms the baselines.

**Weaknesses:**

Although LORA-MOO obtained better indicator values than the baselines in synthetic problem benchmarks, it is difficult for me to find some fundamental differences between LORA-MOO and previous MOAs. Nor could I see what new insights this paper can provide for solving expensive MOPs. LORA-MOO models the convergence of solutions by a surrogate problem of the domination level, which is a common idea in MOO, adopted by many past methods such as NSGA-II. Such a surrogate is intuitive, but it is unclear to me why it could work better than the existing surrogates such as pairwise relation or function values, and why such a surrogate can be successfully modeled by a Gauss process. LORA-MOO uses m-1 surrogates to predict the spherical coordinate, but it seems identical to predicting function values. LORA-MOO also contains many other components, such as EA, PSO, non-dominated sorting, various clustering methods, and some subset selection mechanisms. These components have long been widely adopted by many MOAs, and there are many alternatives available as well. I agree that these components could usually make an algorithm perform better, but this paper does not adequately demonstrate any necessity for such a combination or any connections between these components. The ablation study appears to be a parameter-tuning experiment, presenting some results under different parameters. However, there is no ablation for the many components in the algorithm, so it is unclear what contributions these components actually make.

Expensive optimization problems are closely connected to real-world applications, and many real-world MOPs are indeed expensive problems. Therefore, I believe this is a very valuable research direction. However, the empirical study in this paper is mainly conducted on synthetic problems. DTLZ and WFG have undoubtedly driven the development of the MOO field, but they have also caused a significant number of researchers to focus narrowly on these synthetic test sets. As a result, there are now many algorithms that perform excellently on synthetic test sets but struggle to adapt to real-world problems. The authors have also conducted tests on NASBench, which is crucial for comprehensively demonstrating the algorithm's capabilities, but the results do not seem to be sufficiently convincing.

**Questions:**

1. Regarding typesetting, I recommend using the Roman font for the operators instead of the italic font. For instance, use $\max$, $\exp$, $\cos$ instead of $max$, $exp$, and $cos$.
2. P2, L45. One of the difficulties of expensive optimization problems is the small data size. Pairwise relations can increase the data size, making it possible to train some DL models. Therefore, this is an advantage of pairwise relations. Additionally, why do the authors believe that the amount of data will increase exponentially? At most, there will only be $O(N^2)$ pairs, so the increase is at most quadratic. Considering that DL models can usually be trained in parallel, I don't think efficiency could be a major drawback. The authors repeatedly claim efficiency advantages (L45, L59), but the experimental results do not demonstrate any efficiency advantage over the baseline.
3. P7, Fig. 1. The caption "Evaluations" is clipped.
4. P7, Line 268. IGD/IGD+ uses a subset sampled from the true PF.
5. Page 9, Fig. 4. Why present the normalized runtime instead of the real time?
6. P17, On the reference point for HV. It is inappropriate to set the reference point for HV to (1,1, ...,1). Firstly, the reference point can only be set near 1 if all solutions have already converged well. Additionally, to better measure diversity, the reference point for HV needs to be larger than the nadir point, so it is generally set to at least $1.1$. For expensive multi-objective optimization problems, the reference point needs to be set to an even larger value because most solutions may not have converged well. For example, in Table 10 and Table 14, the HV values are all 0, indicating that the reference point is set too low.

**Limitations:**

This paper does not summarize its limitations. I suggest the authors reconsider the limitations of this work and fully present them in the paper.

---

> ### Author Rebuttal · Authors · 2024-08-07
>
> Weakness:
>
> Response: Thanks for your comments.
>
> As we listed in our contributions (Section 1), the main differences are: 1) Our ordinal-regression-based model which trained on dominance relations and artificial ordinal relations. 2) The idea of modeling surrogates in spherical coordinates and use these surrogates for diversity maintenance.
>
> As far as we know, our ordinal-regression-based model is completely different from almost all surrogates in the literature. It shows that ordinal relation can be approximated using regression model and its effectiveness. In addition, the modeling with spherical coordinates may inspire readers to solve expensive MOPs in different coordinate systems.
>
> We would like to clarify that the idea of using domination level is common in MOO, but the idea of modeling ordinal dominance relations with regression models is rare in expensive MOO.
> Additionally, NSGA-II is not a model-based optimization algorithm and is not designed for expensive MOO, we are confused that why the reviewer mention NSGA-II when talking about ''modeling dominance level''.
>
> Our surrogate is well designed to approximate ordinal relations, it describe the ordinal landscape of objective space and provide the direction of optimization. The details of how to modeling with GP is presented in Section 3.2. We firstly quantified ordinal relations as numerical values by dominance relations and some artificial relations. Then we use GP to approximate the quantified numerical values.
>
> In fact, spherical coordinate are numerical values, while function values are also numerical values.
>
> We would like to clarify that our ablation studies have demonstrated the contributions of our algorithm components. For example:
> - The ablation study reported in Appendix F.2 has demonstrated the contribution of our $\lambda$-dominance (described in Section 3.2.1). When $\lambda$ = 0, the component of $\lambda$-dominance is actually removed from our algorithm. In Table 4, by comparing the results of $\lambda$ = 0 and $\lambda$ > 0, we can observe the contribution of this component.
> - Similarly, our ablation study in Appendix F.3 shows the contribution of our artificial relations (described in Section 3.2.2) since $rp_{ratio}$ = 1 indicates this component is removed.
> - Our ablation study in Appendix F.4 shows the contribution of our clustering-based initialization (see Section 3.3.1) since $n_c$ = 1 indicates this component is removed.
>
> In summary, we can observe the contributions of our algorithm components directly from our ablation studies.
>
> We are confused why the reviewer think our NAS results are not sufficiently convincing.
> Our NAS experiment shows that our LORA-MOO outperforms the comparison algorithms. While the comparison algorithms tend to be converged, our LORA-MOO is still able to reach greater HV values which are close to the maximal HV value on this problem. It would be appreciated if the reviewer could provide some reasons for this comment.
>
> ---
>
> Question 1:
>
> Response: Thanks for your comment. We have revised our font as recommended.
>
> ---
>
> Question 2:
>
> Response: Thanks for your comment.
>
> Increasing the data size is detrimental to Gaussian Processes models and many other modeling techniques, so we argue that it is an advantage of pairwise relations.
>
> For some DL models, although increasing the data size makes it possible to train DL models, it also increases the time cost of model training. Therefore, it is hard to say increasing the data size is an advantage.
> In addition, pairwise relations are used to train classification surrogates (DL models), however, our experiments show that classification-based optimization algorithms (e.g. CSEA, REMO) are inferior to other regression-based optimization algorithms (e.g. KRVEA, KTA2) in optimization performance.
>
> For the word ''exponentially'', we have revised it to ''quadratically'' for improving the accuracy of our statement.
>
> Although DL models can be trained in parallel, the total computational cost has increased, however, we admit that parallel computation alleviates the curse caused by the increased data size.
>
> The efficiency advantage is shown in Fig. 4. It can be seen that, when the number of objectives increases, LORA-MOO's runtime increases in a slower rate than KRVEA and KTA2's runtime. For remaining comparison algorithms, they train fewer surrogates than LORA-MOO, but their optimization performance are significantly less competitive.
>
> ---
>
> Question 3:
>
> Response: Thanks for your comment. We have plotted a new figure to show x-axis clearly.
>
> ---
>
> Question 4:
>
> Response: Thanks for your comment.
>
> We are confused on this comment.
> The true PF is a set consisting of infinite solutions, how can we compute IGD / IGD+ if we do not use an uniformly sampled subset of the true PF as reference points? As far as we know, all studies in the literature used a subset of the true PF to compute IGD / IGD+, as we described in Appendix. D.
>
> ---
>
> Question 5:
>
> Response: Thanks for your comment.
>
> Comparison algorithms are implemented in different programming languages, e.g. Matlab, Python, so it is impossible to compare the real runtime directly. In addition, for many-objective optimization, it is crucial to investigate the relation between runtime and the number of objectives. If the runtime of an algorithm increases rapidly as the number of objectives increases, then such an algorithm would be unsuitable for optimization problems with too many objectives. After the normalization of runtime, it can be directly observed that how runtime would varies with the number of objectives.
>
> ---
>
> Question 6:
>
> Response: Thanks for your comment. We have re-calculated HV values as suggested.
>
> ---
>
> Limitation:
>
> Response:
>
> Currently, our algorithm pick two solutions (by convergence and diversity) for expensive evaluations in each iteration. A dynamic selection strategy can be developed to select dynamic number of solutions for evaluations to improve the evaluation efficiency.

---

> > ### Comment · Reviewer_pAi2 · 2024-08-12
> >
> > Thank you for your response. However, your rebuttal does to satisfactorily address my concern about the novelty of the proposed method. Therefore, I maintain my score. The authors seem confused about my comments, so I provide some clarifications.
> >
> > > We are confused that why the reviewer mention NSGA-II when talking about ''modeling dominance level''
> >
> > NSGA-II uses a dominance level to directly model convergence. I know LORA-MOO is model-based while NSGA-II is not. LORA-MOO just used a regression-based model to model such a dominance level, so it does not seem to provide new insights about how to model convergence.
> >
> > > Question 4: We are confused on this comment.
> >
> > In Line 268, you say "IGD/IGD+ use a set of truth Pareto fronts". What does "a set of truth Pareto fronts" mean? It is more accurate to express as "IGD/IGD+ uses a subset sampled from the true PF."

---

> > > ### Author Response · Authors · 2024-08-12
> > >
> > > Thanks for your comment.
> > >
> > > We would like to clarify that:
> > > 1. In many-objective optimization, it is impossible to model the dominance level in NSGA-II.
> > > - For a many-objective optimization problem, all solutions in the initial training dataset could be non-dominated, indicating all solutions would locate in the same dominance level. Modeling such a dominance level is useless, it just like to model a classification model with all solutions in the same class.
> > >
> > > 2. The modeling of ordinal relations in LORA-MOO is able to solve the difficulties mentioned above, it is completely different from the dominance level in NSGA-II and it overcomes the drawback of dominance level in NSGA-II.
> > > - As described in Section 3.2, the ordinal relations LORA-MOO learned from training dataset are a mixture of lambda-dominance relations and a clustering-based artificial relations. Both of two relations are novel concepts we proposed in this work.
> > > - In addition, there is an ordinal relation mapping before using them to train models, which makes our model stable.
> > >
> > > 3. The novel ordinal regression model is only a part of our contributions.
> > > - We also proposed a spherical coordinate base modeling method to maintain the diversity of non-dominated solutions.

---

> ### Comment · Reviewer_pAi2 · 2024-08-12
>
> Thank you for your response. I know that the strict dominance relation is not suitable for many-objective scenarios. However, there have been many existing relaxed dominance relations such as $\epsilon$-dominance [1], grid-dominance [2], and $k$-optimality [3]. They are designed for many-obj optimization. It is not clear why the proposed $\lambda$-dominance can be more effective than the existing dominance relations.
>
> In short, LORA-MOO seems to be a recombination of some existing methods or similar ideas. The motivation and necessity for such a combination are not clearly presented. There is no theoretical analysis or ablation study for these components. Therefore, I maintain a negative rating for this paper.
>
> [1] Combining convergence and diversity in evolutionary multiobjective optimization, Evolutionary Computation, 2002.
>
> [2] A grid-based fitness strategy for evolutionary many-objective optimization, GECCO 2010.
>
> [3] A fuzzy definition of ‘optimality’ for many criteria optimization problems, TCYB.

---

> ### Author Response · Authors · 2024-08-12
>
> Thanks for your comment.
>
> We would like to clarify the following issues:
>
> $\textbf{About ablation studies and theoretical analysis}$:
>
> - We do have ablation studies in Section 4.2 and Appendix F. More importantly, our ablation studies have already demonstrated the contributions of these algorithm components (Detailed explanations about this are available in our first rebuttal to weaknesses).
> - We have provided a theoretical runtime analysis in the response to another reviewer.
>
> ---
>
> $\textbf{About contributions}$:
> - We would like to highlight that our modeling method for convergence is called $\textbf{ordinal-regression model}$, $\textbf{NOT dominance-regression model}$. Although the modeling process includes some dominance related relations, it also includes a lot of artificial ordinal relations which are NOT dominance level related relations (described in Section 3.2.2).
> - The proposed ${\lambda}$-dominance is $\textbf{only a minor contribution (about 10\\%) in our work}$. We have more contributions on our clustering-based artificial ordinal relations, mapping of ordinal relations, our spherical coordinate based diversity maintenance strategy, our global search strategy ... We are confused that why the reviewer ignores our main contributions while concerning severely on the similarity between our minor contribution and existing dominance related works.
> - LORA-MOO is not a combination or recombination of existing methods. Many components in LORA-MOO are not available in any existing studies, such as the clustering-based artificial relations. Again, we hope the reviewer can focus on our main contributions mentioned above.
> - There is no motivation for combination since our work is not a combination of existing works.
>
> $\quad$
>
> Thanks for your reading.

---

> > ### Comment · Reviewer_pAi2 · 2024-08-13
> >
> > Thank you for your reply.
> >
> > I said, "There is no ablation study **for these components**". The ablation study in Section 4.2 and Appendix F only discussed the parameters, not the contribution of the components. For, example, you should replace one component with another one to demonstrate this component is necessary.
> >
> > I read your "runtime analysis". It is only a time complexity analysis, not a runtime analysis. I think the authors do not know what "runtime analysis" means. You can search "runtime analysis" on Google Scholar to get some good papers to learn.
> >
> > I keep a negative rating for this paper.

---

> > > ### Author Response · Authors · 2024-08-13
> > >
> > > Thanks for your comment.
> > >
> > > As we explained in our initial rebuttal, we do have ablation studies for $\textbf{these components}$:
> > >
> > > The ablation study reported in Appendix F.2 has demonstrated the contribution of our $\lambda$-dominance (described in Section 3.2.1).
> > > When $\lambda$= 0, the component of $\lambda$-dominance is actually $\textbf{removed}$ from our algorithm.
> > > In Table 4, by comparing the results of $\lambda$ = 0 (performance with normal dominance relations) and $\lambda$ > 0 (performance with this component, $\lambda$-dominance relations), we can observe the contribution of this component and why it is necessary to our algorithm (lines 598-600 in our manuscript).
> > >
> > > Similarly, our ablation study in Appendix F.3 shows the contribution of our artificial relations (described in Section 3.2.2) since $rp_{ratio}$ = 1 indicates this component is $\textbf{removed}$. We have provided explanations about the contribution of this component in lines 618-627.
> > >
> > > Our ablation study in Appendix F.4 shows the contribution of our clustering-based initialization (see Section 3.3.1) since
> > > $n_c$ = 1 indicates this component is $\textbf{removed}$ and replaced by a random initialization. The performance of $n_c$ = 1 and that of $n_c$ > 1 are compared and explained (lines 641-646).
> > >
> > > In summary, we can observe the contributions of our algorithm components directly from our ablation studies.
> > >
> > > $\quad$
> > >
> > > ---
> > >
> > > As for theoretical analysis, we admit it is inappropriate to say runtime analysis, but time complexity analysis is provided.
> > >
> > > $\quad$
> > >
> > > ---
> > >
> > > Finally, we wonder to know if we have solved your concerns on contributions.

---

> ### Comment · Reviewer_pAi2 · 2024-08-13
>
> Thank you for your response. I increased my rating. However, it is still not clear to me what new insights this paper could provide in solving expensive optimization problems. I suggest the authors submit this manuscript to TEC rather than NeruIPS. Maybe the audiences there are more interested in this paper.

---

### Official Review · Reviewer_wux8 · 2024-07-10

**Soundness:** 2
**Presentation:** 1
**Contribution:** 2
**Rating:** 3
**Confidence:** 4

**Summary:**

This paper proposes a novel surrogate-assisted evolutionary algorithm named LORA-MOO, the core contribution is the introduce of ordinal-regression-based model  spherical coordinates approximation to SAEA and LORA-MOO can find a good trade-off between optimization efficiency and optimization results.

**Strengths:**

This paper provides a novel perspective for modeling surrogates with high efficiency.

The experiments results seem good.

**Weaknesses:**

Motivation and contributions are limited. To the best of my knowledge, the main contrition of LORA-MOO is introducing ordinal-regression-based model for convergence and spherical coordinates for diversity. However, why do it and what is the connections between them? Besides, the manuscript includes a lot of informal expression. The proposed method is complex and effectiveness is limited.

**Questions:**

a. Why is Many-objective optimization problems called MOOPs for short? What's the abbreviation of Multi-objective optimization problems？I think the author needs to consult more relevant literature to make the expression more formal.

b. Considering that the author mentioned MOBO, but there was no comparison in the experimental stage, it is interesting to provide a comparison with PSL-MOBO[1], qNEHVI[2], DAPSL[3] and so on. In addition, more SOTA methods should be introduced to demonstrate the superiority of LORA-MOO since almost all the compared methods is out of date.

[1] Lin, Xi, et al. "Pareto set learning for expensive multi-objective optimization." Advances in Neural Information Processing Systems 35 (2022): 19231-19247.

[2] Daulton, S.; Balandat, M.; and Bakshy, E. 2021. Parallel bayesian optimization of multiple noisy objectives with expected hypervolume improvement. Advances in Neural Information Processing Systems, 34: 2187–2200.

[3] Lu, Yongfan, Bingdong Li, and Aimin Zhou. "Are You Concerned about Limited Function Evaluations: Data-Augmented Pareto Set Learning for Expensive Multi-Objective Optimization." Proceedings of the AAAI Conference on Artificial Intelligence. Vol. 38. No. 13. 2024.

c.  In all the tables, the author doesn't have any highlights, which can't let readers know what LORA-MOO is good at or not. Moreover, most tables in Appendix miss the comparison (=,-,+) in terms of a single question

d. In the experiments section, I only saw some results presented, but I didn't see any in-depth analysis.

e. Lack of Real-World Application Depth: Although a real-world network architecture search problem is mentioned, the paper does not delve deeply into real-world applications. More case studies or industrial applications should be considered [4].

f. Lack of Theoretical Analysis: The paper focuses heavily on empirical results but lacks a rigorous theoretical analysis to support the empirical findings. Theoretical results, such as convergence proofs or time complexity analysis, would strengthen the paper's contributions.

g．The idea of the introduce of spherical coordinates has been discussed in [5]. What is the main difference between [5] and this paper?
[5] Zhang, Xiaoyuan, et al. "Hypervolume maximization: A geometric view of pareto set learning." Advances in Neural Information Processing Systems 36 (2024).

h. The motivation is not enough. It seems that the main target of LORA-MOO is to enhance efficiency via only training a single model. However, I can’t find obvious advantage in terms of Runtime Comparison. There is not enough motivation to support the introduction of A. There is not enough motivation to support the introduction of spherical coordinates.

i. Why PSO is introduced to conduct offspring generation?

j. Why use S_A as the input of the Kriging model (Algorithm 1)? In my understanding, S_o and S_a are enough for construction of Kriging model.

k. In each iteration, only two solutions (one for convergence and one for diversity) are evaluated. So, why not design a dynamic strategy to determine the number of evaluation solutions considering that the number of promising solutions is varying with the population evoluting.

**Limitations:**

See questions.

---

> ### Author Rebuttal · Authors · 2024-08-07
>
> Weakness:
>
> Response: Thanks for your comment.
>
> We were confused that why the reviewer thought our contributions were limited. Our work proposed a novel model and a novel optimization method. In addition, we noticed that the reviewer was the only reviewer who thought our presentation was not good.
>
> The reason to introducing ordinal-regression-based model is to assist convergence and optimization search via a single model. If the most workload of model-based optimization is completed by a single model, the efficiency of optimization algorithm would be enhanced. For spherical coordinates, they are employed to maintain diversity but will be used only once at the end of optimization search in each iteration. These spherical coordinate models are used with low frequency to improve computational efficiency without compromising solution diversity.
>
> Our experiments show that our algorithm outperforms all the comparison algorithms, we would appreciate it if the reviewer could provide some reasons on the comment of ''effectiveness is limited''.
>
> ---
>
> Questions a:
>
> Response: Thanks for your comment.
>
> Many-objective optimization problems are called MaOOPs for short to distinguish itself from multi-objective optimization problems. However, the abbreviation of many-objective optimization and the abbreviation of multi-objective optimization did not appear in our paper simultaneously, so it is not necessary to make such a difference in abbreviations.
>
> However, we appreciate the reviewer' attitude toward formal expressions and have revised our abbreviations.
>
> ---
>
> Questions b:
>
> Response: Thanks for your comment.
>
> The topic of our work is many-objective optimization instead of multi-objective optimization. As far as we can see, the references mentioned above are all designed for multi-objective optimization.
>
> We actually attempted to run some Multi-objective BO for comparison purpose. However, recent MOBO are mainly Hypervolume-based optimization algorithm, and the computation of hypervolume is very time-consuming for many-objective problems. Our attempts failed as it costed more than one day to complete a single run of MOBO methods for 10-objective problems. We mentioned this in Section 4.1, line 250.
>
> ---
>
> Questions c:
>
> Response: Thanks for your comment.
>
> - For the table in the main paper and the last 8 tables in the Appendix, we have presented the statistical test results between LORA-MOO and every compared algorithm, denoted by symbols +, -, and $\approx$. Readers can understand LORA-MOO performance via these symbols.
> - For 4 tables about ablation studies, we do not have statistical result symbols for each row since the statistical tests are conducted between all LORA-MOO variants. For example, each variant in Table 3 needs to be compared with all other 4 variants. It is impossible to put 4 statistical test results in 1 cell. Therefore, we put the summary of statistical test results at the end of these 4 ablation studies' tables.
>
> We have added highlights to ablation studies' tables for improving clarity.
>
> ---
>
> Questions d:
>
> Response: Thanks for your comment.
>
> Due to the page limitation, we can only report important results in the main paper, but readers can find our ablation studies and in-depth analysis in Appendix F.
>
> ---
>
> Questions e:
>
> Response: Thanks for your comment.
>
> We are conducting more real-world experiments and would report them soon.
>
> ---
>
> Questions f:
>
> Response: Thanks for your comment.
>
> We admit that some theoretical analysis would strengthen our paper. This is a limitation of our work.
>
> ---
>
> Questions g:
>
> Response:  Thanks for your comment.
>
> We have looked into the referred paper but we find very limited similarity between our work and the referred paper. The referred paper is a Hypervolume-based algorithm and it does not approximate spherical coordinates. Oppositely, its model requires an input of spherical coordinate and outputs a solution, but our model inputs solutions and outputs spherical coordinates. We use spherical coordinates in completely different way, the only similarity between our works is that both of us mentioned spherical coordinates in our own work.
>
> ---
>
> Questions h:
>
> Response: Thanks for your comment.
>
> Efficiency is one motivation for regression-based SAEAs. For classification-based SAEAs and previous ordinal-regression-based SAEAs, although they train very few surrogates and are thus efficient, their limited number of surrogates are unable to provide enough information for diversity maintenance. Therefore, these SAEAs have poor performance in when the number of objectives is large.
>
> Based on above two motivations, we make a trade-off and proposed LORA-MOO which is efficient and effective on diversity maintenance (our contributions 1 and 2 that are listed in Section 1).
> From Fig 4, it can be observed that our runtime is shorter than regression-based SAEAs such as KRVEA and KTA2, but our optimization performance is better than other algorithms such as REMO and CSEA.
>
> ---
>
> Questions i:
>
> Response: Thanks for your comment.
>
> PSO is only used as an optimizer for our algorithm. We did not make any modification to PSO or claim any contribution on PSO, it is not a novelty of our work. PSO can be replaced with any evolutionary optimizer for our algorithm.
>
> ---
>
> Questions j:
>
> Response: Thanks for your comment.
>
> $S\_o$ contain only ordinal values and $S_a$ contains only angular coordinates, from the perspective of model training, they are both numerical labels. We need $S_A$ to provide corresponding variables $\textbf{x}$.
>
> ---
>
> Questions k:
>
> Response: Thanks for your comment.
>
> Actually a dynamic strategy is our further work. One of our limitation is the static strategy of LORA-MOO.
>
> ---

---

> > ### Comment · Reviewer_wux8 · 2024-08-12
> >
> > Thank you for your response. My concerns have been partially addressed. Since the lack of additional experiments and theoretical analysis, I have decided to maintain my score.

---

> ### Author Response · Authors · 2024-08-12
>
> Thanks for your comment.
>
> 1. $\textbf{About Experiment}$:
>
> We have conducted a new real-world NAS experiment with different network architecture and 8 objectives (error, parameters, flops, edge gpu latency, edge gpu energy, eyeriss latency, eyeriss energy, and eyeriss arithmetic intensity). The comparison results are consistent with the real-world problem we reported in our manuscript. Our LORA-MOO outperforms the comparison algorithms and have reached a mean HV value of 0.5776 over 30 independent runs. We would add the corresponding figure and descriptions to our manuscript (As we are unable to add figures in our comment now).
>
> ---
>
> 2. $\textbf{Theoretical Analysis}$:
>
> We attempt to add the following time complexity analysis:
>
> Notations:
> - n: the number of training samples.
> - N: the number of test samples.
> - m: the number of objectives.
> - g: the number of generations for reproducing candidate solutions.
> - p: the population size for a generation.
>
> $\quad $
>
> The model used in LORA-MOO is Gaussian Process, the training time complexity is analyzed as follows:
> - Time complexity of covariance matrix computation is $O(n^2)$.
> - Time complexity of Cholesky decomposition and computation of likelihood: $O(n^3)$.
>
> The prediction time complexity is analyzed as follows:
> - Time complexity of computing the covariance between test sample and training samples: $O(n*N)$.
> - The time complexity of predicting the mean:  $O(n*N)$.
> - The time complexity of predicting the variance: $O(n^2*N)$.
>
> In summary, the overall training complexity is $O(n^3)$, and the overall prediction complexity is $O(n^2*N)$.
>
> $\quad $
>
> Now we analyze the time complexity of model-based optimization algorithms, for each iteration, the number of test samples is $p * g$, so the total number of test samples is approximately $N = n * p * g$.
>
> 1. For LORA-MOO, for a $m$-objective optimization problem:
> - The time complexity of training an ordinal model and $m-1$ angular models is $O(n^3* m)$.
> - The time complexity of prediction in the ordinal model is $O(n^3 * g * p)$.
> - The time complexity of prediction in $m-1$ angular models: $O(n^3 * p * (m-1))$.
> - The overtime time complexity in models for LORA-MOO:
> $O(n^3 * (m + g * p + p * m - p)) \approx O(n^3 * (p * g + p * m)) \approx O(n^3 * p * (m + g))$
>
> 2. In comparison, for other optimization algorithms with $m$ surrogate models:
> - The time complexity of training $m$ models: $O(n^3 * m)$
> - The time complexity of prediction: $O(n^3 * g * p * m)$.
> - Time overtime time complexity in $m$ models: $O(n^3 * (m * (1 + g * p))) \approx O(n^3 * p * m * g))$
>
> 3. For other optimization algorithms with only one surrogate model:
> - Time overtime time complexity: $O(n^3 * (1 + g * p)) \approx O(n^3 * p * g)$
>
> $\quad $
>
> Therefore, increasing the number of objectives $m$ has limited impact on the time cost of LORA-MOO ($O(n^3 * p * (m + g))$), but for the comparison algorithms with $m$ surrogate models, their time cost would increase rapidly ($O(n^3 * p * m * g)$).
>
> Although LORA-MOO has $m$ surrogate models in total, its time complexity does not significantly larger than the time complexity of optimization algorithms with only one surrogate model ($O(n^3 * p * g)$).

---

### Official Review · Reviewer_TSjm · 2024-07-13

**Soundness:** 3
**Presentation:** 4
**Contribution:** 2
**Rating:** 4
**Confidence:** 4

**Summary:**

This paper introduces a surrogate assisted method for multi-objective optimization. The approach learns a surrogate function with the ordinal values as the regression labels. The ordinal values are generated using a iterative algorithm with the most dominated solutions having the highest ordinal values. The ordinal values are used to train a Kriging model used to select a point for observation using the convergence criterion. Another point is selected using the Kriging model trained on spherical coordinates via a diversity criterion. The approach has several parameters which are tuned via experimentation on real and benchmark datasets.

**Strengths:**

- The approach provides an innovative way for optimizing multi-objective functions by separating out the two objectives thus simplyfying the problem.
  1) The convergence objective designed to select the best solution wrt the ordinal values
  2) The diversity objective designed to improve the diversity of the Pareto optimal solutions
- It is experimentally shown that the method improves the IGD metric compared to several benchmark and real MOO problems.
- The ideas presented in the paper are well motivated and well presented.

**Weaknesses:**

- The approach presented in the paper is not sufficiently novel. Ordinal regression for multi-objective optimization has been studied before [1]. The differences with related prior work have not been discussed in detail.
- The proposed algorithm has many tunable parameters, and it is unclear how the parameters affect performance on real world problems when they have only been tuned on benchmark problems.
- The real world experiment on NAS shows improved regret eventually, but converges slower than other existing approaches. It is difficult to judge on the effectiveness of this approach based on a single experiment. Experiments on more real world optimization problems are necessary to make a conclusion.
- The paper is missing several notable MOO approaches from the Bayesian optimization community [2,3,4,5].

[1] Yu, Xunzhao, et al. "Domination-based ordinal regression for expensive multi-objective optimization." 2019 IEEE symposium series on computational intelligence (SSCI). IEEE, 2019.

[2] Tu, Ben, et al. "Joint entropy search for multi-objective bayesian optimization." Advances in Neural Information Processing Systems 35 (2022): 9922-9938.

[3] Zhang, Richard, and Daniel Golovin. "Random hypervolume scalarizations for provable multi-objective black box optimization." International conference on machine learning. PMLR, 2020.

[4] Paria, Biswajit, Kirthevasan Kandasamy, and Barnabás Póczos. "A flexible framework for multi-objective bayesian optimization using random scalarizations." Uncertainty in Artificial Intelligence. PMLR, 2020.

[5] Abdolshah, Majid, et al. "Multi-objective Bayesian optimisation with preferences over objectives." Advances in neural information processing systems 32 (2019).

**Questions:**

What parameters were used for the NAS problem and how were they tuned? Do parameters tuned on benchmark problems generalize to real world problems?

---

> ### Author Rebuttal · Authors · 2024-08-07
>
> Weakness 1: The approach presented in the paper is not sufficiently novel. Ordinal regression for multi-objective optimization has been studied before [1]. The differences with related prior work have not been discussed in detail.
>
> Response: Thanks for your comment.
>
> We would like to add the following explanations to Section 2 to emphasize our novelties:
>
> A specific drawback of [1] is the lack of information regarding solution distribution, which results in poor optimization performance when the number of objectives is large. In other words, the method proposed in [1] lacks an efficient diversity maintenance strategy, making it unsuitable for many-objective optimization.
>
> Our work is designed for many-objective optimization. We have the following novelties when compared with [1]:
> - We introduced $\lambda$-dominance to simplify the quantification of ordinal relations.
> - We added artificial relations to alleviate the imbalance of training sets caused by the increasing number of objectives.
> - We developed a spherical coordinate based diversity maintenance strategy to improve the diversity of obtained non-dominated solutions.
> - The reproduction and selection methods are quite different. Our LORA-MOO uses global search, while [1] contains local search.
>
> ---
>
> Weakness 2: The proposed algorithm has many tunable parameters, and it is unclear how the parameters affect performance on real world problems when they have only been tuned on benchmark problems.
>
> Response: Thanks for your comment.
>
> Our benchmark problems have covered diverse features of optimization problems (such as unimodal, multimodal, scaled functions, degenerated Pareto front, shifted Pareto front, and disconnected Pareto front) and we have conducted comprehensive ablation studies on them. When looking at a benchmark problem with specific features, we can observe and conclude how tunable parameters affect the optimization performance on problems with these features.
> Therefore, if we obtained any prior knowledge about what features a real-world problem have, we might be able to tune parameters accordingly (based on the experience we concluded from benchmark problems).
>
> In addition, we argue that it is not proper to tune parameters on real-world problems because for real-world expensive optimization problems, it is unrealistic to tune parameters on them before solving them. The cost of tuning parameters on real expensive problems is unaffordable.
>
> ---
>
> Weakness 3: The real world experiment on NAS shows improved regret eventually, but converges slower than other existing approaches. It is difficult to judge on the effectiveness of this approach based on a single experiment. Experiments on more real world optimization problems are necessary to make a conclusion.
>
> Response: Thanks for your comment.
>
> The compared algorithms converge quickly at the early stage due to their local search strategies (e.g. KTA2 uses only optimal evaluated solutions to reproduce new candidate solutions). However, they both are affected adversely by the side effect of local search: When the number of evaluations reaches 200, they tend to be trapped in local optima.
>
> In comparison, our LORA-MOO uses a global search strategy, so it would be relatively slow at the beginning but converges continuously during the optimization. It can be observed that the convergence speed of LORA-MOO does not slow down when the number of evaluations reaches 300, in comparison, the comparison algorithms have low convergence speeds at that time.
>
> ---
>
> Weakness 4: The paper is missing several notable MOO approaches from the Bayesian optimization community [2,3,4,5].
>
> Response: Thanks for your comment.
>
> We have added these references to our Section 2.2.
>
> ---
>
> Question: What parameters were used for the NAS problem and how were they tuned? Do parameters tuned on benchmark problems generalize to real world problems?
>
> Response:  Thanks for your comment.
>
> We used the parameters we tuned on benchmark problems on the NAS problem directly. The detailed parameter settings are available in Section 4.1 and Appendix F.
>
> It should be noted that a given parameter setting could lead to better results on some optimization problems but also could lead to worse results on other optimization problems. Just like the No Free Lunch rule, it is impossible to find a parameter setting that is optimal to all optimization problems.
>
> Considering our benchmark problems have covered diverse features of optimization problems and we have conducted comprehensive ablation studies on them, we think our parameter setting is optimal for most optimization problems. Therefore, we did not tune our parameters for the NAS problem.

---

> > ### Comment · Reviewer_TSjm · 2024-08-10
> >
> > Thanks for the response. I am keeping my current score as
> > - We cannot draw a concrete conclusion based on results from a single real world experiment.
> > - While there are several improvements proposed in this paper over prior work, the improvements are relatively minor.

---

> > > ### Author Response · Authors · 2024-08-10
> > >
> > > Thanks for your comments.
> > >
> > > ---
> > >
> > > About real-world experiment:
> > >
> > > We have conducted a new real-world NAS experiment with different network architecture and 8 objectives (error, parameters, flops, edge gpu latency, edge gpu energy, eyeriss latency, eyeriss energy, and eyeriss arithmetic intensity). The comparison results are consistent with the real-world problem we reported in our manuscript. Our LORA-MOO outperforms the comparison algorithms and have reached a mean HV value of 0.5776 over 30 independent runs. We would add the corresponding figure and descriptions to our manuscript.
> > >
> > > ---
> > > About contributions:
> > >
> > > We would like to clarify that:
> > > - Only the first difference between our LORA-MOO and OREA we listed above is an improvement over prior work, and this contribution is a very minor contribution in our work.
> > > - The remaining three differences (1.Clustering-based artificial relations, 2.Modeling angles in the spherical coordinate system for diversity maintenance, and 3. A novel global search method with novel initialization and reproduction strategies) are our main contributions. These three contributions are completely new ideas we developed to solve many-objective optimization problems, they do not exist in prior studies and they are not improvements of other works.
> > > - The development of our clustering-based artificial relations defines a novel way to learn ordinal-regression models. It distinguishes our ordinal model from prior work.
> > > - The development of our angle modeling method provides a novel way to maintain diversity, which is different from existing studies.
> > > - OREA was not designed for many-objective optimization, it can be observed from our experiments that OREA only works well on multi-objective problems, while our LORA-MOO outperforms OREA significantly on many-objective optimization problems.
> > >
> > > Although we used ordinal regression model in our work, we hope the reviewer can take a look at our other three main contributions. Just like there are many different classification models in diverse studies, ordinal regression models can also be different and diverse.
> > >
> > > Thanks.

---

### Decision · Program_Chairs · 2024-09-25

**Decision:**

Reject

**Comment:**

This paper develops and studies a surrogate-guided multi-objective optimization approach. The key idea is to use ordinal values as the regression labels to train surrogates, and use them to achieve better trade-off between efficiency of optimization and quality of solutions.

There is a consensus among the reviewers' that the paper is lacking in the following aspects and is not ready for publication.
- Limited novelty as proposed improvements are incremental with respect to prior work.
- Experimental evaluation is limited especially on real-world problems to draw strong conclusions.
- The paper needs to position/contextualize the proposed work better with respect to prior work.

For all the above reasons, I'm recommending to reject this paper and encourage the authors' to improve the paper based on the feedback from reviewers'.